

# Superspin chains from superstring theory

**Nafiz Ishtiaque[1][*], Seyed F. Moosavian[2], Surya Raghavendran[3] and Junya Yagi[4]**

**1** Institute for Advanced Study, Princeton, NJ 08540 USA
**2** Department of Physics, McGill University, Montréal, QC H3A 2T8 Canada
**3** Perimeter Institute for Theoretical Physics, Waterloo, ON N2L 2Y5 Canada
**4** Yau Mathematical Sciences Center, Tsinghua University, Beijing 100084 China

[*] nishtiaque@ias.edu

## Abstract

We present a correspondence between two-dimensional $\mathcal{N} = (2, 2)$ supersymmetric gauge theories and rational integrable $\mathfrak{gl}(m|n)$ spin chains with spin variables taking values in Verma modules. To explain this correspondence, we realize the gauge theories as configurations of branes in string theory and map them by dualities to brane configurations that realize line defects in four-dimensional Chern–Simons theory with gauge group $GL(m|n)$. The latter configurations embed the superspin chains into superstring theory. We also provide a string theory derivation of a similar correspondence, proposed by Nekrasov, for rational $\mathfrak{gl}(m|n)$ spin chains with spins valued in finite-dimensional representations.

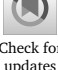

# 1  Introduction

The Bethe/gauge correspondence, discovered by Nekrasov and Shatashvili [1,2] in 2009, connects two seemingly unrelated areas of physics. The Bethe side of the correspondence refers to one-dimensional integrable quantum spin chains. The gauge side is supersymmetric gauge theories.

Arguably the most prominent example of the Bethe/gauge correspondence involves Heisenberg's XXX spin chain and its generalizations. In this example, the eigenvectors of commuting conserved charges of a rational $\mathfrak{gl}(m)$ spin chain are identified with the vacua of a family of gauge theories whose gauge group is the product of $m-1$ unitary gauge groups. The gauge theories have $\mathcal{N} = (4,4)$ supersymmetry broken to $\mathcal{N} = (2,2)$ subgroup by mass deformations, and their gauge and matter contents are encoded in quiver diagrams whose underlying graphs contain the Dynkin diagram of type $A_{m-1}$ as a subgraph.

In 2018, Nekrasov [3] presented a generalization of the above correspondence where the relevant spin chains carry superspins, namely rational $\mathfrak{gl}(m|n)$ spin chains. The corresponding gauge theories are essentially $\mathcal{N} = (2,2)$ supersymmetric, as opposed to having softly broken $\mathcal{N} = (4,4)$ supersymmetry. One of the main results of this paper is an explanation of the origin of this correspondence using superstring theory.

In fact, the goal of the present work is much more ambitious: we wish to place the Bethe/gauge correspondence for superspin chains into a large web of dualities that relate diverse phenomena in which the same superspin chains arise from different supersymmetric gauge theories in various spacetime dimensions.

Many of the phenomena that are expected to constitute this web of dualities are yet to be uncovered, but their specializations to the case of $\mathfrak{gl}(m|0) = \mathfrak{gl}(m)$ are known and have been studied in recent years. Besides the Bethe/gauge correspondence already described, the structures of rational $\mathfrak{gl}(m)$ spin chains (and their trigonometric and elliptic generalizations) have appeared in quantization of the Seiberg–Witten geometries of four-dimensional $\mathcal{N} = 2$ supersymmetric gauge theories [4–6], the action of surface and line defects on supersymmetric indices of four-dimensional supersymmetric gauge theories [7–12], quantization of the Coulomb branches of three-dimensional $\mathcal{N} = 4$ supersymmetric gauge theories [13, 14], and correlation functions of local operators on interfaces in four-dimensional $\mathcal{N} = 4$ super Yang–Mills theory [15], to name a few.

All of these gauge theory setups have realization in string theory, and one suspects that they are related to each other in one way or another via string dualities. This idea has turned out to be true. It was argued in [16] that brane constructions of these setups (except for the last one which we expect is also related) are all dual to brane configurations that realize line defects in a four-dimensional analog of Chern–Simons theory [17–19] with gauge group $GL(m)$. This theory only has a bosonic gauge field, but it is secretly supersymmetric. Indeed, it is equivalent to a holomorphic–topological twist of six-dimensional $\mathcal{N} = (1,1)$ super Yang–Mills theory with gauge group $U(m)$ in the presence of $\Omega$-deformation [20–24]. The six-dimensional theory describes the low-energy dynamics of a stack of $m$ D5-branes, which comprise part of the brane configurations.

Four-dimensional Chern–Simons theory placed on $\mathbb{R}^2 \times \mathbb{C}$ is topological on the plane $\mathbb{R}^2$ and holomorphic on the complex plane $\mathbb{C}$. Due to this holomorphic–topological property, line defects extending along $\mathbb{R}^2$ automatically satisfy the Yang–Baxter relation. Moreover, each line defect carries one complex parameter, its position in $\mathbb{C}$. These facts imply that line defects making up a square lattice in $\mathbb{R}^2$ defines a two-dimensional classical integrable lattice model; their correlation function equals the partition function of the lattice model.

Equivalently, with one of the lattice directions regarded as a time direction, a lattice of line defects in four-dimensional Chern–Simons theory defines a one-dimensional quantum integrable spin chain. For Wilson lines, this spin chain is a rational $\mathfrak{gl}(m)$ spin chain [17–19]. (If one replaces $\mathbb{C}$ in spacetime with $\mathbb{C} \setminus \{0\}$ or an elliptic curve, then one obtains trigonometric or elliptic $\mathfrak{gl}(m)$ spin chain, respectively.) Thus, the fact that the gauge theory setups mentioned above are all dual to line defects in four-dimensional Chern–Simons theory explains the appearance of $\mathfrak{gl}(m)$ spin chains in these setups.

Now, in view of the Bethe/gauge correspondence for rational $\mathfrak{gl}(m|n)$ spin chains, one wonders how one can incorporate it into the picture just described. If one could generalize the string theory realization of the Bethe/gauge correspondence for bosonic spin chains to the superspin chain case, one would generalize, implicitly by string dualities, *all* of the gauge theory phenomena mentioned above to their $\mathfrak{gl}(m|n)$ versions. This is what we aim to achieve.

In this paper, we provide brane constructions of the gauge theories pertinent to the Bethe/gauge correspondence for rational $\mathfrak{gl}(m|n)$ spin chains, and show that they are related by dualities to line defects in four-dimensional Chern–Simons theory with gauge group

GL($m|n$). The latter theory is obtained from two copies of $\Omega$-deformed six-dimensional $\mathcal{N} = (1,1)$ super Yang–Mills theory, with gauge groups U($m$) and U($n$), coupled by a four-dimensional hypermultiplet in the bifundamental representation of U($m$) × U($n$). In turn, this gauge theory setup arises from a stack of $m$ D5-branes intersecting a stack of $n$ D5-branes. This brane construction is another main result of the paper.

Actually, we present two versions of the Bethe/gauge correspondence, one for compact spin chains and one for noncompact spin chains. The difference is whether spin variables are valued in finite-dimensional or infinite-dimensional representations.

We introduce the Bethe/gauge correspondence for noncompact rational $\mathfrak{gl}(m|n)$ spin chains in section 2. The brane constructions of the corresponding gauge theories, as well as the duality relating these theories to line defects in four-dimensional Chern–Simons theory with gauge group GL($m|n$), are discussed in section 3. Discussions in this section provide a string theory explanation for the Bethe/gauge correspondence. The case of compact spin chains is treated in section 4, where we reproduce the correspondence proposed in [3].

It should be remarked that in an inspiring paper [25] in 2010, Orlando and Reffert found the Bethe/gauge correspondence for the rational $\mathfrak{gl}(1|2)$ spin chain with spins taking values in the natural (1|2)-dimensional representation $\mathbb{C}^{1|2}$. Furthermore, they gave a string theory argument to explain dualities between different families of gauge theories corresponding to different choices of Dynkin diagrams of $\mathfrak{gl}(1|2)$. On the spin chain side, these dualities are known as fermionic dualities. In section 3.4 we discuss the fermionic dualities for rational $\mathfrak{gl}(m|n)$ spin chains from a similar point of view.

As we mentioned above, four-dimensional Chern–Simons theory with gauge supergroup can be constructed from two copies of six-dimensional super Yang–Mills theory coupled by four-dimensional matter fields. There is a related construction in topological string theory, which may prove useful in future attempts to put some of the physical arguments given in this work on a rigorous mathematical footing. We describe the topological string construction in appendix A.

The present work unveils only a small part of a collection of phenomena in which superspin chains emerge from supersymmetric gauge theories. It will be extremely interesting to study other, but ultimately related, phenomena whose existence is predicted by string dualities and other tools. We conclude this introduction by stating mathematical conjectures as examples of such phenomena.

In section 2, we define a family of $\mathcal{N} = (2,2)$ supersymmetric gauge theories labeled by the set of ($m + n - 1$)-tuples of nonnegative integers $\mathbb{Z}_{\geq 0}^{m+n-1}$, corresponding to a closed rational $\mathfrak{gl}(m|n)$ spin chain of length $L$ with spins valued in Verma modules. If we turn off all mass parameters and turn on appropriate Fayet–Iliopoulos (FI) parameters, these theories are described in the infrared by effective sigma models. The target space of the sigma model with label $\mathbf{M} = (M_1, \ldots, M_{m+n-1})$ is a Calabi–Yau manifold $\mathcal{M}(\mathbf{M})$ with an action of GL($L$)$^{m+n}$ × GL(1). The topological A-twist of this sigma model, with mass parameters associated with the maximal torus $T$ of GL($L$)$^{m+n}$ × GL(1) turned on, is equivalent to the sector of the spin chain in which there are $M_r$ magnons of type $r$. The highest weights of the Verma modules are determined by the mass parameters.

**Conjecture 1.** The direct sum of equivariant cohomology groups

$$\bigoplus_{\mathbf{M} \in \mathbb{Z}_{\geq 0}^{m+n-1}} H_T\big(\mathcal{M}(\mathbf{M})\big) \tag{1}$$

is a module over $Y(\mathfrak{gl}(m|n))$, isomorphic to the tensor product of $L$ evaluation modules obtained from the Verma modules.

**Conjecture 2.** There is a homomorphism from the Bethe algebra of the Yangian $Y(\mathfrak{gl}(m|n))$ to the direct sum of equivariant quantum cohomology rings

$$\bigoplus_{\mathbf{M}\in\mathbb{Z}_{\geq 0}^{m+n-1}} QH_T\big(\mathcal{M}(\mathbf{M})\big). \tag{2}$$

The first conjecture says that the Hilbert space of states of the A-model is the same as that of the spin chain. For $n = 0$ and $L = 1$, the conjecture is proved in [26]. The second conjecture means that the algebra of local operators of the A-model includes the algebra generated by the commuting conserved charges of the spin chain. For $n = 0$, this conjecture follows from a result of [27].

Similar conjectures can be made for the target spaces of effective sigma models corresponding to compact rational $\mathfrak{gl}(m|n)$ spin chains. The brane configurations in the compact case have been recently considered by Rimanyi and Rozansky [28] from the perspective of geometric construction of R-matrices [27], so we expect that the above conjectures also hold if the target spaces are varieties defined in [28].

# 2 Bethe/gauge correspondence for noncompact superspin chains

The Bethe/gauge correspondence for superspin chains relates a closed spin chain with $\mathrm{GL}(m|n)$ symmetry and two-dimensional gauge theories with $\mathcal{N} = (2, 2)$ supersymmetry. In this section we present a version of the correspondence in which the spin chain consists of spins taking values in infinite-dimensional highest-weight representations of $\mathfrak{gl}(m|n)$. After reviewing some basic facts about $\mathfrak{gl}(m|n)$ and its Verma modules, we introduce the spin chain and its Bethe equations. Then, we introduce the gauge theories and their vacuum equations, and explain in what sense the two sides are equivalent.

## 2.1 $\mathfrak{gl}(m|n)$ and its Verma modules

To begin with, let us review the structures of $\mathfrak{gl}(m|n)$ and its Verma modules, with emphasis on aspects that are important for the Bethe/gauge correspondence.

### 2.1.1 Lie superalgebra $\mathfrak{gl}(m|n)$

Let $\mathbb{C}^{m|n}$ be the vector space graded by $\mathbb{Z}_2 = \{\bar{0}, \bar{1}\}$ whose even subspace $\mathbb{C}_{\bar{0}}^{m|n} = \mathbb{C}^m$ and odd subspace $\mathbb{C}_{\bar{1}}^{m|n} = \mathbb{C}^n$. Let $(b_1, \ldots, b_m)$ and $(f_1, \ldots, f_n)$ be the standard basis of $\mathbb{C}^m$ and that of $\mathbb{C}^n$, respectively. Throughout this section and the next section except section 3.4, we fix an ordered basis $(e_1, \ldots, e_{m+n})$ of $\mathbb{C}^{m|n}$ that is a permutation of $(b_1, \ldots, b_m, f_1, \ldots, f_n)$. The corresponding $\mathbb{Z}_2$-grading $[-]\colon \{1, \ldots, m+n\} \to \mathbb{Z}_2$ is defined by

$$[i] = \begin{cases} \bar{0} & (e_i \in \mathbb{C}^m); \\ \bar{1} & (e_i \in \mathbb{C}^n). \end{cases} \tag{3}$$

The space of endomorphisms $\mathrm{End}(\mathbb{C}^{m|n})$ of $\mathbb{C}^{m|n}$ is also a $\mathbb{Z}_2$-graded vector space, with the even subspace

$$\mathrm{End}(\mathbb{C}^{m|n})_{\bar{0}} = \mathrm{Hom}(\mathbb{C}^m, \mathbb{C}^m) \oplus \mathrm{Hom}(\mathbb{C}^n, \mathbb{C}^n) \tag{4}$$

and the odd subspace

$$\mathrm{End}(\mathbb{C}^{m|n})_{\bar{1}} = \mathrm{Hom}(\mathbb{C}^m, \mathbb{C}^n) \oplus \mathrm{Hom}(\mathbb{C}^n, \mathbb{C}^m). \tag{5}$$

The elementary matrix $E_{ij}$, which has 1 in the $(i,j)$th entry and 0 elsewhere, has grading $[E_{ij}] = [i] + [j]$.

The Lie superalgebra $\mathfrak{gl}(m|n)$ is the $\mathbb{Z}_2$-graded vector space $\mathrm{End}(\mathbb{C}^{m|n})$, endowed with the graded commutator $[-,-]: \mathrm{End}(\mathbb{C}^{m|n}) \otimes \mathrm{End}(\mathbb{C}^{m|n}) \to \mathrm{End}(\mathbb{C}^{m|n})$: for elements $a$, $b$ with homogeneous $\mathbb{Z}_2$-grading,

$$[a,b] = ab - (-1)^{[a][b]}ba. \tag{6}$$

We will distinguish the elements of $\mathfrak{gl}(m|n)$ from those of $\mathrm{End}(\mathbb{C}^{m|n})$ by writing them as $\mathcal{E}_{ij}$ rather than $E_{ij}$. They satisfy the commutation relations

$$[\mathcal{E}_{ij}, \mathcal{E}_{kl}] = \delta_{jk}\mathcal{E}_{il} - (-1)^{([i]+[j])([k]+[l])}\delta_{li}\mathcal{E}_{kj}. \tag{7}$$

The Cartan subalgebra of $\mathfrak{gl}(m|n)$ is generated by

$$H_r = (-1)^{[r]}\mathcal{E}_{rr} - (-1)^{[r+1]}\mathcal{E}_{r+1,r+1}, \quad r = 1,\ldots,m+n-1, \tag{8}$$

and one more diagonal matrix, say $\mathcal{E}_{11}$. The elementary matrix $\mathcal{E}_{ij}$ has the root $\varepsilon_i - \varepsilon_j$, with

$$\varepsilon_i = \mathcal{E}_{ii}^{\vee} \tag{9}$$

being the weight of $e_i$ in the natural $(m+n)$-dimensional representation. The positive roots are $\varepsilon_i - \varepsilon_j$ with $i < j$. The simple roots are

$$\alpha_r = \varepsilon_r - \varepsilon_{r+1}, \quad r = 1,\ldots,m+n-1. \tag{10}$$

The elements having the roots $\alpha_r$ and $-\alpha_r$ are $E_r = \mathcal{E}_{r,r+1}$ and $F_r = \mathcal{E}_{r+1,r}$, respectively. They satisfy

$$[H_r, E_s] = a_{rs}E_s, \tag{11}$$

$$[H_r, F_s] = -a_{rs}F_s, \tag{12}$$

$$[E_r, F_s] = \delta_{rs}(-1)^{[r]}H_r, \tag{13}$$

where

$$a_{rs} = \alpha_s(H_r) = \delta_{rs}\big((-1)^{[r]} + (-1)^{[r+1]}\big) - \delta_{r+1,s}(-1)^{[r+1]} - \delta_{r,s+1}(-1)^{[r]} \tag{14}$$

is the $(r,s)$th entry of the Cartan matrix.

The structure of $\mathfrak{gl}(m|n)$ can be encoded in a Dynkin diagram, in which a simple root $\alpha_r$ is represented by a blank node if $a_{rr} = \pm 2$ and a crossed node if $a_{rr} = 0$, and two nodes $\alpha_r$, $\alpha_s$ are connected by an edge if $a_{rs} \neq 0$. As an example, consider the case with $(m|n) = (3|2)$ and $(e_1,e_2,e_3,e_4,e_5) = (b_1,b_2,f_1,f_2,b_3)$. The associated Dynkin diagram is

$$\underset{\epsilon_1-\epsilon_2}{\bigcirc}\!\!-\!\!-\!\!-\!\!\underset{\epsilon_2-\delta_1}{\otimes}\!\!-\!\!-\!\!-\!\!\underset{\delta_1-\delta_2}{\bigcirc}\!\!-\!\!-\!\!-\!\!\underset{\delta_2-\epsilon_3}{\otimes}, \tag{15}$$

where $\epsilon_i$ and $\delta_i$ are the weights of $b_i$ and $f_i$, respectively.

Let us give a different presentation of the content of the Dynkin diagram in terms of a quiver diagram, which makes the connection to gauge theory transparent.

First, we represent $\varepsilon_i$ by a vertical line, of one of two colors depending on its grading:

$$\varepsilon_i = \begin{cases} | & ([i] = \bar{0}); \\ | & ([i] = \bar{1}). \end{cases} \tag{16}$$

The ordered set $(\varepsilon_1, \ldots, \varepsilon_{m+n})$ is then represented graphically as $m$ vertical lines of one color and $n$ vertical lines of the other color, placed in the order specified by the choice of the $\mathbb{Z}_2$-grading:

$$\tag{17}$$

Next, we put a circle node between each pair of adjacent vertical lines:

$$\tag{18}$$

The $r$th node represents the simple root $\alpha_r$.

Finally, for each pair $(r,s)$ with $a_{rs} \neq 0$, we draw an arrow from the $r$th node to the $s$th node and write the number $a_{rs}$ on the side. We can erase the vertical lines at this stage:

$$\tag{19}$$

This quiver has the same content as the Dynkin diagram (15) modulo the action of the Weyl group $\mathfrak{S}_m \times \mathfrak{S}_n$ which permutes the basis vectors $(e_1, \ldots, e_{m+n})$ without changing the $\mathbb{Z}_2$-grading.

### 2.1.2 Verma modules of $\mathfrak{gl}(m|n)$

A representation of $\mathfrak{gl}(m|n)$ in a $\mathbb{Z}_2$-graded vector space $V$ is a Lie superalgebra homomorphism $\pi \colon \mathfrak{gl}(m|n) \to \mathrm{End}(V)$, where $\mathrm{End}(V)$ is given the structure of a Lie superalgebra by the graded commutator.

The Verma module $M(\lambda) \colon \mathfrak{gl}(m|n) \to \mathrm{End}(V_\lambda)$, with highest weight

$$\lambda = \sum_{i=1}^{m+n} \lambda_i \varepsilon_i, \quad \lambda_i \in \mathbb{C}, \tag{20}$$

is a representation of $\mathfrak{gl}(m|n)$ constructed from a highest-weight vector $|\Omega_\lambda\rangle$ that is an eigenstate of the diagonal matrices:

$$\mathcal{E}_{ii}|\Omega_\lambda\rangle = \lambda_i |\Omega_\lambda\rangle, \tag{21}$$
$$\mathcal{E}_{ij}|\Omega_\lambda\rangle = 0, \quad i < j. \tag{22}$$

The other vectors in $V_\lambda$ are created by the action of lowering operators $\{\mathcal{E}_{ij} \mid i > j\}$ on $|\Omega_\lambda\rangle$, and two vectors are identified if they are related by the commutation relations (7).

More explicitly, the Fock space $V_\lambda$ can be described as follows. Let us introduce an ordering among all lowering operators and name them $x_1, \ldots, x_p$. Then, by the Poincaré–Birkhoff–Witt (PBW) theorem, an element of $V_\lambda$ is a linear combination of states of the form

$$x_1^{n_1} \cdots x_p^{n_p} |\Omega_\lambda\rangle, \quad n_i \in \begin{cases} \mathbb{Z}_{\geq 0} & ([a_i] = \bar{0}); \\ \{0,1\} & ([a_i] = \bar{1}). \end{cases} \tag{23}$$

Verma modules are infinite-dimensional unless $(m|n) = (1|1)$, in which case there is only one lowering operator and it is odd.

Since the lowering operator $\mathcal{E}_{ij}$ changes the weight by $\varepsilon_i - \varepsilon_j = -\alpha_j - \alpha_{j+1} - \cdots - \alpha_{i-1}$, a state of $M(\lambda)$ has a weight of the form

$$\lambda - \sum_{r=1}^{m+n-1} M_r \alpha_r, \quad M_r \in \mathbb{Z}_{\geq 0}. \tag{24}$$

We can also represent this weight graphically. To represent the highest weight $\lambda$, for each vertical line we draw a diagonal line ending on it and write $\lambda_i$ next to the $i$th diagonal line; and to represent the weight (24), we draw $M_r$ horizontal line segments between the $r$th and $(r+1)$st vertical lines. Here is an example for $(M_1, M_2, M_3, M_4) = (2, 3, 2, 1)$:

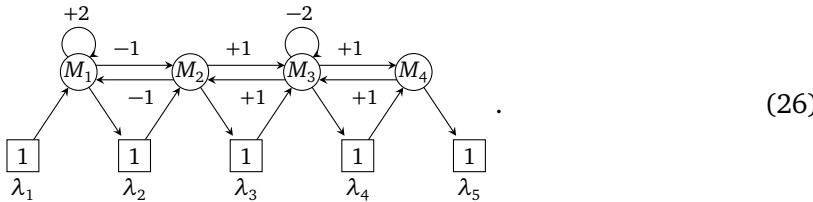

$$\text{(25)}$$

We convert this diagram into a quiver by replacing the diagonal lines with square nodes and writing $M_r$ inside the $r$th circle node and 1 inside the square nodes:

$$\text{(26)}$$

To fix the horizontal positions of the square nodes, we have added arrows connecting circle and square nodes.

### 2.1.3 Tensor products of Verma modules

If $V_1$ and $V_2$ are $\mathbb{Z}_2$-graded vector spaces, the tensor product $V_1 \otimes V_2$ is naturally $\mathbb{Z}_2$-graded. Given two representations $\pi_1 : \mathfrak{gl}(m|n) \to \text{End}(V_1)$ and $\pi_2 : \mathfrak{gl}(m|n) \to \text{End}(V_2)$, the tensor product representation $\pi_1 \otimes \pi_2 : \mathfrak{gl}(m|n) \to \text{End}(V_1 \otimes V_2)$ is defined by

$$(\pi_1 \otimes \pi_2)(x)(v_1 \otimes v_2) = \pi_1(x)v_1 \otimes v_2 + (-1)^{[x][v_1]}v_1 \otimes \pi_2(x)v_2, \tag{27}$$

where $v_1$, $v_2$ and $x$ are homogeneous in $\mathbb{Z}_2$-grading. The tensor products of more than two representations can be defined recursively.

The tensor product of $L$ Verma modules $M(\lambda^1), \ldots, M(\lambda^L)$ has a highest-weight vector $|\Omega_{\lambda^1}\rangle \otimes \cdots \otimes |\Omega_{\lambda^L}\rangle$ with highest weight $\lambda^1 + \cdots + \lambda^L$. A weight of $M(\lambda^1) \otimes \cdots \otimes M(\lambda^L)$ takes the form

$$\sum_{\ell=1}^{L} \lambda^\ell - \sum_{r=1}^{m+n-1} M_r \alpha_r, \quad M_r \in \mathbb{Z}_{\geq 0}. \tag{28}$$

Graphically, we represented it by a diagram similar to the diagram (25) for a weight of a single Verma module, but with $L$ diagonal lines ending on each vertical line. For example, the diagram

$$\text{(29)}$$

represents a weight with $(M_1, M_2, M_3, M_4) = (2, 1, 0, 2)$ in the representation $M(\lambda^1) \otimes M(\lambda^2) \otimes M(\lambda^3)$ of $\mathfrak{gl}(3|2)$.

The corresponding quiver diagram is the same as before, except that the $i$th square node is now labeled $L$ and accompanied by the $L$-tuple $\vec{\lambda}_i = (\lambda_i^1, \ldots, \lambda_i^L)$:

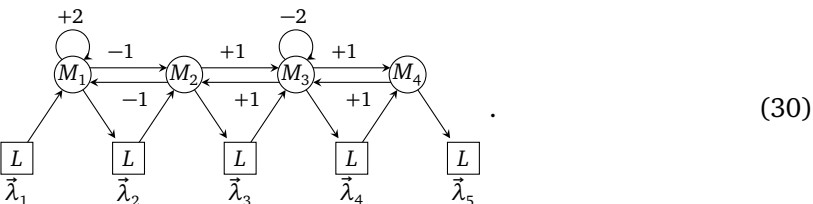

$$(30)$$

The above quiver diagram will be identified with a quiver describing a two-dimensional $\mathcal{N} = (2,2)$ supersymmetric gauge theory that appears on the gauge theory side of the Bethe/gauge correspondence. The graphical representation using lines will be interpreted as a diagram depicting a brane configuration in string theory.

## 2.2 Bethe side

Now we explain the Bethe side of the Bethe/gauge correspondence. The spin chains we consider in this paper are rational $\mathfrak{gl}(m|n)$ spin chains, for which spins take values in representations of $\mathfrak{gl}(m|n)$. More generally, spins in rational $\mathfrak{gl}(m|n)$ spin chains are valued in representations of the Yangian $Y(\mathfrak{gl}(m|n))$ of $\mathfrak{gl}(m|n)$.

### 2.2.1 Yangian

The *Yangian* $Y(\mathfrak{gl}(m|n))$ is a $\mathbb{Z}_2$-graded Hopf algebra, which in particular is a unital associative $\mathbb{Z}_2$-graded algebra. It is generated by elements

$$T_{ij}^{(l)}, \quad i,j = 1,\ldots,m+n, \quad l \in \mathbb{Z}_{>0}, \tag{31}$$

with grading $[T_{ij}^{(l)}] = [i] + [j]$. The level-1 generators $T_{ij}^{(1)}$ span a subalgebra isomorphic to $\mathfrak{gl}(m|n)$, with the identification of generators being $\mathcal{E}_{ij} = (-1)^{[j]} T_{ji}^{(1)}$.

To describe the algebra relations for all generators in a compact form, let us introduce a formal variable $\sigma$ and combine the generators into a single $\mathrm{End}(\mathbb{C}^{m|n}) \otimes Y(\mathfrak{gl}(m|n))$-valued power series in $\sigma^{-1}$:

$$T(\sigma) = \sum_{i,j=1}^{m+n} E_{ij} \otimes T_{ij}(\sigma) = \sum_{i,j=1}^{m+n} \sum_{l=0}^{\infty} \frac{\hbar^l}{\sigma^l} E_{ij} \otimes T_{ij}^{(l)}. \tag{32}$$

Here $T_{ij}^{(0)} = \delta_{ij}$ and $\hbar$ is a complex parameter. We can think of $T(\sigma)$ as an $(m+n) \times (m+n)$ matrix whose entries are elements of $Y(\mathfrak{gl}(m|n))[[\sigma^{-1}]]$; it is called the *monodromy matrix* and is a function of *spectral parameter* $\sigma$. In terms of the monodromy matrix, the algebra relations for $Y(\mathfrak{gl}(m|n))$ are encoded in the *RTT relation*

$$R_{12}(\sigma_1 - \sigma_2) T_1(\sigma_1) T_2(\sigma_2) = T_2(\sigma_2) T_1(\sigma_1) R_{12}(\sigma_1 - \sigma_2). \tag{33}$$

This is a relation between elements in $\mathrm{End}(\mathbb{C}^{m|n}) \otimes \mathrm{End}(\mathbb{C}^{m|n}) \otimes Y(\mathfrak{gl}(m|n))[[\sigma^{-1}]]$, and the subscript(s) on an operator indicate which factor(s) of $\mathbb{C}^{m|n}$ the operator acts on.

The operator $R_{12}(\sigma) \in \mathrm{End}(\mathbb{C}^{m|n}) \otimes \mathrm{End}(\mathbb{C}^{m|n})$ that appears in the RTT relation is the *rational $\mathfrak{gl}(m|n)$ R-matrix*. It is given by

$$R_{12}(\sigma) = \sigma I \otimes I + \hbar P_{12}, \tag{34}$$

where $I$ is the identity matrix and

$$P_{12} = \sum_{i,j=1}^{m+n} (-1)^{[j]} E_{ij} \otimes E_{ji} \,. \tag{35}$$

The permutation operator $P_{12}$ swaps tensor factors as

$$P_{12}(e_i \otimes e_j) = (-1)^{[i][j]} e_j \otimes e_i \,. \tag{36}$$

The R-matrix commutes with the automorphism group $\mathrm{GL}(m|n)$ of $\mathbb{C}^{m|n}$:

$$[g \otimes g, R_{12}(\sigma)] = 0 \,, \quad g \in \mathrm{GL}(m|n) \,. \tag{37}$$

The dynamics of a closed rational $\mathfrak{gl}(m|n)$ spin chain is generated by the *transfer matrix*

$$t(g,\sigma) = \mathrm{str}_{\mathbb{C}^{m|n}} \big( g\, T(\sigma) \big) = \sum_{i=1}^{m+n} (-1)^{[i]} g_{ij} T_{ji}(\sigma) \,, \quad g \in \mathrm{GL}(m|n) \,. \tag{38}$$

The supertrace taken over $\mathbb{C}^{m|n}$ corresponds to the topology of the spin chain which is closed, and $g$ twists the periodic boundary condition.

Multiplying both sides of the RTT relation (33) by $g \otimes g \otimes 1$ from the left and $R_{12}(\sigma_1 - \sigma_2)^{-1}$ from the right, then using the symmetry (37) of the R-matrix and taking the supertrace over $\mathbb{C}^{m|n} \otimes \mathbb{C}^{m|n}$, we see that transfer matrices for a fixed $g$ commute with each other:

$$[t(g,\sigma_1), t(g,\sigma_2)] = 0 \,. \tag{39}$$

Therefore, if we expand $t(g,\sigma)$ in powers of $\sigma^{-1}$, the coefficients are mutually commuting elements of $Y(\mathfrak{gl}(m|n))$. They generate a commutative subalgebra called the *Bethe algebra* (or the *Baxter algebra*) of $Y(\mathfrak{gl}(m|n))$.

### 2.2.2 Representations of $Y(\mathfrak{gl}(m|n))$

While the Yangian $Y(\mathfrak{gl}(m|n))$ and its Bethe algebra are the algebraic structures underlying rational $\mathfrak{gl}(m|n)$ spin chains, to get a concrete physical realization of a spin chain we need to specify a representation of $Y(\mathfrak{gl}(m|n))$.

A representation $\rho \colon Y(\mathfrak{gl}(m|n)) \to \mathrm{End}(V)$ of $Y(\mathfrak{gl}(m|n))$ maps the monodromy matrix to an $\mathrm{End}(V)$-valued matrix

$$\rho\big(T(\sigma)\big) = \sum_{i,j=1}^{m+n} \sum_{l=0}^{\infty} \frac{\hbar^l}{\sigma^l} E_{ij} \otimes \rho(T_{ij}^{(l)}) \,. \tag{40}$$

Conversely, an $\mathrm{End}(V)$-valued matrix satisfying the RTT relation determines a representation of $Y(\mathfrak{gl}(m|n))$.

Given a representation $\pi \colon \mathfrak{gl}(m|n) \to \mathrm{End}(V)$ of $\mathfrak{gl}(m|n)$, we obtain a one-parameter family of representations $\pi_\zeta \colon Y(\mathfrak{gl}(m|n)) \to \mathrm{End}(V)$, $\zeta \in \mathbb{C}$, of $Y(\mathfrak{gl}(m|n))$ by

$$\pi_\zeta\big(T(\sigma)\big) = I \otimes \mathrm{id}_V + \frac{\hbar}{\sigma - \zeta} \sum_{i,j=1}^{m+n} (-1)^{[j]} E_{ij} \otimes \pi(\mathcal{E}_{ji}) \,. \tag{41}$$

This is known as the *evaluation module* for $\pi$, and $\zeta$ is called the *inhomogeneity parameter*. Note that we have

$$\pi_\zeta\big((-1)^{[j]} T_{ji}^{(1)}\big) = \pi(\mathcal{E}_{ij}) \,. \tag{42}$$

For a representation $\pi\colon \mathfrak{gl}(m|n) \to \mathrm{End}(V)$ of $\mathfrak{gl}(m|n)$, let us define a one-parameter family of representations $\pi^c\colon \mathfrak{gl}(m|n) \to \mathrm{End}(V)$, $c \in \mathbb{C}$, by

$$\pi^c(\mathcal{E}_{ij}) = \pi(\mathcal{E}_{ij}) + (-1)^{[i]} c \delta_{ij} \, \mathrm{id}_V \ . \tag{43}$$

In the associated Yangian representations, the parameter $c$ is related to a shift in the inhomogeneity parameter. Suppose that $\rho\colon Y(\mathfrak{gl}(m|n)) \to \mathrm{End}(V)$ is a representation of $Y(\mathfrak{gl}(m|n))$. Then, $\rho(T(\sigma))$ satisfies the RTT equation, and for any function $f$ of $\sigma$, the RTT equation is still satisfied when $\rho(T(\sigma))$ is replaced by $f(\sigma)\rho(T(\sigma))$. Therefore, if $f(\sigma)$ can be expanded in a power series in $\sigma^{-1}$ starting from 1, then $f(\sigma)\rho(T(\sigma))$ defines a new representation of $Y(\mathfrak{gl}(m|n))$. Since

$$\pi^c_\zeta\big(T(\sigma)\big) = \pi_\zeta(T(\sigma)) + \frac{c\hbar}{\sigma - \zeta} I \otimes \mathrm{id}_V = \frac{\sigma - \zeta + c\hbar}{\sigma - \zeta} \pi_{\zeta - c\hbar}\big(T(\sigma)\big), \tag{44}$$

we see that $\pi^c_\zeta$ and $\pi_{\zeta - c\hbar}$ are related in this manner.

To construct tensor product representations of $Y(\mathfrak{gl}(m|n))$, we use the coproduct $\Delta\colon Y(\mathfrak{gl}(m|n)) \to Y(\mathfrak{gl}(m|n)) \otimes Y(\mathfrak{gl}(m|n))$. The map $\Delta$ is defined by the formula

$$\Delta\big(T(\sigma)\big) = \sum_{i,j,k=1}^{m+n} (-1)^{([i]+[k])([k]+[j])} E_{ij} \otimes T_{ik}(\sigma) \otimes T_{kj}(\sigma). \tag{45}$$

Given two representations $\rho_1\colon Y(\mathfrak{gl}(m|n)) \to \mathrm{End}(V_1)$ and $\rho_2\colon Y(\mathfrak{gl}(m|n)) \to \mathrm{End}(V_2)$, the tensor product representation $\rho_1 \dot\otimes \rho_2\colon Y(\mathfrak{gl}(m|n)) \to \mathrm{End}(V_1 \otimes V_2)$ is defined by

$$\rho_1 \dot\otimes \rho_2 = (\rho_1 \otimes \rho_2) \circ \Delta . \tag{46}$$

A calculation shows that $(\rho_1 \dot\otimes \rho_2)(T(\sigma))$ satisfies the RTT relation.

### 2.2.3 The spin chain

Now, fix a positive integer $L$, and choose $L$ highest weights

$$\vec{\lambda} = (\lambda^1, \ldots, \lambda^L) \tag{47}$$

of $\mathfrak{gl}(m|n)$ and $L$ inhomogeneity parameters

$$\vec{\zeta} = (\zeta^1, \ldots, \zeta^L). \tag{48}$$

Also, choose a diagonal element[1] of $\mathrm{GL}(m|n)$:

$$g = \mathrm{diag}(e^{\phi_1}, \ldots, e^{\phi_{m+n}}), \quad \phi_i \in \mathbb{C} . \tag{49}$$

We consider the closed rational $\mathfrak{gl}(m|n)$ spin chain of length $L$, with the spin at the $\ell$th site valued in the evaluation module $M(\lambda^\ell)_{\zeta^\ell}$ for the Verma module $M(\lambda^\ell)$ and the periodic boundary condition twisted by $g$.

The Hilbert space of states of this spin chain is the tensor product

$$V_{\vec{\lambda}} = \bigotimes_{\ell=1}^{L} V_{\lambda^\ell}, \tag{50}$$

---

[1]The spin chain can be defined for any choice of $g$, not necessarily diagonal ones. However, nondiagonalizable choices of $g$ do not appear to have a clear interpretation on the gauge theory side of the Bethe/gauge correspondence.

and the $L$ spins can be thought of as a single spin in the tensor product representation

$$M(\vec{\lambda})_{\vec{\zeta}} = M(\lambda^1)_{\zeta^1} \dot{\otimes} \cdots \dot{\otimes} M(\lambda^L)_{\zeta^L}. \tag{51}$$

The transfer matrix of the spin chain

$$M(\vec{\lambda})_{\vec{\zeta}}\big(t(g,\sigma)\big) \tag{52}$$

generates commuting conserved charges acting on $V_{\vec{\lambda}}$, making the spin chain integrable. The Hamiltonian is a linear combination of these charges.

### 2.2.4   Bethe equations

The Hilbert space of the spin chain is spanned by vectors that simultaneously diagonalize the commuting conserved charges, or equivalently, diagonalize the transfer matrix (52) for all values of the spectral parameter $\sigma$. These eigenvectors, referred to as *Bethe vectors*, are the main characters from the Bethe side of the Bethe/gauge correspondence.

The Bethe vectors for the rational $\mathfrak{gl}(m|n)$ spin chain have been constructed by Bethe ansatz methods [29, 30]. The construction starts with the highest-weight vector $|\Omega_{\vec{\lambda}}\rangle$ of the tensor product representation

$$M(\vec{\lambda}) = \bigotimes_{\ell=1}^{L} M(\lambda^{\ell}). \tag{53}$$

This state is called the *pseudovacuum* and satisfies

$$M(\vec{\lambda})_{\vec{\zeta}}\big(T_{ii}(\sigma)\big)|\Omega_{\vec{\lambda}}\rangle = \left(\prod_{\ell=1}^{L} \frac{\sigma - \zeta^{\ell} + (-1)^{[i]}\lambda_i^{\ell}\hbar}{\sigma - \zeta^{\ell}}\right)|\Omega_{\vec{\lambda}}\rangle, \tag{54}$$

$$M(\vec{\lambda})_{\vec{\zeta}}\big(T_{ij}(\sigma)\big)|\Omega_{\vec{\lambda}}\rangle = 0, \quad i > j. \tag{55}$$

According to our graphical notation, the pseudovacuum is represented by a diagram with no horizontal segments:

$$|\Omega_{\vec{\lambda}}\rangle = \begin{smallmatrix}\lambda_1^1\\\lambda_1^2\\\lambda_1^3\end{smallmatrix}\!\!\!\diagup\!\!\! \quad \begin{smallmatrix}\lambda_2^1\\\lambda_2^2\\\lambda_2^3\end{smallmatrix}\!\!\!\diagup\!\!\! \quad \begin{smallmatrix}\lambda_3^1\\\lambda_3^2\\\lambda_3^3\end{smallmatrix}\!\!\!\diagup\!\!\! \quad \begin{smallmatrix}\lambda_4^1\\\lambda_4^2\\\lambda_4^3\end{smallmatrix}\!\!\!\diagup\!\!\! \quad \begin{smallmatrix}\lambda_5^1\\\lambda_5^2\\\lambda_5^3\end{smallmatrix}\!\!\!\diagup\!\!\! . \tag{56}$$

Excited states are obtained from the pseudovacuum by the action of creation operators $M(\vec{\lambda})_{\vec{\zeta}}(T_{ij}(\sigma))$, $i < j$. The operator $M(\vec{\lambda})_{\vec{\zeta}}(T_{ij}(\sigma))$ contains $M(\vec{\lambda})(\mathcal{E}_{ji})$ and changes the $\mathfrak{gl}(m|n)$ weight by $\varepsilon_j - \varepsilon_i = -\alpha_i - \alpha_{i+1} - \cdots - \alpha_{j-1}$. Roughly speaking, we can interpret this action as creating a single quasi-particle, or a *magnon*, of rapidity $\sigma$ and type $r$ for each $r = i, i+1, \ldots, j-1$. Graphically, we think of it as creating a horizontal line connecting the $i$th and $j$th vertical lines:

$$T_{13}(\sigma_1)T_{25}(\sigma_2)T_{45}(\sigma_3)|\Omega_{\vec{\lambda}}\rangle \sim \quad . \tag{57}$$

This is, however, not a precise correspondence because the left-hand side depends on the ordering of creation operators.

The operator $T_{ji}(\sigma)$, $i < j$, changes the weight by $\alpha_i + \alpha_{i+1} + \cdots + \alpha_{j-1}$, so annihilates magnons of type $m = i, i+1, \ldots, j-1$. It removes one horizontal line from each of the

intervals between the $i$th and $j$th vertical lines. If there is no horizontal line to remove, then the state is annihilated.

Eigenvectors of the transfer matrix are excited states constructed by certain linear combinations of creation operators. It turns out that a Bethe vector with the $\mathfrak{gl}(m|n)$ weight (28) is specified by a *Bethe root*

$$(\{\sigma_1^1,\ldots,\sigma_1^{M_1}\},\{\sigma_2^1,\ldots,\sigma_2^{M_2}\},\ldots,\{\sigma_{m+n-1}^1,\ldots,\sigma_{m+n-1}^{M_{m+n-1}}\}), \tag{58}$$

which is a solution of the *Bethe equations*

$$
e^{\tau_r}\prod_{s=1}^{m+n-1}\prod_{b_s=1}^{M_s}\frac{\sigma_r^{a_r}-\sigma_s^{b_s}+\frac{1}{2}a_{rs}\hbar}{\sigma_r^{a_r}-\sigma_s^{b_s}-\frac{1}{2}a_{rs}\hbar}
$$
$$
=(-1)^{\delta_{[r],[r+1]}}\prod_{\ell=1}^{L}\frac{\sigma_r^{a_r}-\zeta^\ell+(-1)^{[r]}\lambda_r^\ell\hbar-\frac{1}{2}c_r\hbar}{\sigma_r^{a_r}-\zeta^\ell+(-1)^{[r+1]}\lambda_{r+1}^\ell\hbar-\frac{1}{2}c_r\hbar},
$$
$$
a_r=1,\ldots,M_r,\quad r=1,\ldots,m+n-1. \tag{59}
$$

Here we have defined

$$\tau_r=(-1)^{[r+1]}\phi_{r+1}-(-1)^{[r]}\phi_r \tag{60}$$

and

$$c_i=\sum_{j=1}^{i}(-1)^{[j]}. \tag{61}$$

We represent this Bethe vector by a diagram with $M_r$ horizontal lines between the $r$th and $(r+1)$st vertical lines, with labels $\{\sigma_r^1,\ldots,\sigma_r^{M_r}\}$. Therefore, the diagram

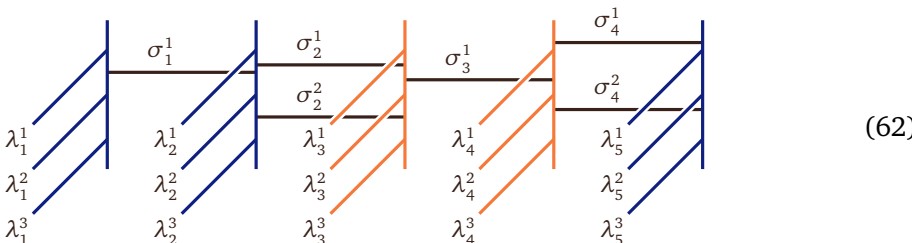

$$\tag{62}$$

represents a Bethe vector of the closed $\mathfrak{gl}(3|2)$ rational spin chain of length $L=3$ that belongs to the magnon sector $(M_1,M_2,M_3,M_4)=(1,2,1,2)$ and corresponds to the Bethe root $(\{\sigma_1^1\},\{\sigma_2^1,\sigma_2^2\},\{\sigma_3^1\},\{\sigma_4^1,\sigma_4^2\})$.

Note that the Bethe equations are invariant under the shift

$$\zeta^\ell\mapsto\zeta^\ell+c^\ell\hbar,\quad \lambda_i^\ell\mapsto\lambda_i^\ell+(-1)^{[i]}c^\ell,\quad c^\ell\in\mathbb{C}. \tag{63}$$

This is a consequence of the relation (44) between a representation with shifted highest weight and a representation with shifted inhomogeneity parameter. Since multiplying the transfer matrix by a function of the spectral parameter does not change its eigenvectors, the Bethe equations remain the same if we shift the highest weights and inhomogeneity parameters as above.

## 2.3  Gauge side

Now we turn to the gauge side of the Bethe/gauge correspondence. The closed rational $\mathfrak{gl}(m|n)$ spin chain discussed above corresponds to a family of two-dimensional $\mathcal{N}=(2,2)$ supersymmetric gauge theories whose field contents are described by quivers. These theories have supersymmetric vacua that are in one-to-one correspondence with the Bethe vectors of the spin chain. For background knowledge on $\mathcal{N}=(2,2)$ supersymmetric gauge theories, we refer the reader to [31].

### 2.3.1 The gauge theories

The magnon sectors of the spin chain (the weight spaces of the Hilbert space (50)) are labeled by $(m + n - 1)$-tuples of nonnegative integers. The sector with magnon numbers $(M_1, \ldots, M_{m+n-1})$ corresponds to a theory with the product gauge group

$$U(M_1) \times \cdots \times U(M_{m+n-1}).\tag{64}$$

Correspondingly, the theory has vector multiplets $V_r$, $r = 1, \ldots, m+n-1$, one for each unitary gauge group factor.

In addition, the theory has various chiral multiplets. If $[r] = [r + 1]$, then there is one chiral multiplet transforming in the adjoint representation of $U(M_r)$:[2]

$$\phi_r \in \mathrm{Hom}(\mathbb{C}^{M_r}, \mathbb{C}^{M_r}), \quad [r] = [r + 1], \quad r = 1, \ldots, m + n - 1.\tag{65}$$

There are also chiral multiplets

$$P_i \in \mathrm{Hom}(\mathbb{C}^{M_{i-1}}, \mathbb{C}^{M_i}), \quad i = 2, \ldots, m + n - 1,\tag{66}$$

$$\widetilde{P}_i \in \mathrm{Hom}(\mathbb{C}^{M_i}, \mathbb{C}^{M_{i-1}}), \quad i = 2, \ldots, m + n - 1,\tag{67}$$

$$Q_i \in \mathrm{Hom}(\mathbb{C}^{L}, \mathbb{C}^{M_{i-1}}), \quad i = 2, \ldots, m + n,\tag{68}$$

$$\widetilde{Q}_i \in \mathrm{Hom}(\mathbb{C}^{M_i}, \mathbb{C}^{L}), \quad i = 1, \ldots, m + n - 1.\tag{69}$$

It is convenient to introduce the notations

$$\phi_r = 0, \quad [r] \neq [r + 1],\tag{70}$$

and

$$P_1 = \widetilde{P}_1 = P_{m+n} = \widetilde{P}_{m+n} = Q_1 = \widetilde{Q}_{m+n} = 0.\tag{71}$$

These chiral multiplets are coupled by the superpotential

$$W = \sum_{r=1}^{m+n-1} \mathrm{tr}_{\mathbb{C}^{M_r}}\Big(\phi_r \widetilde{P}_{r+1} P_{r+1} - \phi_r P_r \widetilde{P}_r + P_r Q_r \widetilde{Q}_r$$
$$+ \big((-1)^{[r]} - (-1)^{[r+1]}\big) \widetilde{P}_{r+1} P_{r+1} P_r \widetilde{P}_r\Big).\tag{72}$$

The terms involving adjoint chiral multiplets are the cubic superpotentials required for $\mathcal{N} = (4, 4)$ supersymmetry, which the theory possesses if either $m = 0$ or $n = 0$. The last quartic terms are present only for the gauge group factors without adjoint chiral multiplets [32].[3]

The field content of the theory can be encoded in a quiver diagram. This is the same quiver as the one that specifies a weight in the tensor product of $L$ Verma modules of $\mathfrak{gl}(m|n)$. Here is the quiver for the now-familiar $\mathfrak{gl}(3|2)$ example, with the arrows labeled with the corresponding chiral multiplets:

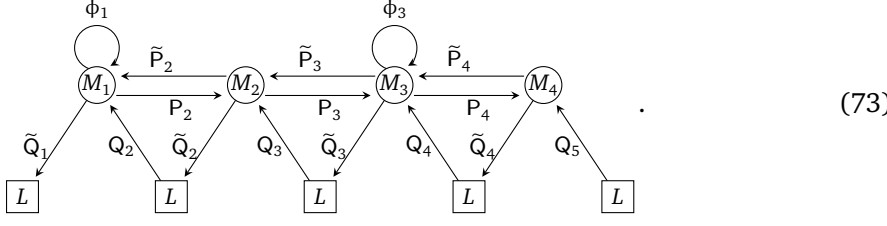

$$\tag{73}$$

---

[2]Here and thereafter, statements about fields such as the one that follows only indicate the representations in which they are valued.

[3]The quartic terms can be understood as follows. Suppose that $[r] \neq [r + 1]$, say $[r] = 0$ and $[r + 1] = 1$, and introduce a pair of chiral multiplets $\phi_r^{\pm}$ in the adjoint representation of $U(M_r)$. These multiplets are massive and couple with bifundamental chiral multiplets via superpotential terms of the form $\mathrm{tr}_{\mathbb{C}^{M_r}}(\phi_r^- \widetilde{P}_{r+1} P_{r+1} - \phi_r^+ P_r \widetilde{P}_r + m \phi_r^+ \phi_r^-)$. Integrating out $\phi_r^{\pm}$, we get the quartic term. From the point of view of the brane construction discussed in section 3.1, we imagine the situation in which $NS5_r$ and $NS5_{r+1}$ are almost orthogonal but not quite. The adjoint chiral multiplets $\phi_r^{\pm}$ correspond to the positions of D2-branes along $\mathbb{R}^2_{\pm\hbar}$.

A circle node labeled $M$ is a U($M$) gauge group. The theory has $m + n$ copies of U($L$) flavor groups, denoted here by square nodes. We name them U($L$)$_1$, U($L$)$_2$, ..., U($L$)$_{m+n}$ from left to right. The terms in the superpotential (72) correspond to closed paths of length three and four in the quiver.

Apart from the U($L$)$^{m+n}$ flavor symmetry, the theory has an important U(1) global symmetry preserved by the superpotential. We call it U(1)$_\hbar$. The charges of the chiral multiplets under U(1)$_\hbar$ are

$$\phi_r : 2(-1)^{[r]}, \tag{74}$$

$$\mathsf{P}_i : -(-1)^{[i]}, \tag{75}$$

$$\widetilde{\mathsf{P}}_i : -(-1)^{[i]}, \tag{76}$$

$$\mathsf{Q}_i : \frac{1}{2}(-1)^{[i]}, \tag{77}$$

$$\widetilde{\mathsf{Q}}_i : \frac{1}{2}(-1)^{[i]}. \tag{78}$$

For $\phi_r$ and $\mathsf{P}_i$, $\widetilde{\mathsf{P}}_i$, their charges coincide with the corresponding Cartan matrix elements.

Most parameters of the spin chain correspond in the gauge theory to the twisted masses with respect to the global symmetry U($L$)$^{m+n} \times$ U(1)$_\hbar$. We can turn on the twisted masses as follows. First, we couple vector multiplets for U($L$)$^{m+n} \times$ U(1)$_\hbar$ to the chiral multiplets and gauge the global symmetry. Then, we give vacuum expectation values to the adjoint scalar fields in these vector multiplets. Finally, we take the limit in which the gauge couplings for U($L$)$^{m+n} \times$ U(1)$_\hbar$ go to zero, thereby freezing the vector multiplets just added. The vacuum expectation values of the scalar fields appear as complex mass parameters for the chiral multiplets, which are the twisted masses in question.

To the scalar field for U($L$)$_i$, we give the vacuum expectation value

$$\mathrm{diag}(\mu_i^1, \ldots, \mu_i^L). \tag{79}$$

This yields twisted masses $-\mu_i^\ell$ to $\mathsf{Q}_i$ and $+\mu_i^\ell$ to $\widetilde{\mathsf{Q}}_i$. To the scalar field for U(1)$_\hbar$, we give the vacuum expectation value $\hbar/2$. This yields a twisted mass $q\hbar/2$ to a chiral multiplet that has charge $q$ under U(1)$_\hbar$. To summarize, the twisted masses of the chiral multiplets are

$$(\phi_r)^{a_r}{}_{b_r} : (-1)^{[r]}\hbar, \tag{80}$$

$$(\mathsf{P}_i)^{a_i}{}_{b_{i-1}} : -\frac{1}{2}(-1)^{[i]}\hbar, \tag{81}$$

$$(\widetilde{\mathsf{P}}_i)^{a_{i-1}}{}_{b_i} : -\frac{1}{2}(-1)^{[i]}\hbar, \tag{82}$$

$$(\mathsf{Q}_i)^{a_{i-1}}{}_\ell : -\mu_i^\ell + \frac{1}{4}(-1)^{[i]}\hbar, \tag{83}$$

$$(\widetilde{\mathsf{Q}}_i)^\ell{}_{a_i} : +\mu_i^\ell + \frac{1}{4}(-1)^{[i]}\hbar. \tag{84}$$

Lastly, we need FI parameters and theta angles in order to account for the twist parameters for the periodic boundary condition of the spin chain. To do so, from each vector multiplet $V_r$ we construct an adjoint twisted chiral multiplet $\Sigma_r$ whose lowest component is the vector multiplet scalar $\sigma_r$. Then, we choose complexified FI parameters

$$t_1, \ldots, t_{m+n-1} \in \mathbb{C}/2\pi i\mathbb{Z}, \tag{85}$$

and turn on the twisted superpotential

$$\widetilde{W} = -\sum_{r=1}^{m+n-1} t_r \,\mathrm{tr}\,\Sigma_r. \tag{86}$$

The real and imaginary parts of $t_r$ are related to the FI parameter $r_r$ and the theta angle $\theta_r$ for $U(M_r)$ as $t_r = r_r - i\theta_r$.

### 2.3.2 Vacuum equations

We are interested in the vacua of this theory when it is defined on a periodic space. For $\mathcal{N} = (2,2)$ supersymmetric gauge theories, the exact low-energy effective descriptions are known and we can use these descriptions to determine their vacua. See [31] for detailed discussions in the abelian case. Nonabelian examples are treated in [33].

Let $\Sigma_r^{a_r}$, $a_r = 1, \ldots, M_r$, be the diagonal components of $\Sigma_r$, and let $\Sigma$ and $\sigma$ collectively denote $\{\Sigma_r^{a_r}\}$ and their scalar components $\{\sigma_r^{a_r}\}$, respectively.

If the fields $\sigma$ take generic large values and are slowly varying, the chiral multiplets and the off-diagonal components of the vector multiplets (or equivalently the corresponding twisted chiral multiplets) can be considered as having large masses due to higgsing. Their masses are

$$(\Sigma_r)^{a_r}{}_{b_r} : \sigma_r^{a_r} - \sigma_r^{b_r} \,, \tag{87}$$

$$(\phi_r)^{a_r}{}_{b_r} : \sigma_r^{a_r} - \sigma_r^{b_r} + (-1)^{[r]}\hbar \,, \tag{88}$$

$$(P_i)^{a_i}{}_{b_{i-1}} : \sigma_i^{a_i} - \sigma_{i-1}^{b_{i-1}} - \frac{1}{2}(-1)^{[i]}\hbar \,, \tag{89}$$

$$(\widetilde{P}_i)^{a_{i-1}}{}_{b_i} : \sigma_{i-1}^{a_{i-1}} - \sigma_i^{b_i} - \frac{1}{2}(-1)^{[i]}\hbar \,, \tag{90}$$

$$(Q_i)^{a_{i-1}}{}_\ell : \sigma_{i-1}^{a_{i-1}} - \mu_i^\ell + \frac{1}{4}(-1)^{[i]}\hbar \,, \tag{91}$$

$$(\widetilde{Q}_i)^\ell{}_{a_i} : \mu_i^\ell - \sigma_i^{a_i} + \frac{1}{4}(-1)^{[i]}\hbar \,, \tag{92}$$

where $a_r$, $b_r$ are indices for the $U(M_r)$ gauge group factor. After integrating out these heavy fields, we are left with an effective description of the theory that involves only $\Sigma$.

The effective theory is determined solely by a single holomorphic function of $\sigma$, the *effective twisted superpotential* $\widetilde{W}_{\text{eff}}$, and it can be calculated exactly at one-loop order. Integrating out a chiral multiplet whose mass due to the higgsing is $m(\sigma)$ contributes to $\widetilde{W}_{\text{eff}}(\sigma)$ by the term

$$-m(\sigma)\big(\log m(\sigma) - 1\big). \tag{93}$$

An off-diagonal component of a vector multiplet also contributes in the same way [34]. Integrating out high-energy modes of $\Sigma$ does not alter the form of $\widetilde{W}_{\text{eff}}$ [31].

The vacua of the theory are the solutions of the vacuum equations

$$\exp\left(\frac{\partial \widetilde{W}_{\text{eff}}(\sigma)}{\partial \sigma_r^{a_r}}\right) = 1\,, \quad a_r = 1, \ldots, M_r\,, \quad r = 1, \ldots, m+n-1\,. \tag{94}$$

These equations are invariant under shifts of the exponent by integer multiples of $2\pi i$, reflecting the fact that the imaginary part of the exponent is the effective theta angle. In the case at hand, the vacuum equations read

$$e^{t_r}(-1)^{M_r+1}\left(\prod_{b_r=1}^{M_r} \frac{\sigma_r^{a_r} - \sigma_r^{b_r} + (-1)^{[r]}\hbar}{\sigma_r^{b_r} - \sigma_r^{a_r} + (-1)^{[r]}\hbar}\right)^{\delta_{[r],[r+1]}} \prod_{\ell=1}^{L} \frac{\sigma_r^{a_r} - \mu_{r+1}^\ell + \frac{1}{4}(-1)^{[r+1]}\hbar}{\mu_r^\ell - \sigma_r^{a_r} + \frac{1}{4}(-1)^{[r]}\hbar}$$

$$\times \prod_{b_{r-1}=1}^{M_{r-1}} \frac{\sigma_r^{a_r} - \sigma_{r-1}^{b_{r-1}} - \frac{1}{2}(-1)^{[r]}\hbar}{\sigma_{r-1}^{b_{r-1}} - \sigma_r^{a_r} - \frac{1}{2}(-1)^{[r]}\hbar} \prod_{b_{r+1}=1}^{M_{r+1}} \frac{\sigma_r^{a_r} - \sigma_{r+1}^{b_{r+1}} - \frac{1}{2}(-1)^{[r+1]}\hbar}{\sigma_{r+1}^{b_{r+1}} - \sigma_r^{a_r} - \frac{1}{2}(-1)^{[r+1]}\hbar} = 1\,,$$

$$a_r = 1, \ldots, M_r\,, \quad r = 1, \ldots, m+n-1\,, \tag{95}$$

with $M_0 = M_{m+n} = 0$. The factor $e^{t_r}$ comes from the tree-level twisted superpotential (86), and the factor $(-1)^{M_r+1}$ comes from the off-diagonal components of the vector multiplets. Using the Cartan matrix (14), we can rewrite these equations as

$$e^{\tau_r} \prod_{s=1}^{m+n-1} \prod_{b_s=1}^{M_s} \frac{\sigma_r^{a_r} - \sigma_s^{b_s} + \frac{1}{2}a_{rs}\hbar}{\sigma_r^{a_r} - \sigma_s^{b_s} - \frac{1}{2}a_{rs}\hbar} \prod_{\ell=1}^{L} \frac{\sigma_r^{a_r} - \mu_{r+1}^\ell + \frac{1}{4}(-1)^{[r+1]}\hbar}{\sigma_r^{a_r} - \mu_r^\ell - \frac{1}{4}(-1)^{[r]}\hbar} = (-1)^{\delta_{[r],[r+1]}}, \qquad (96)$$

where

$$\tau_r = t_r + i\pi\big((1 - \delta_{[r],[r+1]})(M_r + 1) + M_{r-1} + M_{r+1} + L\big). \qquad (97)$$

### 2.3.3 Twisted chiral ring

The above two-dimensional gauge theory has $\mathcal{N} = (2,2)$ supersymmetry, generated by four supercharges $Q_\pm$, $\overline{Q}_\pm$. Under a vector U(1) R-symmetry, $\overline{Q}_\pm$ and $Q_\pm$ have charges $+1$ and $-1$, while under an axial U(1) R-symmetry, $\overline{Q}_+$, $Q_-$ have charge $+1$ and $\overline{Q}_-$, $Q_+$ have charge $-1$. The theory is unitary and the supercharges satisfy the reality conditions $Q_\pm^* = \overline{Q}_\pm$. It turns out that the theory has unbroken vector U(1) R-symmetries, and this fact implies the absence of certain central charges $Z$, $Z^*$ in the $\mathcal{N} = (2,2)$ supersymmetry algebra.

The linear combination $Q = \overline{Q}_+ + Q_-$ satisfies

$$Q^2 = \widetilde{Z}, \qquad (98)$$

with $\widetilde{Z}$ being another central charge. In the gauge theory that we are considering,

$$\widetilde{Z} = \hbar F_\hbar + \sum_{i=1}^{m+n} \sum_{\ell=1}^{L} \mu_i^\ell F_i^\ell, \qquad (99)$$

where $F_\hbar$ is the generator of U(1)$_\hbar$ and $F_i^\ell$, $\ell = 1, \ldots, L$, are the generators of the maximal torus of U(L)$_i$. Since $Q$ squares to zero in the sector in which $\widetilde{Z} = 0$, we can define the $Q$-cohomology in the space of $\widetilde{Z}$-invariant states and in the algebra of $\widetilde{Z}$-invariant operators. The subalgebra of the latter consisting of the elements represented by local operators is the *twisted chiral ring* of the theory.

The $\mathcal{N} = (2,2)$ supersymmetry algebra with $Z = Z^* = 0$ says that the Hamiltonian $H$ satisfies $\{Q, Q^*\} = 2H$. Therefore, $H$ is positive semidefinite and vacuum states are annihilated by $Q$ and $Q^*$. In particular, vacua have $\widetilde{Z} = 0$. According to Hodge theory, the $Q$-cohomology of states is isomorphic to the space of vacua.

Besides the Hamiltonian, the momentum $P$ is also $Q$-exact: $\{Q, Q_+ - \overline{Q}_-\} = 2P$. It follows that translations act trivially in the $Q$-cohomology. In particular, the twisted chiral ring is commutative since we can switch the order of two local $Q$-cohomology classes along the time axis by moving them around continuously inside the two-dimensional spacetime.

In fact, for the theory considered here, not just the Hamiltonian and the momentum but the entire stress tensor is $Q$-exact. As a consequence, the $Q$-cohomology of states and the twisted chiral ring are topological, and there is a state–operator correspondence between them: the two are isomorphic as vector spaces.

Being topological, the twisted chiral ring can be computed in the effective theory. As a vector space, it is the space of polynomials in the scalar fields $\{\sigma_r^{a_r}\}$ modulo the action of the Weyl group of the gauge group and the relations imposed by the vacuum equations (96). On a vacuum state, specified by a solution of the vacuum equations, an element of the twisted chiral ring acts by evaluation on the solution. Therefore, a vacuum is a simultaneous eigenstate of the elements of the twisted chiral ring.

## 2.4 The correspondence

The Bethe equations (59) and the vacuum equations (96) coincide under the identification

$$\mu_i^\ell = \zeta^\ell - (-1)^{[i]}\left(\lambda_i^\ell + \frac{1}{4}\right)\hbar + \frac{1}{2}c_i\hbar, \tag{100}$$

together with the obvious identification between parameters for which we have been using the same symbols. (We are measuring twisted masses in an appropriate unit so that they are numbers here.)

Thus, the vacua of the gauge theory are identified with the Bethe vectors of the corresponding magnon sector of the spin chain. Under this identification, the elements of the twisted chiral ring are identified with the commuting conserved charges of the spin chain. This is the statement of the Bethe/gauge correspondence.

One conclusion we can immediately draw from the Bethe/gauge correspondence is that the gauge theory has no supersymmetric vacuum unless the assignment $(M_1, \ldots, M_{m+n-1})$ of the ranks of the unitary gauge groups corresponds to a weight of $M(\lambda^1) \otimes \cdots \otimes M(\lambda^L)$. For example, for $(m|n) = (1|1)$, supersymmetry is broken if and only if

$$M_1 > L \tag{101}$$

because the fermionic lowering operator can be applied at most $L$ times, at which point all spin sites are occupied by fermionic excitations. This is consistent with the known result that supersymmetry is broken in a two-dimensional $\mathcal{N} = (2,2)$ supersymmetric gauge theory with gauge group $U(M)$, $L_f$ fundamental chiral multiplets and $L_a$ antifundamental chiral multiplets if $M > \max(L_f, L_a)$ [35].

# 3 String theory realization of the Bethe/gauge correspondence

Although we have presented the Bethe/gauge correspondence for noncompact rational $\mathfrak{gl}(m|n)$ superspin chains, we have not yet explained why such a correspondence should exist. In this section we provide an explanation using string theory.

We will discuss how to construct the vacua of the relevant gauge theories using branes, and how to map these brane configurations to other ones that realize configurations of line defects in four-dimensional Chern–Simons theory with gauge group $G = GL(m|n)$. The emergence of integrable spin chains is understood naturally in the latter setup.

Moreover, we will give an explanation of fermionic dualities known in the literature of integrable superspin chains.

## 3.1 Brane construction of the gauge theory vacua

The gauge theory and its vacua described in sections 2.3 can be constructed with branes in string theory. In fact, we have already represented the corresponding Bethe vectors graphically in a way that makes the connection to the brane construction transparent.

### 3.1.1 Semiclassical type IIA configuration

The construction uses NS5-branes

$$\text{NS5}_i, \quad i = 1, \ldots, m+n, \tag{102}$$

D4-branes

$$\text{D4}_i^\ell, \quad i = 1, \ldots, m+n, \quad \ell = 1, \ldots, L, \tag{103}$$

and D2-branes

$$\mathrm{D2}_r^{a_r}, \quad r = 1, \ldots, m+n-1, \quad a_r = 1, \ldots, M_r, \tag{104}$$

in type IIA superstring theory. The indices $i$ and $r$ are $\mathbb{Z}_2$-graded as before.

First, let us consider the case in which all FI parameters are zero. In this case, a semiclassical brane configuration for a vacuum state of the gauge theory is summarized as follows:

$$
\begin{array}{rclcccccccc}
\text{Spacetime}: & \mathbb{R} \times \check{S}^1 \times & \mathbb{C} & \times & \mathbb{R}_X & \times & \mathbb{R}_Y & \times \mathbb{R}_{+\hbar}^2 \times \mathbb{R}_{-\hbar}^2, \\
\text{NS5}_i\ ([i]=\bar{0}): & \mathbb{R} \times \check{S}^1 \times & \mathbb{C} & \times & \{X_i\} & \times & \{Y_i\} & \times \mathbb{R}_{+\hbar}^2 \times \{0\}, \\
\text{NS5}_i\ ([i]=\bar{1}): & \mathbb{R} \times \check{S}^1 \times & \mathbb{C} & \times & \{X_i\} & \times & \{Y_i\} & \times \{0\} \times \mathbb{R}_{-\hbar}^2, \\
\text{D4}_i^\ell\ ([i]=\bar{0}): & \mathbb{R} \times \check{S}^1 \times & \{\mu_i^\ell\} & \times & \{X_i\} & \times & [Y_i, \infty) & \times \mathbb{R}_{+\hbar}^2 \times \{0\}, \\
\text{D4}_i^\ell\ ([i]=\bar{1}): & \mathbb{R} \times \check{S}^1 \times & \{\mu_i^\ell\} & \times & \{X_i\} & \times & [Y_i, \infty) & \times \{0\} \times \mathbb{R}_{-\hbar}^2, \\
\text{D2}_r^{a_r}: & \mathbb{R} \times \check{S}^1 \times & \{\sigma_r^{a_r}\} & \times & [X_r, X_{r+1}] & \times & \{\widetilde{Y}_r\} & \times \{0\} \times \{0\}.
\end{array}
\tag{105}
$$

In the spacetime, $\check{S}^1$ is a circle, $\mathbb{R}_X$ and $\mathbb{R}_Y$ are lines, and $\mathbb{R}_{+\hbar}^2$ and $\mathbb{R}_{-\hbar}^2$ are planes. Corresponding to the vanishing FI parameters, we have

$$Y_i = \widetilde{Y}_r = 0, \tag{106}$$

for all $i$ and $r$.

All of these branes wrap the cylinder $\mathbb{R} \times \check{S}^1$, which is the spacetime of the gauge theory. The branes $\text{NS5}_i$ and $\text{D4}_i^\ell$ extend over $\mathbb{R}_{(-1)^{[i]}\hbar}^2$ and are located at the origin of $\mathbb{R}_{(-1)^{[i]+\bar{1}}\hbar}^2$. Moreover, $\text{NS5}_i$ extends over $\mathbb{C}$, whereas $\text{D4}_i^\ell$ extends along $\mathbb{R}_X$ and ends on $\text{NS5}_i$. Along $\mathbb{R}_X$, the NS5-branes are ordered according to the ordered basis of $\mathbb{C}^{m|n}$ specifying the Dynkin diagram of $\mathfrak{gl}(m|n)$:

$$X_1 < X_2 < \cdots < X_{m+n}. \tag{107}$$

The graphical representation (62) for a Bethe vector can be reinterpreted as the above brane configuration. In that picture, the vertical direction is the direction of $\mathbb{C}$ and the horizontal direction is $\mathbb{R}_X$; the vertical lines are the NS5-branes. The diagonal lines ending on the vertical ones are the D4-branes. The horizontal line segments between the $r$th and $(r+1)$st vertical lines are the D2-branes $\text{D2}_r^{a_r}$, $a_r = 1, \ldots, M_r$, suspended between the two NS5-branes:

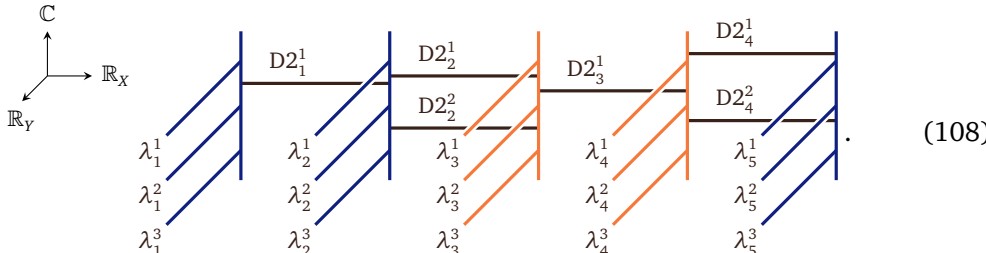

$$\tag{108}$$

Strings stretched between $\text{D2}_r^{a_r}$ and $\text{D2}_r^{b_r}$ produce the components $(\mathsf{V}_r)^{a_r}{}_{b_r}$ and $(\mathsf{V}_r)^{b_r}{}_{a_r}$ of the vector multiplet $\mathsf{V}_r$ for the gauge group factor $\mathrm{U}(M_r)$. The D2-branes can move along $\mathbb{C}$, and the position of $\text{D2}_r^{a_r}$ in $\mathbb{C}$ determines the scalar field $\sigma_r^{a_r}$ of the twisted chiral multiplet $\Sigma_r^{a_r}$.

If $[r] = [r+1]$, the D2-branes suspended between $\text{NS5}_r$ and $\text{NS5}_{r+1}$ can also move along $\mathbb{R}_{(-1)^{[r]}\hbar}^2$, over which the two NS5-branes extend. Accordingly, in this case strings with both ends attached on these D2-branes give rise to an additional chiral multiplet, namely the adjoint chiral multiplet $\phi_r$. The positions of the D2-branes in $\mathbb{R}_{(-1)^{[r]}\hbar}^2$ are the diagonal components of the scalar field in $\phi_r$.

Strings stretched between $\text{D2}_{r-1}^{a_{r-1}}$ and $\text{D2}_r^{b_r}$ yield the components $(\mathsf{P}_r)_{a_{r-1}}{}^{b_r}$ and $(\widetilde{\mathsf{P}}_r)^{b_r}{}_{a_{r-1}}$ of the bifundamental chiral multiplets between $\mathrm{U}(M_{r-1})$ and $\mathrm{U}(M_r)$. Strings from $\text{D2}_i^{a_i}$ to $\text{D4}_i^\ell$ are responsible for $(\widetilde{\mathsf{Q}}_i)^\ell{}_{a_i}$ and those from $\text{D4}_i^\ell$ to $\text{D2}_{i-1}^{a_{i-1}}$ give $(\mathsf{Q}_i)^{a_{i-1}}{}_\ell$.

Various parameters of the gauge theory are identified as follows. The gauge coupling for $U(M_r)$ is $\sqrt{g_s/l_s(X_{r+1}-X_r)}$, where $g_s$ is the string coupling and $l_s$ is the string length. The position of $D4_i^\ell$ in $\mathbb{C}$ determines the twisted mass $\mu_i^\ell$. The FI parameter for $U(M_r)$ is $(Y_{r+1}-Y_r)/g_sl_s$, while the theta angle is given by the difference in the periodic scalars on $NS5_r$ and $NS5_{r+1}$ (up to a shift by $i\pi$ which we will explain shortly). Since we are taking $Y_i = 0$ for all $i$, all FI parameters are zero. Introducing the twisted masses proportional to $\hbar$ requires turning on a nontrivial B-field. For the moment we take $\hbar = 0$.

The rotation symmetry of the directions orthogonal to the D2-brane worldvolumes becomes a global symmetry of the gauge theory. The rotation symmetry $U(1)_\mathbb{C}$ of $\mathbb{C}$ is an axial R-symmetry, under which the vector multiplet scalars have charge 2. The rotation symmetries $U(1)_{\mathbb{R}^2_{\pm\hbar}}$ of $\mathbb{R}^2_{\pm\hbar}$ are vector R-symmetries. The adjoint chiral multiplet $\phi_r$ has charge $(2,0)$ or $(0,2)$ under $U(1)_{\mathbb{R}^2_{+\hbar}} \times U(1)_{\mathbb{R}^2_{-\hbar}}$, depending on whether $[r] = \bar{0}$ or $\bar{1}$.

Now, let us turn on FI parameters by displacing the NS5-branes along $\mathbb{R}_Y$ by different amounts. As we vary their positions, the D2-branes suspended between them get rotated in $\mathbb{R}_X \times \mathbb{R}_Y$ by various angles. Such a configuration no longer preserves supersymmetry. If the twisted masses are generic, the D2-branes cannot stretch between D4-branes without breaking supersymmetry either. Moreover, if $M_r > L$ and $[r] \neq [r+1]$, suspending $M_r$ D2-branes between $NS5_r$ and the $L$ D4-branes ending on $NS5_{r+1}$ (or between $NS5_{r+1}$ and the $L$ D4-branes ending on $NS5_r$) breaks supersymmetry by the s-rule. It appears that there are no supersymmetric vacua for generic FI parameters, twisted masses and magnon numbers.

This analysis is semiclassical, however. Quantum mechanically, D4-branes bend NS5-branes on which they end and the conclusion is altered.

### 3.1.2 Lift to M-theory

Important aspects of the quantum corrections to the above brane configuration can be understood by uplift to M-theory. The M-theory spacetime contains an additional compact direction $S_M^1$. Let $\vartheta$ be its coordinate with period $2\pi$, and introduce a complex coordinate $Y + i\vartheta$ for the cylinder $\mathbb{R}_Y \times S_M^1$. Further introducing $w = e^{-(Y+i\vartheta)}$, we map the cylinder to the punctured complex plane $\mathbb{C}^\times$.

All NS5-branes and D4-branes are lifted to M5-branes in M-theory. For each $i$, $NS5_i$ and $D4_i^\ell$, $\ell = 1, \ldots, L$, merge into a single M5-brane $M5_i$, wrapping a Riemann surface $\Sigma_i$ in $\mathbb{C} \times \mathbb{C}^\times$. The D2-brane $D2_r^{a_r}$ is lifted to an M2-brane $M2_r^{a_r}$ stretched between $M5_r$ and $M5_{r+1}$. Hence, the brane configuration in M-theory is as follows:

$$
\begin{aligned}
\text{Spacetime:} & \ \mathbb{R} \times \check{S}^1 \times & \mathbb{C} \times \mathbb{C}^\times & \times & \mathbb{R}_X & \times \mathbb{R}^2_{+\hbar} \times \mathbb{R}^2_{-\hbar}, \\
M5_i \ ([i]=\bar{0}): & \ \mathbb{R} \times \check{S}^1 \times & \Sigma_i & \times & \{X_i\} & \times \mathbb{R}^2_{+\hbar} \times \{0\}, \\
M5_i \ ([i]=\bar{1}): & \ \mathbb{R} \times \check{S}^1 \times & \Sigma_i & \times & \{X_i\} & \times \{0\} \times \mathbb{R}^2_{-\hbar}, \\
M2_r^{a_r}: & \ \mathbb{R} \times \check{S}^1 \times & \{(\sigma_r^{a_r}, w_r^{a_r})\} & \times & [X_r, X_{r+1}] & \times \{0\} \times \{0\}.
\end{aligned}
\tag{109}
$$

In terms of the coordinates $(z,w)$ of $\mathbb{C} \times \mathbb{C}^\times$, the Riemann surface $\Sigma_i$ is defined by the equation

$$
w = q_i \prod_{\ell=1}^{L} (z - \mu_i^\ell),
\tag{110}
$$

where $q_i$ is a constant. The zero of $w$ at $z = \mu_i^\ell$ describes $D4_i^\ell$, which extends to $+\infty$ in $\mathbb{R}_Y$. If $\Sigma_r$ and $\Sigma_{r+1}$ intersect in $\mathbb{C} \times \mathbb{C}^\times$, then $M2_r^{a_r}$ can be placed at an intersection point so that its worldvolume is orthogonal to the M5-branes.

Therefore, the M2-branes can be suspended between the M5-branes in a manner that preserves supersymmetry if $M2_r^{a_r}$ is placed at $z = \sigma_r^{a_r}$ and the coordinate $\sigma_r^{a_r}$, for each $r$ and $a_r$,

satisfies the condition

$$q_r \prod_{\ell=1}^{L}(\sigma_r^{a_r} - \mu_r^\ell) = w_r = q_{r+1} \prod_{\ell=1}^{L}(\sigma_r^{a_r} - \mu_{r+1}^\ell). \tag{111}$$

Comparing these equations with the vacuum equations (96) for $\hbar = 0$,

$$e^{\tau_r} \prod_{\ell=1}^{L} \frac{\sigma_r^{a_r} - \mu_{r+1}^\ell}{\sigma_r^{a_r} - \mu_r^\ell} = (-1)^{\delta_{[r],[r+1]}}, \tag{112}$$

we see that they coincide if we identify

$$(-1)^{\delta_{[r],[r+1]}} e^{\tau_r} = \frac{q_{r+1}}{q_r}. \tag{113}$$

Since $\vartheta_i = (-1)^{[i]} \arg q_i$ is the classical value of the periodic scalar field on $NS5_i$, the difference $(-1)^{[r+1]}\vartheta_{r+1} - (-1)^{[r]}\vartheta_r$ is indeed equal to $\theta_r$, up to a shift by $i\pi$.

### 3.1.3 Turning on $\hbar$

Finally, we explain how to make $\hbar \neq 0$. The global symmetry $U(1)_\hbar$ is the antidiagonal subgroup of $U(1)_{\mathbb{R}^2_{+\hbar}} \times U(1)_{\mathbb{R}^2_{-\hbar}}$. To turn on the twisted masses for $U(1)_\hbar$, we follow the fluxtrap procedure [36, 37]. This is done as follows.

First, we compactify $\mathbb{C}$ to a torus $T^2 \cong \mathbb{C}/(R_1\mathbb{Z} + iR_2\mathbb{Z})$ in the type IIA setup and apply T-duality on both directions of $T^2$. The D2-branes become D4-branes wrapping the dual torus $\check{T}^2 \cong \mathbb{C}/(\check{R}_1\mathbb{Z} + i\check{R}_2\mathbb{Z})$. Next, we twist the product between $\check{T}^2$ and $\mathbb{R}^2_{+\hbar} \times \mathbb{R}^2_{-\hbar}$ by the action of $U(1)_\hbar$. (More precisely, we replace $\check{T}^2 \times \mathbb{R}^2_{+\hbar} \times \mathbb{R}^2_{-\hbar}$ with the quotient of $\mathbb{C} \times \mathbb{R}^2_{+\hbar} \times \mathbb{R}^2_{-\hbar}$ such that translations on $\mathbb{C}$ by $\check{R}_1$ and $i\check{R}_2$ are accompanied by the action of the elements $\exp(\mathrm{Re}\,\hbar)$ and $\exp(\mathrm{Im}\,\hbar)$ of $U(1)_\hbar$, respectively.) Last, we apply T-duality on $\check{T}^2$ and decompactify $T^2$ to $\mathbb{C}$ by taking $R_1, R_2 \to \infty$. This last T-duality yields a certain B-field due to the twist in the product between $\check{T}^2$ and $\mathbb{R}^2_{+\hbar} \times \mathbb{R}^2_{-\hbar}$ introduced earlier.

From the point of view of the gauge theory, the first step amounts to lifting the two-dimensional theory on $\mathbb{R} \times \check{S}^1$ to a four-dimensional theory on $\mathbb{R} \times \check{S}^1 \times \check{T}^2$. Then, the second step turns on a holonomy for the background gauge field for $U(1)_\hbar$. The last step dimensionally reduces the four-dimensional theory back to two dimensions. Since the components of a four-dimensional gauge field along $\check{T}^2$ become the complex scalar field for the corresponding two-dimensional gauge field, this procedure induces the twisted masses for $U(1)_\hbar$.

## 3.2 Four-dimensional Chern–Simons theory

We can convert the brane configuration (105) to a configuration realizing line defects in four-dimensional Chern–Simons theory with gauge group $GL(m|n)$. To do so, we apply T-duality along $\check{S}^1$ and then S-duality.

Under these dualities, $\check{S}^1$ is mapped to the dual circle $S^1$, the NS5-branes are mapped to D5-branes, the D4-branes are mapped to D3-branes, and the D2-branes are mapped to F1-branes (fundamental strings). Thus we obtain the following type IIB setup:

$$
\begin{array}{rccccccccc}
\text{Spacetime:} & \mathbb{R} \times & S^1 & \times & \mathbb{C} & \times & \mathbb{R}_X & \times & \mathbb{R}_Y & \times \mathbb{R}^2_{+\hbar} \times \mathbb{R}^2_{-\hbar}, \\
D5_i\,([i]=\bar{0}): & \mathbb{R} \times & S^1 & \times & \mathbb{C} & \times & \{X_i\} & \times & \{Y_i\} & \times \mathbb{R}^2_{+\hbar} \times \{0\}, \\
D5_i\,([i]=\bar{1}): & \mathbb{R} \times & S^1 & \times & \mathbb{C} & \times & \{X_i\} & \times & \{Y_i\} & \times \{0\} \times \mathbb{R}^2_{-\hbar}, \\
D3_i^\ell\,([i]=\bar{0}): & \mathbb{R} \times & \{y_i^\ell\} & \times & \{\mu_i^\ell\} & \times & \{X_i\} & \times & [Y_i,\infty) & \times \mathbb{R}^2_{+\hbar} \times \{0\}, \\
D3_i^\ell\,([i]=\bar{1}): & \mathbb{R} \times & \{y_i^\ell\} & \times & \{\mu_i^\ell\} & \times & \{X_i\} & \times & [Y_i,\infty) & \times \{0\} \times \mathbb{R}^2_{-\hbar}, \\
F1_r^{a_r}: & \mathbb{R} \times & \{\tilde{y}_r^{a_r}\} & \times & \{\sigma_r^{a_r}\} & \times & [X_r,X_{r+1}] & \times & \{\widetilde{Y}_r\} & \times \{0\} \times \{0\}.
\end{array} \tag{114}
$$

The positions of the D3-branes and the F1-branes on $S^1$ are given by the holonomies around $\check{S}^1$ of the gauge fields on their counterparts in the type IIA setup. The B-field inducing the twisted masses for $\mathrm{U}(1)_\hbar$ becomes a Ramond–Ramond (RR) two-form field in the new setup.

As in the two-dimensional theory discussed before, the vacuum sector of the theory on the D5-branes, with $\mathbb{R}$ taken to be the time direction, is captured by the cohomology with respect to a certain supercharge. This supercharge is dual to the supercharge $Q$ of the two-dimensional theory, and we will use the same symbol to denote it.

We claim that the $Q$-invariant sector of the theory, which governs the $Q$-cohomology, is equivalent to four-dimensional Chern–Simons theory with gauge group $\mathrm{GL}(m|n)$.

Before demonstrating this equivalence, let us remark that related brane constructions have appeared in the literature.[4] In [41], Mikhaylov and Witten gave a brane construction of three-dimensional Chern–Simons theory with gauge group $\mathrm{GL}(m|n)$, extending the construction for gauge group $\mathrm{GL}(m)$ given in [43]. Their construction uses $m$ D4-branes and $n$ D4-branes ending on an NS5-brane from opposite sides. The supergroup Chern–Simons theory appears at the intersection of the three kinds of branes. In [39], a construction of four-dimensional Chern–Simons theory with gauge group $\mathrm{GL}(m)$ was proposed. In this construction, $m$ D4-branes end on an NS5-brane.

### 3.2.1 Case with $m = 0$ or $n = 0$

In the case in which all D5-branes are of even type ($n = 0$) or of odd type ($m = 0$), the result just described was derived in [16]. Let us briefly review the derivation in [16].

For $n = 0$, the worldvolume theory on the D5-branes is a deformation of six-dimensional $\mathcal{N} = (1,1)$ super Yang–Mills theory with gauge group $\mathrm{U}(m)$, placed on $\mathbb{R} \times S^1 \times \mathbb{C} \times \mathbb{R}^2_{+\hbar}$. The deformation is what is often called an $\Omega$-*deformation* and controlled by $\hbar$: we have

$$Q^2 = \hbar F_\hbar \,, \tag{115}$$

where $F_\hbar$ is the generator of $\mathrm{U}(1)_\hbar$. (Here we are considering the situation in which there are no D3-branes, hence no $\mathrm{U}(L)$ flavor symmetries.) In the six-dimensional theory, $\mathrm{U}(1)_\hbar$ is the antidiagonal subgroup of the rotation group $\mathrm{U}(1)_{\mathbb{R}^2_{+\hbar}}$ on $\mathbb{R}^2_{+\hbar}$ and the subgroup $\mathrm{U}(1)_{\mathbb{R}^2_{-\hbar}}$ of the R-symmetry group $\mathrm{Spin}(4)$ coming from the rotation symmetry of $\mathbb{R}^2_{-\hbar}$.

Six-dimensional (Euclidean) $\mathcal{N} = (1,1)$ super Yang–Mills theory on $\mathbb{R} \times S^1 \times \mathbb{C} \times \mathbb{R}^2_{+\hbar}$ reduces to two-dimensional $\mathcal{N} = (8,8)$ super Yang–Mills theory on $\mathbb{R}^2_{+\hbar}$ by dimensional reduction. In the undeformed case (when $\hbar = 0$), the supercharge $Q$ belongs to an $\mathcal{N} = (2,2)$ subalgebra of the $\mathcal{N} = (8,8)$ supersymmetry algebra. Accordingly, $\mathcal{N} = (1,1)$ super Yang–Mills theory on $\mathbb{R} \times S^1 \times \mathbb{C} \times \mathbb{R}^2_{+\hbar}$ may be thought of as an $\mathcal{N} = (2,2)$ supersymmetric gauge theory on $\mathbb{R}^2_{+\hbar}$, with infinite-dimensional target space and gauge group. The $\Omega$-deformation of the six-dimensional theory induces an $\Omega$-deformation of the two-dimensional theory.

In general, the $Q$-invariant sector of an $\Omega$-deformed $\mathcal{N} = (2,2)$ supersymmetric gauge theory on $\mathbb{R}^2$ is equivalent to a zero-dimensional theory [16, 23, 24]. Let $\mathcal{G}$ be the gauge group

---

[4]To relate these brane constructions to ours, we endow $\mathbb{R}^2_{+\hbar} \times \mathbb{R}^2_{-\hbar}$ with a Taub–NUT metric. (The $Q$-invariant sector is independent of the choice of metric as long as it preserves the rotational symmetries of $\mathbb{R}^2_{+\hbar}$ and $\mathbb{R}^2_{-\hbar}$.) If we regard the Taub–NUT space as a circle fibration over $\mathbb{R}^3$, then $\mathbb{R}^2_{+\hbar} \times \{0\}$ and $\{0\} \times \mathbb{R}^2_{-\hbar}$ are two semi-infinite cigar-shaped subspaces extending in the opposite directions such that their tips touch at the origin of $\mathbb{R}^3$; see [38], appendix B. T-duality in the direction of the circle fibers produces an NS5-brane which sits at the origin of $\mathbb{R}^3$ and extends in the directions transverse to the Taub–NUT space. The D5-branes wrapping the two cigars are turned into D4-branes ending on the NS5-brane from two sides. Considering the case with $n = 0$, we reproduce the construction of [39]. The field theory counterpart of this T-duality was analyzed in [40]. From the D4–NS5 brane configuration we obtain the brane configuration of [41], roughly speaking, by further replacing $\mathbb{C}$ with a cylinder, taking T-duality in the circumferential direction of the cylinder, and decompactifying the dual cylinder. (Such T-duality was considered in [42].)

of the theory and $\mathcal{G}_{\mathbb{C}}$ be its complexification. By $\mathcal{N} = (2, 2)$ supersymmetry, the chiral multiplets take values in a Kähler manifold $\mathcal{X}$ with $\mathcal{G}_{\mathbb{C}}$-action. The superpotential is a $\mathcal{G}_{\mathbb{C}}$-invariant holomorphic function $\mathcal{W}$ on $\mathcal{X}$. The path integral with insertion of $Q$-invariant observables localizes to a $\mathcal{G}_{\mathbb{C}}$-invariant submanifold $\gamma$ of $\mathcal{X}$. This submanifold is essentially a Lefschetz thimble: $\gamma$ is the union of all gradient flows generated by the real part of $\mathcal{W}/\hbar$, terminating on the $\mathcal{G}_{\mathbb{C}}$-orbit of a chosen critical point of $\mathcal{W}$. (For simplicity, we assume that the critical points of $\mathcal{W}$ are nondegenerate up to the $\mathcal{G}_{\mathbb{C}}$-action.) The localized path integral takes the form

$$\int_{\gamma/\mathcal{G}_{\mathbb{C}}} \exp\left(\frac{2\pi}{\hbar}\mathcal{W}\right)\mathcal{O}, \tag{116}$$

where $\mathcal{O}$ descends from the $Q$-invariant observables inserted in the path integral. This is the path integral for a zero-dimensional gauge theory with gauge group $\mathcal{G}_{\mathbb{C}}$ and target space $\gamma$. The action functional is $-2\pi\mathcal{W}/\hbar$.

The remarkable aspect of this localization phenomenon is that the gauge group gets complexified. In the localization process, some fermionic fields have zero modes. They may be regarded as ghost fields for partial gauge fixing that breaks $\mathcal{G}_{\mathbb{C}}$ down to $\mathcal{G}$. Even though the action functional is holomorphic and its real part is not bounded from below, the integral (116) can converge since $\mathrm{Re}(\mathcal{W}/\hbar)$ gets smaller and smaller along the backward gradient flows in $\gamma$.

For the six-dimensional $\mathcal{N} = (1, 1)$ super Yang–Mills theory on $\mathbb{R} \times S^1 \times \mathbb{C} \times \mathbb{R}^2_{+\hbar}$, the gauge group $\mathcal{G}$ is the space of maps from $\mathbb{R} \times S^1 \times \mathbb{C}$ to $\mathrm{U}(m)$. In addition to the vector multiplet, the theory has three chiral multiplets in the adjoint representation of $\mathcal{G}$. Their scalar fields are $Q$-invariant and can be combined into a one-form on $\mathbb{R} \times S^1 \times \mathbb{C}$:

$$\mathcal{A} = (A_x + iX)\mathrm{d}x + (A_y + iY)\mathrm{d}y + A_{\bar{z}}\,\mathrm{d}\bar{z}. \tag{117}$$

Here, $A_x$, $A_y$ are the components of the gauge field along $\mathbb{R} \times S^1$, $A_{\bar{z}}$ is the antiholomorphic component of the gauge field along $\mathbb{C}$, and $X$, $Y$ are two of the four scalar fields of the six-dimensional theory associated to motions along $\mathbb{R}_X$ and $\mathbb{R}_Y$, respectively. The superpotential is given by

$$\mathcal{W} = -\frac{i}{e^2} \int_{\mathbb{R} \times S^1 \times \mathbb{C}} \mathrm{d}z \wedge \mathrm{tr}\left(\mathcal{A} \wedge \mathrm{d}\mathcal{A} + \frac{2}{3}\mathcal{A} \wedge \mathcal{A} \wedge \mathcal{A}\right), \tag{118}$$

where $e$ is the gauge coupling and $\mathrm{tr}$ is an invariant symmetric bilinear form on the Lie algebra of $\mathrm{U}(m)$, which we can take to be the trace in the defining representation.

According to the localization argument, the $\Omega$-deformation of the six-dimensional theory is equivalent to a zero-dimensional gauge theory. This zero-dimensional theory has infinite-dimensional target space and gauge group, and can be more naturally interpreted as a four-dimensional gauge theory. Its action is

$$-\frac{2\pi i}{\hbar e^2} \int_{\Sigma \times \mathbb{C}} \mathrm{d}z \wedge \mathrm{tr}_{\mathbb{C}^m}\left(\mathcal{A}^{00} \wedge \mathrm{d}\mathcal{A}^{00} + \frac{2}{3}\mathcal{A}^{00} \wedge \mathcal{A}^{00} \wedge \mathcal{A}^{00}\right). \tag{119}$$

Here, we have written the partial $\mathfrak{gl}(m)$ connection (117) as $\mathcal{A}^{00}$ to emphasize its place in the Lie superalgebra $\mathfrak{gl}(m|n)$ that will arise later. This is the action for four-dimensional Chern–Simons theory. Thus we conclude that the $Q$-invariant sector of the $\Omega$-deformed six-dimensional $\mathcal{N} = (1, 1)$ super Yang–Mills theory on $\mathbb{R} \times S^1 \times \mathbb{C} \times \mathbb{R}^2_{+\hbar}$ with gauge group $\mathrm{U}(m)$ is equivalent to four-dimensional Chern–Simons theory on $\mathbb{R} \times S^1 \times \mathbb{C}$ with gauge group $\mathrm{GL}(m)$ and coupling given by $\hbar$.

Similarly, if we consider the case $m = 0$, the worldvolume theory on the D5-branes is an $\Omega$-deformed six-dimensional $\mathcal{N} = (1, 1)$ super Yang–Mills theory on $\mathbb{R} \times S^1 \times \mathbb{C} \times \mathbb{R}^2_{-\hbar}$ with gauge

group $U(n)$. Its $Q$-invariant sector is equivalent to four-dimensional Chern–Simons theory on $\mathbb{R} \times S^1 \times \mathbb{C}$ with gauge group $GL(n)$ and action

$$+\frac{2\pi i}{\hbar e^2} \int_{\mathbb{R} \times S^1 \times \mathbb{C}} dz \wedge \mathrm{tr}_{\mathbb{C}^n}\left( \mathcal{A}^{11} \wedge d\mathcal{A}^{11} + \frac{2}{3}\mathcal{A}^{11} \wedge \mathcal{A}^{11} \wedge \mathcal{A}^{11} \right), \tag{120}$$

with the partial $\mathfrak{gl}(n)$ connection $\mathcal{A}^{11}$ defined in the same way as $\mathcal{A}^{00}$.

### 3.2.2 Case with nonzero $m$ and $n$

Let us turn to the case in which $m$ and $n$ are both nonzero. In this case, the two sets of D5-branes lead to two copies of four-dimensional Chern–Simons theory on $\mathbb{R} \times S^1 \times \mathbb{C}$, one with gauge group $GL(m)$ and the other with gauge group $GL(n)$, with opposite couplings. Arranging $\mathcal{A}^{00}$ and $\mathcal{A}^{11}$ into a matrix

$$\mathcal{A}^0 = \begin{pmatrix} \mathcal{A}^{00} & 0 \\ 0 & \mathcal{A}^{11} \end{pmatrix}, \tag{121}$$

we can write the sum of their action functionals as

$$-\frac{2\pi i}{\hbar e^2} \int_{\mathbb{R} \times S^1 \times \mathbb{C}} dz \wedge \mathrm{str}_{\mathbb{C}^{m|n}}\left( \mathcal{A}^0 \wedge d\mathcal{A}^0 + \frac{2}{3}\mathcal{A}^0 \wedge \mathcal{A}^0 \wedge \mathcal{A}^0 \right). \tag{122}$$

The two copies are coupled through strings stretched between the two sets of D5-branes. These strings produce a four-dimensional $\mathcal{N} = 2$ hypermultiplet on $\mathbb{R} \times S^1 \times \mathbb{C}$ in the bifundamental representation of $GL(m) \times GL(n)$. It consists of bosonic complex scalars

$$q \in \mathrm{Hom}(\mathbb{C}^m, \mathbb{C}^n), \tag{123}$$

$$\tilde{q}^\dagger \in \mathrm{Hom}(\mathbb{C}^m, \mathbb{C}^n), \tag{124}$$

$$q^\dagger \in \mathrm{Hom}(\mathbb{C}^n, \mathbb{C}^m), \tag{125}$$

$$\tilde{q} \in \mathrm{Hom}(\mathbb{C}^n, \mathbb{C}^m), \tag{126}$$

and fermionic Weyl spinors

$$\psi \in \mathrm{Hom}(\mathbb{C}^m, \mathbb{C}^n), \tag{127}$$

$$\tilde{\psi}^\dagger \in \mathrm{Hom}(\mathbb{C}^m, \mathbb{C}^n), \tag{128}$$

$$\psi^\dagger \in \mathrm{Hom}(\mathbb{C}^n, \mathbb{C}^m), \tag{129}$$

$$\tilde{\psi} \in \mathrm{Hom}(\mathbb{C}^n, \mathbb{C}^m). \tag{130}$$

In the absence of coupling to the two copies of four-dimensional Chern–Simons theory, the bifundamental hypermultiplet preserves eight supercharges. The supercharge $Q$ is a linear combination of these supercharges such that the generators of translations on $\mathbb{R} \times S^1$ and antiholomorphic translations on $\mathbb{C}$ are $Q$-exact. By redefining fields if necessary, we can take $Q$ to be the supercharge used in the holomorphic–topological twist studied in [44], with the parameter $t = i$.

It turns out that most of the action for the hypermultiplet is $Q$-exact. The remaining part of the action can be expressed in a suggestive form. Endow the cylinder $\mathbb{R} \times S^1$ with a complex coordinate $w$, and define

$$\mathcal{A}^{10} = -\tilde{\psi}^\dagger_- dw + \psi_- d\bar{w} - \frac{1}{2}(\tilde{\psi}^\dagger_+ - \psi_+)d\bar{z}, \tag{131}$$

$$\mathcal{A}^{01} = \psi^\dagger_- dw + \tilde{\psi}_- d\bar{w} + \frac{1}{2}(\tilde{\psi}^\dagger_+ + \psi^\dagger_+)d\bar{z} \tag{132}$$

and

$$c^{10} = 4\mathrm{i}q^{\dagger}, \tag{133}$$

$$c^{01} = 4\tilde{q}^{\dagger}, \tag{134}$$

$$b^{10} = \tilde{q}, \tag{135}$$

$$b^{01} = -\mathrm{i}q, \tag{136}$$

$$B^{10} = \mathrm{i}(\tilde{\psi}_{+} - \psi_{+}^{\dagger}), \tag{137}$$

$$B^{01} = -\mathrm{i}(\psi_{+} + \tilde{\psi}_{+}^{\dagger}). \tag{138}$$

We introduce a matrix

$$\mathcal{A}^{1} = \begin{pmatrix} 0 & \mathcal{A}^{01} \\ \mathcal{A}^{10} & 0 \end{pmatrix} \tag{139}$$

and matrices $c^{1}$, $b^{1}$, $B^{1}$ defined likewise. On these matrices $Q$ acts by

$$Q \cdot \mathcal{A}^{1} = -\mathrm{d}'c^{1}, \tag{140}$$

$$Q \cdot c^{1} = 0, \tag{141}$$

$$Q \cdot b^{1} = B^{1}, \tag{142}$$

$$Q \cdot B^{1} = 0, \tag{143}$$

where

$$\mathrm{d}' = \mathrm{d} - \mathrm{d}z\,\partial_{z} = \mathrm{d}w\,\partial_{w} + \mathrm{d}\bar{w}\,\partial_{\bar{w}} + \mathrm{d}\bar{z}\,\partial_{\bar{z}}. \tag{144}$$

The non-$Q$-exact part of the action is

$$-\frac{2\pi\mathrm{i}}{\hbar e^{2}} \int_{\mathbb{R} \times S^{1} \times \mathbb{C}} \mathrm{d}z \wedge \mathrm{str}_{\mathbb{C}^{m|n}}(\mathcal{A}^{1} \wedge \mathrm{d}\mathcal{A}^{1}). \tag{145}$$

(Since this is quadratic in fermions, the prefactor is inessential.)

To describe the intersecting D5-branes, we couple this bifundamental hypermultiplet to the two copies of four-dimensional Chern–Simons theory by identifying the flavor groups $\mathrm{GL}(m)$ and $\mathrm{GL}(n)$ with the gauge groups of the latter. Concretely, we replace the de Rham differential that appears in the above formulas with the gauge-covariant differential

$$\mathrm{d}_{\mathcal{A}^{0}} = \mathrm{d} + \mathcal{A}^{0}. \tag{146}$$

Thus, the action of $Q$ on the fields is modified to

$$Q \cdot \mathcal{A}^{1} = -\mathrm{d}'_{\mathcal{A}^{0}}c^{1}, \tag{147}$$

$$Q \cdot c^{1} = 0, \tag{148}$$

$$Q \cdot b^{1} = B^{1}, \tag{149}$$

$$Q \cdot B^{1} = 0, \tag{150}$$

and the action functional for the bifundamental hypermultiplet becomes

$$-\frac{2\pi\mathrm{i}}{\hbar e^{2}} \int_{\mathbb{R} \times S^{1} \times \mathbb{C}} \mathrm{d}z \wedge \mathrm{str}_{\mathbb{C}^{m|n}}(\mathcal{A}^{1} \wedge \mathrm{d}_{\mathcal{A}^{0}}\mathcal{A}^{1}). \tag{151}$$

Combining $\mathcal{A}^{0}$ and $\mathcal{A}^{1}$ into a single matrix

$$\mathcal{A} = \mathcal{A}^{0} + \mathcal{A}^{1} = \begin{pmatrix} \mathcal{A}^{00} & \mathcal{A}^{01} \\ \mathcal{A}^{10} & \mathcal{A}^{11} \end{pmatrix}, \tag{152}$$

we can write the total action, which is the sum of the actions (122) and (151), as

$$-\frac{2\pi i}{\hbar e^2} \int_{\mathbb{R}\times S^1 \times \mathbb{C}} \mathrm{d}z \wedge \mathrm{str}_{\mathbb{C}^{m|n}}\left( \mathcal{A} \wedge \mathrm{d}\mathcal{A} + \frac{2}{3}\mathcal{A} \wedge \mathcal{A} \wedge \mathcal{A} \right). \tag{153}$$

This is the action for four-dimensional Chern–Simons theory with gauge group $\mathrm{GL}(m|n)$.

Before concluding that we have obtained the desired theory, we need to solve two problems. First, although the above action is invariant under $\mathrm{GL}(m|n)$ gauge transformation, the gauge group of the theory is still $\mathrm{GL}(m) \times \mathrm{GL}(n)$, not $\mathrm{GL}(m|n)$. Second, the gauge-invariant action (151) for the bifundamental hypermultiplet is not $Q$-invariant due to the coupling to $\mathcal{A}^0$. Its $Q$-variation gives

$$-\frac{4\pi i}{\hbar e^2} \int_{\mathbb{R}\times S^1 \times \mathbb{C}} \mathrm{d}z \wedge \mathrm{str}_{\mathbb{C}^{m|n}}(\mathcal{A}^1 \wedge [\mathcal{F}^0, c^1]), \tag{154}$$

where $\mathcal{F}^0$ is the curvature of $\mathcal{A}^0$.

The two problems are solved simultaneously if we correct the $Q$-action on $\mathcal{A}^0$ and $B^1$ to

$$Q \cdot \mathcal{A}^0 = \{\mathcal{A}^1, c^1\}, \tag{155}$$

$$Q \cdot B^1 = \frac{1}{2}[\{c^1, c^1\}, b^1]. \tag{156}$$

With this modification, the $Q$-variation of the bosonic action (122) cancels that of the fermionic action (151). At the same time, $c^1$, $b^1$ and $B^1$ can now be interpreted as a ghost, an antighost and an auxiliary field used in the Becchi–Rouet–Stora–Tyutin (BRST) procedure for partial gauge fixing of $\mathrm{GL}(m|n)$ down to $\mathrm{GL}(m) \times \mathrm{GL}(n)$ [45].

To make the last point more explicit, let us introduce a ghost $c^0$, an antighost $b^0$ and an auxiliary field $B^0$ for gauge fixing of $\mathrm{GL}(m) \times \mathrm{GL}(n)$. The BRST charge $Q_{\mathrm{B}}$ acts on the fields by

$$Q_{\mathrm{B}} \cdot \mathcal{A}^0 = -\mathrm{d}'_{\mathcal{A}_0} c^0, \tag{157}$$

$$Q_{\mathrm{B}} \cdot c^0 = \frac{1}{2}\{c^0, c^0\}, \tag{158}$$

$$Q_{\mathrm{B}} \cdot b^0 = B^0, \tag{159}$$

$$Q_{\mathrm{B}} \cdot B^0 = 0 \tag{160}$$

and

$$Q_{\mathrm{B}} \cdot \mathcal{A}^1 = \{c^0, \mathcal{A}^1\}, \tag{161}$$

$$Q_{\mathrm{B}} \cdot c^1 = [c^0, c^1], \tag{162}$$

$$Q_{\mathrm{B}} \cdot b^1 = [c^0, b^1], \tag{163}$$

$$Q_{\mathrm{B}} \cdot B^1 = \{c^0, B^1\}. \tag{164}$$

Let us postulate that

$$Q \cdot c^0 = -\frac{1}{2}\{c^1, c^1\}, \tag{165}$$

$$Q \cdot b^0 = 0, \tag{166}$$

$$Q \cdot B^0 = 0. \tag{167}$$

Then, the modified BRST charge

$$\widehat{Q} = Q_{\mathrm{B}} + Q \tag{168}$$

satisfies $\widehat{Q}^2 = 0$ and

$$\widehat{Q} \cdot \mathcal{A} = -d'_{\mathcal{A}} c \,, \tag{169}$$

$$\widehat{Q} \cdot c = \frac{1}{2} \{c^0, c^0\} - \frac{1}{2} \{c^1, c^1\} + [c^0, c^1] \,, \tag{170}$$

$$\widehat{Q} \cdot b = B \,, \tag{171}$$

$$\widehat{Q} \cdot B = 0 \,, \tag{172}$$

where

$$c = c^0 + c^1 \,, \tag{173}$$

$$b = b^0 + b^1 \,, \tag{174}$$

$$B = B^0 + B^1 + [c^0, b^1] \,. \tag{175}$$

The $\widehat{Q}$-cohomology computes the mixed Lie superalgebra cohomology defined in [46].

Thus, we conclude that the $Q$-invariant sector of the theory on the intersecting D5-branes is four-dimensional Chern–Simons theory with gauge group GL($m|n$).

## 3.3 Emergence of the spin chain

The D3-branes and the F1-branes in the type IIB setup (114) intersect the D5-branes along lines in $\mathbb{R} \times S^1 \times \mathbb{C}$. As such, they create line defects in the four-dimensional Chern–Simons theory on $\mathbb{R} \times S^1 \times \mathbb{C}$, extending along $\mathbb{R}$ and supported at points in $\mathbb{C}$. Such a configuration of line defects in four-dimensional Chern–Simons theory is naturally identified with an integrable spin chain [17–19]. We now show that this spin chain is precisely the one that appears in the Bethe/gauge correspondence.

### 3.3.1 Line defects and spin chains

Let us first explain the relation between line defects in four-dimensional Chern–Simons theory and integrable spin chains.

Consider four-dimensional Chern–Simons theory on $\mathbb{R} \times \mathbb{R} \times \mathbb{C}$, with gauge group $G$ which we take to be a complex simple Lie supergroup. Its field is a partial $G$-connection of the form

$$\mathcal{A} = \mathcal{A}_x \, dx + \mathcal{A}_y \, dy + \mathcal{A}_{\bar{z}} \, d\bar{z} \,. \tag{176}$$

We insert line defects

$$\mathcal{L}^\ell \,, \quad \ell = 1, \dots, L \,, \tag{177}$$

extending in the $x$-direction, which we regard as the time direction. Along the $y$-axis, we arrange $\mathcal{L}^1, \dots, \mathcal{L}^L$ in the ascending order. They are supported at points $\zeta^1, \dots, \zeta^L$ in $\mathbb{C}$.

Solutions of the equation of motion for four-dimensional Chern–Simons theory, away from the line defects, are flat connections. Away from the line defects, flat connections can be gauged away. Then, all information about the state of the theory is localized in the neighborhoods of the line defects, and the Hilbert space $V$ factorises into the tensor product of the spaces attached to the line defects:

$$V = V^1 \otimes \cdots \otimes V^L \,. \tag{178}$$

This is identified with the Hilbert space of an open spin chain with $L$ sites. The space $V^\ell$ supported on $\mathcal{L}^\ell$ is the state space for the $\ell$th spin.

The four-dimensional Chern–Simons theory is topological on $\mathbb{R} \times \mathbb{R}$ and holomorphic on $\mathbb{C}$. Due to the topological invariance on $\mathbb{R} \times \mathbb{R}$, the Hamiltonian is zero. To change the state,

we can insert a Wilson line extending in the $y$-direction, crossing the $L$ line defects introduced earlier. This Wilson line is a non-gauge-invariant operator acting on $V$ and interpreted as a monodromy matrix $T(\sigma)$ in the spin chain. The spectral parameter $\sigma$ is the position of the Wilson line in $\mathbb{C}$, and the holomorphy on $\mathbb{C}$ implies that $T(\sigma)$ is holomorphic in $\sigma$.

If we introduce two Wilson lines and make them intersect in $\mathbb{R} \times \mathbb{R}$, we get an R-matrix at the intersection. The two sides of the RTT relation (33) correspond to two different configurations of two open Wilson lines crossing each other and the line defects $\mathcal{L}^1, \ldots, \mathcal{L}^L$. The topological invariance on $\mathbb{R} \times \mathbb{R}$ and the existence of the extra dimensions $\mathbb{C}$ imply the equivalence of the two configurations.

This R-matrix can be computed by perturbation theory [19], and it was found to be the R-matrix for the rational spin chain with $G$ symmetry. Therefore, this setup produces an open rational spin chain. For $G = \mathrm{GL}(m|n)$, the R-matrix is the one given in (34), up to some equivalence relations.

To obtain a closed spin chain, we simply compactify the $y$-axis $\mathbb{R}$ to $S^1$. Now, flat connections have global gauge-invariant information, namely the holonomy around $S^1$. The holonomy is fixed by the boundary condition at infinity and becomes a parameter of the spin chain. Flat connections can still be gauged away almost everywhere. Away from the line defects, we can make them vanish except on a single line parallel to the line defects and placed between $\mathcal{L}^L$ and $\mathcal{L}^1$, say. The holonomy is then identified with the twist parameter $g$ of the periodic boundary condition in the spin chain. A Wilson loop winding around $S^1$ gives a transfer matrix $t(g, \sigma)$ evaluated in the representation of the Wilson loop.

### 3.3.2 Line defects created by D3-branes

Let us return to the setup for the Bethe/gauge correspondence.

By the topological invariance on $\mathbb{R} \times S^1$, the positions of the D3-branes on $S^1$ do not matter. For each $\ell$, we gather the $m + n$ D3-branes $\mathrm{D3}_i^\ell$, $i = 1, \ldots, m+n$, to the same position $y^\ell$ on $S^1$:

$$y_1^\ell = \cdots = y_{m+n}^\ell = y^\ell . \tag{179}$$

Since they are also located at the same point $\zeta^\ell$ in $\mathbb{C}$ up to first order in $\hbar$, we can regard them as creating a single line defect $\mathcal{L}^\ell$ supported on the line $\mathbb{R} \times \{y^\ell\} \times \{\zeta^\ell\}$ in $\mathbb{R} \times S^1 \times \mathbb{C}$, treating the differences $\mu_i^\ell - \zeta^\ell$, $i = 1, \ldots, m+n$, in the positions in $\mathbb{C}$ as parameters of the line defect. The D3-branes thus create $L$ line defects $\mathcal{L}^1, \ldots, \mathcal{L}^L$.

From the Bethe/gauge correspondence we know what the Hilbert space $V^\ell$ of $\mathcal{L}^\ell$ must be: it is the evaluation module of the Yangian $Y(\mathfrak{gl}(m|n))$ with spectral parameter $\zeta^\ell$, obtained from the Verma module $V_{\lambda^\ell}$ of $\mathfrak{gl}(m|n)$. The F1-branes represent excitations in this Hilbert space.

Let us derive this Hilbert space from the point of view of brane dynamics. In the four-dimensional Chern–Simons theory on $\mathbb{R} \times S^1 \times \mathbb{C}$, the positions of the D5-branes in $\mathbb{R}_X$ and $\mathbb{R}_Y$ parametrize the vacuum expectation values of the gauge fields along the topological directions. For the purpose of identifying the Hilbert space of the line defect, we can consider the situation in which all D5-branes are coincident, say $X_i = Y_i = 0$ for all $i$.

The $Q$-invariant sector of the theory living on the D3-branes that create $\mathcal{L}^\ell$ is the BF theory with gauge group $G = \mathrm{GL}(m|n)$, defined on $\mathbb{R} \times [0, \infty)$. This theory can be obtained from four-dimensional Chern–Simons theory on $\mathbb{R} \times [0, \infty) \times \mathbb{C}$ by dimensional reduction on $\mathbb{C}$. It has the action

$$\frac{1}{\hbar} \int_{\mathbb{R} \times [0,\infty)} \sigma(\mathrm{d}\eta + \eta \wedge \eta), \tag{180}$$

where $\eta$ is the gauge field and $\sigma$ is a scalar field valued in $\mathfrak{g}^*$, the dual of the Lie algebra $\mathfrak{g} = \mathfrak{gl}(m|n)$. The field $\sigma$ comes from the reduction of the antiholomorphic component $A_{\bar{z}}$ of

the four-dimensional gauge field on $\mathbb{C}$.

One way to see that this is the right theory is to note that upon exchanging the directions of $S^1$ and $\mathbb{R}_Y$, the D3-branes are T-dual to D5-branes on $\mathbb{R} \times S^1 \times \mathbb{C} \times \mathbb{R}^2_{\pm \hbar}$ (after compactifying $\mathbb{C}$ to a torus). Since the D5-branes are described by four-dimensional Chern–Simons theory with gauge group $GL(m|n)$, the D3-branes are described by the BF theory. The exchange of two directions amounts to swapping the role of the real and imaginary parts of $A_y$, and does not alter the analysis in any essential way.

We view the BF theory as a Poisson sigma model [47, 48] with target space $\mathfrak{g}^*$. The space of functions on $\mathfrak{g}^*$ is the symmetric algebra $S(\mathfrak{g})$ of $\mathfrak{g}$. Linear functions are elements of $\mathfrak{g}$, and the Poisson bracket between them is given by the Lie bracket. Extending the Poisson bracket to $S(\mathfrak{g})$ by the Leibniz rule, we endow $\mathfrak{g}^*$ with the Poisson structure. The action (180) of the BF theory is related to the action of the Poisson sigma model by integration by parts.

There are two boundaries in the spacetime $\mathbb{R} \times [0, \infty) \subset \mathbb{R} \times \mathbb{R}_Y$, one at $Y = 0$ and the other at $Y = \infty$. It is more convenient to think of the spacetime of the theory as the limit of

$$\mathbb{R} \times [0, r], \tag{181}$$

as $r \to \infty$. Physically, we can realize this setup by making the D3-brane $D3^\ell_i$ end on an NS5-brane $NS5^\ell_i$ with worldvolume

$$\mathbb{R} \times \{y^\ell\} \times \{\mu^\ell_i\} \times \mathbb{R}_X \times \{r\} \times \mathbb{R}^2_{+\hbar} \times \mathbb{R}^2_{-\hbar} \subset \mathbb{R} \times S^1 \times \mathbb{C} \times \mathbb{R}_X \times \mathbb{R}_Y \times \mathbb{R}^2_{+\hbar} \times \mathbb{R}^2_{-\hbar}. \tag{182}$$

Since the BF theory is topological, the value of $r$ does not matter. In particular, we can take the limit $r \to 0$. In this limit the theory reduces to a one-dimensional quantum mechanical system. This quantum mechanical system describes the line defect after coupling to the four-dimensional Chern–Simons theory.

The Hilbert space of the BF theory depends on the boundary conditions imposed on the two boundaries $\mathbb{R} \times \{0\}$ and $\mathbb{R} \times \{r\}$. On each boundary, we impose a boundary condition that defines what is known as a coisotropic brane in the context of Poisson sigma models [49].

The boundary condition on $\mathbb{R} \times \{0\}$ is simple. The imaginary part of the component $\eta_x$ of the gauge field along $\mathbb{R}$ parametrizes the positions of the D3-branes in $\mathbb{R}_X$. These are necessarily fixed to the positions of the D5-branes on the boundary where the D3-branes end on the D5-branes. By holomorphy, the real part must also obey the Dirichlet boundary condition. Thus we have

$$\eta|_{\mathbb{R} \times \{0\}} = 0. \tag{183}$$

The field $\sigma$ is unconstrained on the boundary. This boundary condition completely breaks the gauge symmetry. The global symmetry $G$ on the boundary is used for coupling to the four-dimensional Chern–Simons theory by gauging.

We propose that the boundary condition on $\mathbb{R} \times \{r\}$ is determined by the positions of the NS5-branes in $\mathbb{C}$ as follows. The diagonal part of $\sigma$ parametrizes these positions. Let

$$\mathfrak{g} = \mathfrak{n}_- \oplus \mathfrak{h} \oplus \mathfrak{n}_+ \tag{184}$$

be the triangular decomposition of $\mathfrak{g}$ with respect to the chosen basis; thus $\mathfrak{h}$ is spanned by diagonal matrices, and $\mathfrak{n}_+$ and $\mathfrak{n}_-$ are spanned by strictly upper triangular matrices and strictly lower triangular matrices, respectively. Then, the boundary condition is

$$\eta|_{\mathbb{R} \times \{r\}} \in \mathfrak{b}, \tag{185}$$

$$\sigma|_{\mathbb{R} \times \{r\}} \in \mathfrak{n}^*_- + \underline{\lambda}^\ell, \tag{186}$$

where $\mathfrak{b} = \mathfrak{h} \oplus \mathfrak{n}_+$ is the Borel subalgebra and $\underline{\lambda}^\ell$ is an element of $\mathfrak{h}^*$. This condition breaks the gauge group on the boundary to the Borel subgroup $B$ whose Lie algebra is $\mathfrak{b}$.[5]

By the state–operator correspondence, the Hilbert space is isomorphic to the space of observables supported at the junction of the above two coisotropic branes. This is a bimodule over the algebras of local observables on the two boundaries.

Let $\{T_\alpha\}_{\alpha=1}^{\dim \mathfrak{n}_-}$ be a basis of $\mathfrak{n}_-$ and extend it a basis $\{T_a\}_{a=1}^{\dim \mathfrak{g}}$ of $\mathfrak{g}$. Let $\sigma_a = \langle T_a, \sigma \rangle$. Classically, the algebra of local observables is the algebra of gauge-invariant polynomials in $\{\sigma_a\}$.

On the boundary $\mathbb{R} \times \{0\}$, local observables are simply polynomials in $\{\sigma_a\}$ since there is no gauge symmetry there. The algebra of local observables on the boundary is therefore $S(\mathfrak{g})$ at the classical level. Quantum corrections lead to a noncommutative deformation of $S(\mathfrak{g})$. At the quantum level, the algebra is isomorphic to the universal enveloping algebra of $\mathfrak{g}$ with bracket $\hbar[-,-]$; if $[T_a, T_b] = \sum_{c=1}^{\dim \mathfrak{g}} f_{ab}{}^c T_c$, then $[\sigma_a, \sigma_b] = \hbar \sum_{c=1}^{\dim \mathfrak{g}} f_{ab}{}^c \sigma_c$ quantum mechanically [50, 51]. In our setup $\hbar$ is a complex parameter rather than a formal parameter, so the algebra is isomorphic to $U(\mathfrak{g})$, with $\sigma_a$ mapped to $\hbar T_a$. (The quantization map from $S(\mathfrak{g})$ to $U(\mathfrak{g})$ is complicated for polynomials of higher degree.)

On the boundary $\mathbb{R} \times \{r\}$, the algebra of local observables is trivial. The boundary condition says that local operators are constructed entirely from $\{\sigma_\alpha\}$. Nontrivial polynomials cannot commute with $\mathfrak{h}$ and, in particular, cannot be $B$-invariant. The only local observables are multiples of the identity operator.

At the junction, the two boundary conditions combined imply that observables are polynomials in $\{\sigma_\alpha\}$. As a vector space, the space of observables is generated from the "highest-weight vector" 1 by the action of "creation operators" $\{\sigma_\alpha\}$. On this vector space the algebra $U(\mathfrak{g})$ acts. Thus, the Hilbert space is a Verma module of $U(\mathfrak{g})$, as expected.

It remains to show that the highest weight of the Verma module is determined by the positions of the D3-branes. Classically, the boundary condition on $\mathbb{R} \times \{r\}$ implies that the highest weight of the module is $\underline{\lambda}^\ell$. There is, however, a quantum correction which shifts the highest weight.[6] The highest weight is actually

$$\lambda^\ell = \underline{\lambda}^\ell - \rho \,, \tag{187}$$

where $\rho$ is the Weyl vector defined in terms of the character of the $\mathfrak{b}$-module $\mathfrak{g}/\mathfrak{b}$ as

$$\rho(-) = -\frac{1}{2} \operatorname{str}_{\mathfrak{g}/\mathfrak{b}} \operatorname{ad}(-) \,. \tag{188}$$

Since $\rho(\mathfrak{n}_+) = \rho([\mathfrak{b}, \mathfrak{b}]) = 0$, $\rho$ can be regarded as an element of $\mathfrak{h}^*$. This is the graded half sum of positive roots. When $\mathfrak{g}$ is an ordinary Lie algebra, $\rho$ is the ordinary Weyl vector and the above shift was derived in [52] based on results from [53, 54].

For $\mathfrak{g} = \mathfrak{gl}(m|n)$, we have

$$\rho = \sum_{\substack{k,l=1 \\ k<l}}^{m+n} \frac{1}{2}(-1)^{[k]+[l]}(\varepsilon_k - \varepsilon_l) = \sum_{i=1}^{m+n} \frac{1}{2}(-1)^{[i]}(m-n+(-1)^{[i]}-2c_i)\varepsilon_i \,. \tag{189}$$

---

[5]The choice of Borel subgroup is determined by the positions of the D5-branes in $\mathbb{R}_X$, which are in turn given by the vacuum expectation value of the time component $\mathcal{A}_x = A_x + iX$ of the gauge field; the $\mathfrak{u}(m|n)$-valued field $X$ has the vacuum expectation value $\langle X \rangle = \sum_{i=1}^{m+n} iX_i \mathcal{E}_{ii}$. In this background, a state evolving for duration $T$ is scaled by the factor $\exp(T\langle \mathcal{A}_x \rangle) = \exp(-T \sum_{i=1}^{m+n} X_i \mathcal{E}_{ii})$. (We have set $\langle A_x \rangle = 0$ for simplicity.) Therefore, if we compactify the time direction to a circle (say, of radius 1), as one does when computing the partition function of the lattice model equivalent to the spin chain, then the periodic boundary condition is twisted by the action of $\exp(-\sum_{i=1}^{m+n} X_i \mathcal{E}_{ii})$. In the magnon sector $(M_1, \ldots, M_{m+n-1})$, this action is multiplication by $\exp(-\sum_{\ell=1}^L \sum_{i=1}^{m+n} \lambda_i^\ell X_i + \sum_{r=1}^{m+n-1} M_r(X_r - X_{r+1}))$. We must have the ordering (107) for the partition function to be a power series in small variables.

[6]This shift can be understood as originating from normal ordering of creation and annihilation operators, which correspond to the positive and negative roots. We will see a similar shift in section 4.1.

Comparing the relations (187) and (306), we find

$$(-1)^{[i]}\left(\underline{\lambda}_i^\ell - \frac{1}{4}\right) = \frac{1}{\hbar}(\zeta^\ell - \mu_i^\ell) + \frac{1}{2}(m - n - c_i). \tag{190}$$

### 3.3.3 Line defects for parabolic Verma modules of scalar type

There is a generalization of the above brane construction which produces line defects in parabolic Verma modules of scalar type.

Let $(l_1, \ldots, l_s)$ be an ordered partition of $m + n$: $\sum_{\alpha=1}^s l_\alpha = m + n$. The partition specifies a parabolic subalgebra $\mathfrak{p}$ of $\mathfrak{gl}(m|n)$, namely the subalgebra consisting of upper-triangular block-diagonal matrices with diagonal blocks of orders $l_1$, $\ldots$, $l_s$. A character $\chi$ of $\mathfrak{p}$ is determined by an $s$-tuple of complex numbers $(\chi_1, \ldots, \chi_s)$ as

$$\chi(-) = \mathrm{str}(\chi^\vee -), \tag{191}$$

where the matrix $\chi^\vee$ is given by

$$\chi^\vee = \mathrm{diag}(\chi_1, \ldots, \chi_1, \ldots, \underbrace{\chi_\alpha, \ldots, \chi_\alpha}_{l_\alpha \text{ times}}, \ldots, \chi_s, \ldots, \chi_s). \tag{192}$$

For each $\alpha$, we take $l_\alpha$ D3-branes and make them end on a single NS5-brane on one side. On the other side, they end on separate D5-branes as in the previous construction. In total, we have $m + n$ D3-branes suspended between $m + n$ D5-branes and $s$ NS5-branes.

On the D3-branes we get the BF theory with gauge group GL($m|n$). The boundary condition on the D5-brane side is the same as before. On the NS5-brane side, the boundary condition is

$$\eta|_{\mathbb{R}\times\{r\}} \in \mathfrak{p}, \tag{193}$$

$$\sigma|_{\mathbb{R}\times\{r\}} \in \mathfrak{p}^\perp + \underline{\lambda}^\ell, \tag{194}$$

where $\mathfrak{p}^\perp$ is the annihilator of $\mathfrak{p}$ in $\mathfrak{g}^*$ and $\underline{\lambda}^\ell$ is a character of $\mathfrak{p}$. Classically (that is to say, when $\underline{\lambda}^\ell$ is of order $\hbar^{-1}$ and quantities of order $\hbar^0$ are ignored), the value of $\underline{\lambda}_\alpha^\ell$ is the position of the $\alpha$th NS5-brane in $\mathbb{C}$.

We expect that the Hilbert space of the BF theory with these boundary conditions is a parabolic Verma module of scalar type

$$U(\mathfrak{g}) \otimes_{U(\mathfrak{p})} \mathbb{C}_{\underline{\lambda}^\ell - \rho}. \tag{195}$$

The character $\rho$ of $\mathfrak{p}$ is defined by

$$\rho = -\frac{1}{2}\mathrm{str}_{\mathfrak{g}/\mathfrak{p}}\mathrm{ad}_\mathfrak{p} \tag{196}$$

and $\mathbb{C}_{\underline{\lambda}^\ell - \rho}$ is the one-dimensional $U(\mathfrak{p})$-module determined by the character $\underline{\lambda}^\ell - \rho$.

## 3.4 Fermionic Dualities

As we have emphasized in our discussions, the Lie superalgebra $\mathfrak{gl}(m|n)$ does not possess a unique Dynkin diagram. A Dynkin diagram is specified by a choice of ordered basis of $\mathbb{C}^{m|n}$ (or a choice of $\mathbb{Z}_2$-grading if we identify Dynkin diagrams related by the action of the Weyl group), and different choices are related by a series of certain adjacent transpositions, called odd reflections. Under odd reflections, a highest-weight representation is mapped to a highest-weight representation, but the highest weight is not preserved because the definition of raising and lowering operators is altered.

Odd reflections change how we describe representations of $\mathfrak{gl}(m|n)$, and the description of the Bethe vectors of the superspin chain is changed accordingly. The map from the Bethe vectors for one choice of ordered basis to another is known as a *fermionic duality*.

Fermionic dualities have been studied before purely from an algebraic perspective [55–60], with a notable exception of the work by Orlando and Reffert [25] where they employed the point of view of string theory to discuss the fermionic dualities for the supersymmetric *t-J* model, which is the rational $\mathfrak{gl}(1|2)$ spin chain with spins valued in the natural representation $\mathbb{C}^{1|2}$. Here we offer a string theory explanation for important aspects of fermionic dualities for the rational $\mathfrak{gl}(m|n)$ spin chain with Verma modules, namely their action on highest weights and magnon numbers.

### 3.4.1 Odd reflections and fermionic dualities

Recall from section 2.1 that the definitions of positive and simple roots depend on a choice of ordered basis $(e_1, \ldots, e_{m+n})$ of $\mathbb{C}^{m|n}$, which is a permutation of $(b_1, \ldots, b_m, f_1, \ldots, f_n)$ such that $(b_1, \ldots, b_m)$ and $(f_1, \ldots, f_n)$ are the standard bases of $\mathbb{C}^m$ and $\mathbb{C}^n$, respectively. There is a natural identification between these basis vectors and their weights:

$$(e_1, \ldots, e_{m+n}) \longleftrightarrow (\varepsilon_1, \ldots, \varepsilon_{m+n}), \tag{197}$$

$$(b_1, \ldots, b_m) \longleftrightarrow (\epsilon_1, \ldots, \epsilon_m), \tag{198}$$

$$(f_1, \ldots, f_n) \longleftrightarrow (\delta_1, \ldots, \delta_n). \tag{199}$$

In the following discussion we will consider permutations of $(\varepsilon_1, \ldots, \varepsilon_{m+n})$ induced by those of $(e_1, \ldots, e_{m+n})$.

For a given choice of ordered basis $(\varepsilon_1, \ldots, \varepsilon_{m+n})$ of the dual of the Cartan subalgebra of $\mathfrak{gl}(m|n)$, the set of positive roots is

$$\Phi^+ = \{\varepsilon_i - \varepsilon_j \mid i < j\} \tag{200}$$

and the set of simple roots is

$$\Pi = \{\varepsilon_r - \varepsilon_{r+1} \mid r = 1, \ldots, m+n-1\}. \tag{201}$$

A root $\varepsilon_i - \varepsilon_j$ is said to be *even* if $[i] = [j]$ and *odd* if $[i] \neq [j]$.

Pick an odd simple root $\alpha_s = \varepsilon_s - \varepsilon_{s+1}$ and apply to the ordered basis the adjacent transposition $\sigma_s \colon \{1, \ldots, m+n\} \to \{1, \ldots, m+n\}$ interchanging $\varepsilon_s$ and $\varepsilon_{s+1}$:

$$(\varepsilon_{\sigma_s(1)}, \ldots, \varepsilon_{\sigma_s(s-1)}, \varepsilon_{\sigma_s(s)}, \varepsilon_{\sigma_s(s+1)}, \varepsilon_{\sigma_s(s+2)}, \ldots, \varepsilon_{\sigma_s(m+n)})$$
$$= (\varepsilon_1, \ldots, \varepsilon_{s-1}, \varepsilon_{s+1}, \varepsilon_s, \varepsilon_{s+2}, \ldots, \varepsilon_{m+n}). \tag{202}$$

The adjacent transposition alters the notion of positive and simple roots. For the new ordered basis $(\varepsilon_{\sigma_s(1)}, \ldots, \varepsilon_{\sigma_s(m+n)})$, the set of positive roots is

$$\Phi_{\alpha_s}^+ = \{\varepsilon_{\sigma_s(i)} - \varepsilon_{\sigma_s(j)} \mid i < j\} = \{-\alpha_s\} \cup \Phi^+ \setminus \{\alpha_s\} \tag{203}$$

and the set of simple roots is

$$\Pi_{\alpha_s} = \{\varepsilon_{\sigma_s(r)} - \varepsilon_{\sigma_s(r+1)} \mid r = 1, \ldots, m+n-1\}$$
$$= \{\varepsilon_{s-1} - \varepsilon_{s+1}, \varepsilon_{s+1} - \varepsilon_s, \varepsilon_s - \varepsilon_{s+2}\} \cup \Pi \setminus \{\alpha_{s-1}, \alpha_s, \alpha_{s+1}\}. \tag{204}$$

This automorphism of the root system which transforms the positive and simple roots is called the *odd reflection* with respect to the odd simple root $\alpha_s$.

As an example, take $(m|n) = (3|2)$ and $(\varepsilon_1, \varepsilon_2, \varepsilon_3, \varepsilon_4, \varepsilon_5) = (\epsilon_1, \epsilon_2, \delta_1, \delta_2, \epsilon_3)$. This choice of ordered basis gives the Dynkin diagram (15). There are two odd simple roots, $\alpha_2 = \epsilon_2 - \delta_1$ and $\alpha_4 = \delta_2 - \epsilon_3$, represented by the crossed nodes. Reflection with respect to $\alpha_2$ swaps $\epsilon_2$ and $\delta_1$, leading to the new ordered basis $(\epsilon_1, \delta_1, \epsilon_2, \delta_2, \epsilon_3)$. The Dynkin diagram corresponding to the reflected simple roots is

$$\underset{\epsilon_1 - \delta_1}{\otimes}\rule[0.5ex]{1cm}{0.4pt}\underset{\delta_1 - \epsilon_2}{\otimes}\rule[0.5ex]{1cm}{0.4pt}\underset{\epsilon_2 - \delta_2}{\otimes}\rule[0.5ex]{1cm}{0.4pt}\underset{\delta_2 - \epsilon_3}{\otimes}\,. \tag{205}$$

We see that all simple roots are now odd. Reflection of the original ordered basis with respect to $\alpha_4$ results in the ordered basis $(\epsilon_1, \epsilon_2, \delta_1, \epsilon_3, \delta_2)$ and the Dynkin diagram

$$\underset{\epsilon_1 - \epsilon_2}{\bigcirc}\rule[0.5ex]{1cm}{0.4pt}\underset{\epsilon_2 - \delta_1}{\otimes}\rule[0.5ex]{1cm}{0.4pt}\underset{\delta_1 - \epsilon_3}{\otimes}\rule[0.5ex]{1cm}{0.4pt}\underset{\epsilon_3 - \delta_2}{\otimes}\,. \tag{206}$$

Odd reflections change the characterization of highest weights. Let us see how Verma modules are transformed. Fix an ordered basis of $\mathbb{C}^{m|n}$ and consider the Verma module $M(\lambda)$, with the highest-weight vector $|\Omega_\lambda\rangle$. Let $\alpha_s$ be an odd root. After the odd reflection about $\alpha_s$, the roles of the raising operator $\mathcal{E}_{s,s+1}$ and the lowering operator $\mathcal{E}_{s+1,s}$ are exchanged, while all other lowering operators remain unchanged. Consequently, the state

$$|\Omega_{\lambda'}\rangle = \mathcal{E}_{s+1,s}|\Omega_\lambda\rangle \tag{207}$$

is annihilated by all elements of the new set of raising operators, that is, it is a highest-weight state with respect to the new ordered basis. According to the PBW theorem (23), the states of the form

$$x_1^{n_1} \cdots x_{p-1}^{n_{p-1}} \mathcal{E}_{s+1,s}^{n_p} |\Omega_\lambda\rangle = x_1^{n_1} \cdots x_{p-1}^{n_{p-1}} \mathcal{E}_{s,s+1}^{1-n_p} |\Omega_{\lambda'}\rangle\,, \tag{208}$$

form a basis of the Fock space $V_\lambda$ for $M(\lambda)$, where $(x_1, \ldots, x_{p-1}, \mathcal{E}_{s+1,s})$ is an ordered set of lowering operators in the original ordered basis. (Note that $n_p$ is either 0 or 1.) By the PBW theorem, we see that $M(\lambda)$ is the Verma module $M(\lambda')$ with respect to the new ordered basis, with

$$\lambda' = \lambda - \alpha_s\,. \tag{209}$$

In the spin chain, the highest weights of the Verma modules placed at the spin sites are transformed by an odd reflection. The weight of each state of the spin chain remains the same, so the magnon numbers must be transformed as

$$\sum_{\ell=1}^{L} (\lambda^\ell)' - \sum_{r=1}^{m+n-1} M_r' \alpha_r' = \sum_{\ell=1}^{L} \lambda^\ell - \sum_{r=1}^{m+n-1} M_r \alpha_r\,, \quad \alpha_r' = \varepsilon_{\sigma_s(r)} - \varepsilon_{\sigma_s(r+1)}\,, \tag{210}$$

or more explicitly,

$$M_r' = \begin{cases} L + M_{s-1} + M_{s+1} - M_s & (r = s)\,; \\ M_r & (r \neq s)\,. \end{cases} \tag{211}$$

The transformations of the highest weights and magnon numbers change the Bethe equations. Of course, this is merely a change in the description of the spin chain states, so the solutions of the new Bethe equations are in one-to-one correspondence with the solutions of the original Bethe equations. This correspondence is called the fermionic duality generated by the odd reflection.

### 3.4.2 Fermionic duality from string theory

In the D1–D3–NS5 duality frame that is S-dual to (114), the choice of ordered basis of $\mathbb{C}^{m|n}$ is reflected in the ordering of NS5-branes along $\mathbb{R}_X$; they are ordered as $\mathrm{NS5}_1, \mathrm{NS5}_2, \ldots, \mathrm{NS5}_{m+n}$ from left to right. Therefore, the string theory interpretation of the reflection about an odd root $\alpha_s$ is clear: it swaps the positions of $\mathrm{NS5}_s$ and $\mathrm{NS5}_{s+1}$. We wish to understand the effect of the exchange of positions on the highest weights and magnon numbers.

Recall that one of the boundary conditions for the BF theory that emerges on the D3-branes creating the line defect $\mathcal{L}^\ell$ is specified by the parameters $\underline{\lambda}_i^\ell$, $i = 1, \ldots, m + n$. The position of $\mathrm{D3}_i^\ell$ in $\mathbb{C}$ is given by

$$\zeta^\ell - (-1)^{[i]} \underline{\lambda}_i^\ell. \tag{212}$$

Indeed, if we take $\underline{\lambda}_i^\ell$ to be of order $\hbar^{-1}$, by the relation (190) this quantity coincides to order $\hbar^0$ with the twisted mass $\mu_i^\ell$, which is identified with the classical location of $\mathrm{D3}_i^\ell$. Swapping the positions of $\mathrm{NS5}_s$ and $\mathrm{NS5}_{s+1}$ also exchanges $\mathrm{D3}_s$ and $\mathrm{NS5}_{s+1}$ while keeping their locations in $\mathbb{C}$ fixed. Thus, $\{\underline{\lambda}_i^\ell\}$ are transformed to new values $\{(\underline{\lambda}_i^\ell)'\}$ such that $(\underline{\lambda}_s^\ell)' = \underline{\lambda}_{s+1}^\ell$ and $(\underline{\lambda}_{s+1}^\ell)' = \underline{\lambda}_s^\ell$ in the new ordered basis. This simply means that we have

$$(\underline{\lambda}^\ell)' = \sum_{i=1}^{m+n} (\underline{\lambda}^\ell)'_i \varepsilon_{\sigma_s(i)} = \underline{\lambda}^\ell. \tag{213}$$

Although $\underline{\lambda}^\ell$ is invariant under the odd reflection, the Weyl vector $\rho$ is transformed to a new Weyl vector $\rho'$. Since $\rho$ is the half sum of even positive roots minus the half sum of odd positive roots, from the relation (203) between $\Phi^+$ and $\Phi_{\alpha_s}^+$ we see

$$\rho' = \rho + \alpha_s. \tag{214}$$

Then, the relation (187) between $\underline{\lambda}^\ell$ and the highest weight $\lambda^\ell$ shows that $\lambda^\ell$ is transformed to $(\lambda^\ell)'$ according to the formula (209).

Exchanging the pairs $(\mathrm{NS5}_s, \mathrm{D3}_s)$ and $(\mathrm{NS5}_{s+1}, \mathrm{D3}_{s+1})$ does not only transform the highest weights, but also change the magnon numbers. In the brane picture, we can understand this phenomenon as creation and annihilation of D1-branes due to the Hanany–Witten effect [61].[7] In order to exchange the positions of the brane pairs, we first need to move each D1-brane between $\mathrm{NS5}_s$ and $\mathrm{NS5}_{s+1}$ so that one of its end is attached to one of the NS5-branes, say $\mathrm{NS5}_{s+1}$. Then, we displace $\mathrm{NS5}_{s+1}$ into the page and start moving it to the left. At one point the D3-branes ending on $\mathrm{NS5}_{s+1}$ pass through $\mathrm{NS5}_s$. As a result, the D1-branes ending on these D3-branes are annihilated and a new D1-brane is created on each of those D3-branes that did not have D1-brane ending on it. In the case in which $[s] = \bar{0}$, $[s + 1] = \bar{1}$ and $(M_{s-1}, M_s, M_{s+1}) = (1, 2, 1)$, the process looks as follows:

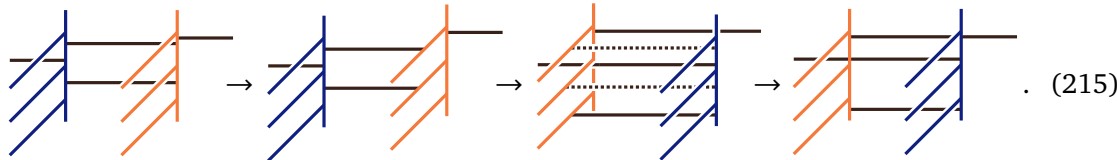

$$. \tag{215}$$

The dotted lines indicate the annihilation of D1-branes. We see that the numbers of D1-branes between NS5-branes transform as in the formula (211).

---

[7]In general, Hanany–Witten processes for type IIA brane configurations for two-dimensional $\mathcal{N} = (2,2)$ supersymmetric gauge theories classically suffer from ambiguities, which are only resolved if one takes brane bending into account or lifts the configurations to M-theory. [33]. In the present case, such ambiguities do not arise because the relevant gauge group has the same number of fundamental and antifundamental chiral multiplets.

In the D2–D4–NS5 duality frame, the above manipulation is expected to lead to an infrared duality of $\mathcal{N} = (2,2)$ supersymmetric gauge theories. Indeed, there is a known duality transformation that sends a theory with $U(N_c)$ gauge group, $N_f$ fundamental chiral multiplets and $N_a$ antifundamental chiral multiplets to a theory with $U(N_c')$ gauge group, $N_a$ fundamental chiral multiplets and $N_f$ antifundamental chiral multiplets, plus mesons transforming in the bifundamental representation of the flavor group $U(N_a) \times U(N_f)$ [62]. The rank of the dual gauge group is $N_c' = \max(N_f, N_a) - N_c$. This is consistent with what we have found since $N_c = M_s$ and $N_f = N_a = L + M_{s-1} + M_{s+1}$ in our case. However, it appears that the mesons are absent from our final brane configuration. Fortunately, the mesons, being neutral under the gauge symmetry, do not affect the Bethe equations.

# 4 Bethe/gauge correspondence for compact superspin chains

The superspin chains that appear in the Bethe/gauge correspondence discussed in the previous sections are noncompact, meaning that they carry spins valued in infinite-dimensional representations of the Yangian $Y(\mathfrak{gl}(m|n))$. Spin chains whose spins are valued in finite-dimensional representations are said to be *compact*.

In this section we discuss the Bethe/gauge correspondence for compact rational $\mathfrak{gl}(m|n)$ spin chains. We will follow a line of reasoning similar to our treatment of the noncompact case, but in the reverse direction: we start with the construction of line defects for finite-dimensional representations in four-dimensional Chern–Simons theory, then identify their brane realization and apply dualities to deduce the corresponding two-dimensional quiver gauge theories.

## 4.1 Covariant and contravariant representations of $\mathfrak{gl}(m|n)$

Finite-dimensional representations of $\mathfrak{gl}(m|n)$ are most easily discussed in the *distinguished grading*, in which

$$[i] = \begin{cases} \bar{0} & (i \leq m); \\ \bar{1} & (i > m). \end{cases} \tag{216}$$

For this reason, in this section we exclusively use the distinguished grading. We will write a weight $\lambda = \sum_{i=1}^{m+n} \lambda_i \varepsilon_i$ as $(\lambda_1, \ldots, \lambda_m | \lambda_{m+1}, \ldots, \lambda_{m+n})$.

The Verma module $M(\lambda)$ with highest weight $\lambda$ contains a unique maximal submodule. In the distinguished grading, the corresponding simple quotient module $L(\lambda)$ is finite-dimensional if and only if

$$\lambda_i - \lambda_{i+1} \in \mathbb{Z}_{\geq 0}, \quad i = 1, \ldots, \ldots, m+n-1, \quad i \neq m, \tag{217}$$

in other words, if and only if $(\lambda_1, \ldots, \lambda_m)$ and $(\lambda_{m+1}, \ldots, \lambda_{m+n})$ are highest weights of finite-dimensional irreducible representations of $\mathfrak{gl}(m)$ and $\mathfrak{gl}(n)$, respectively. Any finite-dimensional irreducible representation of $\mathfrak{gl}(m|n)$ is isomorphic to $L(\lambda)$ for some $\lambda$.

We will consider two classes of finite-dimensional irreducible representations of $\mathfrak{gl}(m|n)$, called covariant representations and contravariant representations. Covariant representations appear in tensor products of copies of the natural representation

$$\mathbb{C}^{m|n} = L\big((1, 0, \ldots, 0 | 0, \ldots, 0)\big), \tag{218}$$

whereas contravariant representations arise from tensor products of copies of the dual repre-

sentation[8]

$$(\mathbb{C}^{m|n})^* = L\big((0,\ldots,0|0,\ldots,0,-1)\big). \tag{219}$$

Both covariant and contravariant representations are indexed by the so-called $(m|n)$-hook partitions.

A partition $Y = (Y_1, \ldots, Y_{l(Y)})$ of size $|Y|$ and length $l(Y)$ is an $l(Y)$-tuple of positive integers such that $Y_1 \geq \cdots \geq Y_{l(Y)}$ and $Y_1 + \cdots + Y_{l(Y)} = |Y|$. It can be represented by a Young diagram with $l(Y)$ rows, with the $\alpha$th row consisting of $Y_\alpha$ boxes. The conjugate partition $Y'$ has the Young diagram that is the transpose of the Young diagram for $Y$.

A partition $Y$ is said to be $(m|n)$-*hook* if $Y_{m+1} \leq n$. If $Y$ is an $(m|n)$-hook partition, then $Y'$ is an $(n|m)$-hook partition, $Y'_{n+1} \leq m$. We let $\mathcal{H}_{m|n}$ denote the set of all $(m|n)$-hook partitions.

For an $(m|n)$-hook partition $Y$, we define the integral weight

$$Y^\natural = (Y_1, \ldots, Y_m | \langle Y'_1 - m \rangle, \ldots, \langle Y'_n - m \rangle), \tag{220}$$

where $\langle a \rangle = \max(0, a)$. The even part of $Y^\natural$ is represented by the Young diagram formed by the first $m$ rows of $Y$. The Young diagram for the odd part of $Y^\natural$ is the transpose of the remainder of $Y$, and its length is less than or equal to $n$ by the $(m|n)$-hook condition.

Let $Y$ be an $(m|n)$-hook partition. The *covariant representation labeled by $Y$* is the highest-weight representation $L(Y^\natural)$. The *contravariant representation labeled by $Y$* is the dual representation $L(Y^\natural)^* = L(\widetilde{Y}^\natural)$. Its highest weight $\widetilde{Y}^\natural$ equals the minus of the lowest weight of $L(Y^\natural)$ and is given by

$$\widetilde{Y}^\natural = (-\langle Y_m - n \rangle, \ldots, -\langle Y_1 - n \rangle | -Y'_n, \ldots, -Y'_1). \tag{221}$$

## 4.2 Line defects in covariant and contravariant representations

Now we construct quantum mechanical systems whose Hilbert spaces are covariant and contravariant representations of $\mathfrak{gl}(m|n)$. Coupled to four-dimensional Chern–Simons theory with gauge group $\mathrm{GL}(m|n)$, they describe line defects valued in these finite-dimensional irreducible representations.

Let $K, \overline{K}$ be nonnegative integers and consider a pair of fields

$$\varphi \in \mathrm{Hom}(\mathbb{C}^{K|\overline{K}}, \mathbb{C}^{m|n}), \tag{222}$$

$$\tilde{\varphi} \in \mathrm{Hom}(\mathbb{C}^{m|n}, \mathbb{C}^{K|\overline{K}}) \tag{223}$$

transforming in the bifundamental representations of $\mathrm{GL}(m|n) \times \mathrm{GL}(K|\overline{K})$. Their components are $\mathbb{Z}_2$-graded, with the grading given by

$$[\varphi^i_\alpha] = [\tilde{\varphi}^\alpha_i] = [i] + [\alpha], \tag{224}$$

where $i$ and $\alpha$ are indices for $\mathbb{C}^{m|n}$ and $\mathbb{C}^{K|\overline{K}}$, respectively. The even components are bosonic and the odd ones are fermionic. The action of the theory is

$$\frac{1}{\hbar} \int_{\mathbb{R}} \mathrm{str}_{\mathbb{C}^{m|n}}(\varphi \, \mathrm{d}\tilde{\varphi}). \tag{225}$$

---

[8] The dual $\pi^*$ of a representation $\pi$ is given by $\pi^* = \tau \circ \pi$, where $\tau(X) = -X^{\mathrm{st}}$ is the Chevalley automorphism. The supertranspose $X^{\mathrm{st}}$ of $X$ is defined by $X^{\mathrm{st}}_{ij} = (-1)^{([i]+[j])[j]} X_{ji}$. Our definition of supertranspose differs from a commonly used definition by a factor of $(-1)^{[i]+[j]}$. With this definition, the quantum mechanical action (225) is invariant under the natural action of $\mathrm{GL}(m|n) \times \mathrm{GL}(K|\overline{K})$.

We will find it convenient to define

$$\eta_\alpha^i = \frac{i}{\hbar}\varphi_\alpha^i, \tag{226}$$

$$\chi_i^\alpha = (-1)^{[\alpha]}\tilde{\varphi}_i^\alpha, \tag{227}$$

and

$$\tilde{\chi}_\alpha^i = (-1)^{[i][\alpha]+[\alpha]}\eta_\alpha^i, \tag{228}$$

$$\tilde{\eta}_i^\alpha = (-1)^{[i][\alpha]+[\alpha]}\frac{i}{\hbar}\chi_i^\alpha. \tag{229}$$

Then, the canonical commutation relations read

$$[\eta_\alpha^i, \chi_j^\beta] = [\tilde{\chi}_\alpha^i, \tilde{\eta}_j^\beta] = \delta_\alpha^\beta \delta_j^i. \tag{230}$$

The theory has a $GL(m|n) \times GL(K|\overline{K})$ global symmetry. The associated conserved charges are

$$q_{ij} = \sum_{\alpha=1}^{K+\overline{K}} \chi_i^\alpha \eta_\alpha^j + (-1)^{[i]} c \delta_{ij}, \tag{231}$$

$$Q_{\alpha\beta} = -\sum_{i=1}^{m+n} (-1)^{([\alpha]+[\beta])[\alpha]} \chi_i^\beta \eta_\alpha^i + (-1)^{[\alpha]} C \delta_{\alpha\beta} \tag{232}$$

and satisfy the $\mathfrak{gl}(m|n) \oplus \mathfrak{gl}(K|\overline{K})$ commutation relations:

$$[q_{ij}, q_{kl}] = \delta_{jk} q_{il} - (-1)^{([i]+[j])([k]+[l])} \delta_{li} q_{kj}, \tag{233}$$

$$[Q_{\alpha\beta}, Q_{\gamma\delta}] = \delta_{\beta\gamma} Q_{\alpha\delta} - (-1)^{([\alpha]+[\beta])([\gamma]+[\delta])} \delta_{\delta\alpha} Q_{\gamma\beta}, \tag{234}$$

$$[q_{ij}, Q_{\alpha\beta}] = 0. \tag{235}$$

The constants $c$ and $C$ account for the ambiguity in operator ordering and will be fixed in a moment. Under $GL(m|n) \times GL(K|\overline{K})$, the sets of fields $\{\chi_i^\alpha\}$ and $\{\tilde{\chi}_\alpha^i\}$ transform as the standard basis vectors for $\mathbb{C}^{m|n} \otimes (\mathbb{C}^{K|\overline{K}})^*$ and $(\mathbb{C}^{m|n})^* \otimes \mathbb{C}^{K|\overline{K}}$, respectively:

$$[q_{ij}, \chi_k^\alpha] = \delta_{jk} \chi_i^\alpha = \sum_{l=1}^{m+n} (E_{ij})_{lk} \chi_l^\alpha, \tag{236}$$

$$[Q_{\alpha\beta}, \chi_i^\gamma] = -(-1)^{([\alpha]+[\beta])[\alpha]} \delta_{\alpha\gamma} \chi_i^\beta = \sum_{\delta=1}^{K+\overline{K}} (-E_{\alpha\beta}^{\text{st}})_{\delta\gamma} \chi_i^\delta. \tag{237}$$

$$[q_{ij}, \tilde{\chi}_\alpha^k] = -(-1)^{([i]+[j])[i]} \delta_{ki} \tilde{\chi}_\alpha^j = \sum_{l=1}^{m+n} (-E_{ij}^{\text{st}})_{lk} \tilde{\chi}_\alpha^l, \tag{238}$$

$$[Q_{\alpha\beta}, \tilde{\chi}_\gamma^i] = \delta_{\beta\gamma} \tilde{\chi}_\alpha^i = \sum_{\delta=1}^{K+\overline{K}} (E_{\alpha\beta})_{\delta\gamma} \tilde{\chi}_\delta^i. \tag{239}$$

For the construction of line defects we actually break the $GL(K|\overline{K})$ symmetry. Let us gauge the Borel subgroup of $GL(K|\overline{K})$. We introduce an associated gauge field

$$\mathcal{B} = \sum_{\substack{\alpha,\beta=1 \\ \alpha \leq \beta}}^{K+\overline{K}} \mathcal{B}^{\alpha\beta} E_{\alpha\beta} \tag{240}$$

and couple it to the theory by replacing the de Rham differential d with $d + \mathcal{B}$. For the moment we treat $\mathcal{B}$ as a background field and give it a diagonal value

$$\mathcal{B} = \text{diag}(b_1, \ldots, b_{K+\overline{K}}). \tag{241}$$

In this background, the action becomes

$$\frac{1}{\hbar} \int_{\mathbb{R}} \sum_{i=1}^{m+n} \sum_{\alpha=1}^{K+\overline{K}} (-1)^{[i]} (\varphi_\alpha^i d\tilde{\varphi}_i^\alpha + b_\alpha \varphi_\alpha^i \tilde{\varphi}_i^\alpha), \tag{242}$$

and the $\text{GL}(K|\overline{K})$ symmetry is broken to the stabilizer of the gauge field, which is generically the maximal torus.

The coupling to the gauge field does not affect the canonical commutation relations, but modifies the Hamiltonian. Before the introduction of the gauge field, the theory was topological and the Hamiltonian was zero. It is now given by

$$H = i\hbar \sum_{\alpha=1}^{K+\overline{K}} b_\alpha Q_{\alpha\alpha}. \tag{243}$$

The Hilbert space of the theory is a $\mathbb{Z}_2$-graded Fock space constructed from a vacuum state $|0\rangle$ by the action of the creation operators. The action of $\varphi_\alpha^i$ changes $H$ by $i\hbar b_\alpha$ while $\tilde{\varphi}_i^\alpha$ changes $H$ by $-i\hbar b_\alpha$. Those component fields that increase $\text{Re}(iH/\hbar)$ are creation operators, and those that decrease it are annihilation operators.[9] We can think of $\text{Re}(iH/\hbar)$ as energy.

Suppose that we give the background value such that

$$0 < \text{Re}\, b_{K+\overline{K}} < \text{Re}\, b_{K+\overline{K}-1} < \cdots < \text{Re}\, b_1. \tag{244}$$

Then, $\chi_i^\alpha$ is a creation operator and $\eta_\alpha^i$ is an annihilation operator. Requiring the vacuum to be invariant under (the maximal torus of) $\text{GL}(K|\overline{K})$, we find $c = C = 0$. Let $\mathcal{F}$ be the corresponding Fock space.

The Fock space $\mathcal{F}$ decomposes into tensor products of covariant representations of $\mathfrak{gl}(m|n)$ and contravariant representations of $\mathfrak{gl}(K|\overline{K})$ [63, 64]:

$$\mathcal{F} = \bigoplus_{Y \in \mathcal{H}_{m|n} \cap \mathcal{H}_{K|\overline{K}}} L(Y_{m|n}^\natural) \otimes L(Y_{K|\overline{K}}^\natural)^*. \tag{245}$$

(We use subscripts to distinguish weights for $\mathfrak{gl}(m|n)$ and $\mathfrak{gl}(K|\overline{K})$.) For example, the first excited states take the form

$$\sum_{i=1}^{m+n} \sum_{\alpha=1}^{K+\overline{K}} c_\alpha^i \chi_i^\alpha |0\rangle \tag{246}$$

and span a subspace isomorphic to

$$\mathbb{C}^{m|n} \otimes (\mathbb{C}^{K|\overline{K}})^*, \tag{247}$$

as can be seen from the commutation relations (236) and (237).

This Hilbert space is too large, and we need to reduce it to a single covariant representation $L(Y_{m|n}^\natural)$ of $\mathfrak{gl}(m|n)$. To do so, we impose constraints that singles out the summand $L(Y_{m|n}^\natural) \otimes L(Y_{K|\overline{K}}^\natural)^*$ and further projects it to the subspace of lowest-energy states. Since the

---

[9]If the time axis is compactified to a circle of radius 1, then the partition function involves trace twisted by $\exp(-iH/\hbar)$. Creation operators should make this factor smaller.

raising operator $Q_{\alpha\beta}$, $\alpha < \beta$, changes the energy $\mathrm{Re}(\mathrm{i}H/\hbar)$ by $-\mathrm{Re}\,b_\alpha + \mathrm{Re}\,b_\beta < 0$, the lowest-energy states have the highest weight with respect to $\mathfrak{gl}(K|\overline{K})$.

We implement this projection by making $\mathcal{B}$ dynamical. The vacuum expectation value of $\mathcal{B}$ is given by the diagonal matrix (241). Let us add to the action the Chern–Simons term

$$-\mathrm{i}\int_{\mathbb{R}} \widetilde{\underline{Y}}^{\natural}_{K|\overline{K}}(\mathcal{B}). \tag{248}$$

Then, the equations of motion for $\mathcal{B}$ are

$$Q_{\alpha\beta} = 0, \quad \alpha < \beta \tag{249}$$

and[10]

$$Q_{\alpha\alpha} = (\widetilde{\underline{Y}}^{\natural}_{K|\overline{K}})_\alpha + \frac{m-n}{2}. \tag{250}$$

The former equations restrict the Fock space to the subspace of states that contains highest-weight vectors of covariant representations of $\mathfrak{gl}(K|\overline{K})$:

$$\bigoplus_{Y \in \mathcal{H}_{m|n} \cap \mathcal{H}_{K|\overline{K}}} L(Y^{\natural}_{m|n}) \otimes |\Omega_{\widetilde{Y}^{\natural}_{K|\overline{K}}}\rangle. \tag{251}$$

With the choice

$$\widetilde{Y}^{\natural}_{K|\overline{K}} = \widetilde{\underline{Y}}^{\natural}_{K|\overline{K}} - \frac{m-n}{2}\sum_{\alpha=1}^{K+\overline{K}} \varepsilon_\alpha, \tag{252}$$

the second equation selects the highest weight $\widetilde{Y}^{\natural}_{K|\overline{K}}$, thereby reducing the Hilbert space to the covariant representation $L(Y^{\natural}_{m|n}) \otimes |\Omega_{\widetilde{Y}^{\natural}_{K|\overline{K}}}\rangle$ of $\mathfrak{gl}(m|n)$.

In order to construct line defects in contravariant representations, we take

$$\mathrm{Re}\,b_{K+\overline{K}} < \mathrm{Re}\,b_{K+\overline{K}-1} < \cdots < \mathrm{Re}\,b_1 < 0. \tag{253}$$

In this case, $\tilde{\chi}^\alpha_i$ is a creation operator and $\tilde{\eta}^i_\alpha$ is an annihilation operator. The corresponding Fock space $\widetilde{\mathcal{F}}$ decomposes as

$$\widetilde{\mathcal{F}} = \bigoplus_{Y \in \mathcal{H}_{m|n} \cap \mathcal{H}_{K|\overline{K}}} L(Y^{\natural}_{m|n})^* \otimes L(Y^{\natural}_{K|\overline{K}}). \tag{254}$$

Making $\mathcal{B}$ dynamical and adding the Chern–Simons term

$$-\mathrm{i}\int_{\mathbb{R}} \underline{Y}^{\natural}_{K|\overline{K}}(\mathcal{B}) \tag{255}$$

to the action, we can reduce the Hilbert space to the contravariant representation $L(Y^{\natural})^*$, with

$$Y^{\natural}_{K|\overline{K}} = \underline{Y}^{\natural}_{K|\overline{K}} - \frac{m-n}{2}\sum_{\alpha=1}^{K+\overline{K}} \varepsilon_\alpha. \tag{256}$$

---

[10]The Weyl quantization of the classical expression of $Q_{\alpha\alpha}$ equals $Q_{\alpha\alpha} + (-1)^{[\alpha]}(m-n)/2$.

### 4.3 Brane construction of line defects

The quantum mechanical system discussed above can be constructed with D3-branes and D5-branes. Let us remove from the brane system (114) the semi-infinite D3-branes and the F1-branes stretched between the D5-branes, and instead introduce infinite D3-branes D3$_\alpha$, $\alpha = 1$, $\ldots, K + \overline{K}$:

$$
\begin{aligned}
\text{Spacetime:} \quad & \mathbb{R} \times S^1 \times \mathbb{C} \times & \mathbb{R}_X & \times \mathbb{R}_Y \times \mathbb{R}^2_{+\hbar} \times \mathbb{R}^2_{-\hbar}, \\
\text{D5}_i \ (i \le m): \quad & \mathbb{R} \times S^1 \times \mathbb{C} \times & \{X_i\} & \times \{Y_i\} \times \mathbb{R}^2_{+\hbar} \times \{0\}, \\
\text{D5}_i \ (i > m): \quad & \mathbb{R} \times S^1 \times \mathbb{C} \times & \{X_i\} & \times \{Y_i\} \times \{0\} \times \mathbb{R}^2_{-\hbar}, \\
\text{D3}_\alpha \ (\alpha \le K): \quad & \mathbb{R} \times \{y\} \times \{\zeta\} \times & \{-\operatorname{Re} b_\alpha\} & \times \mathbb{R}_Y \times \mathbb{R}^2_{+\hbar} \times \{0\}, \\
\text{D3}_\alpha \ (\alpha > K): \quad & \mathbb{R} \times \{y\} \times \{\zeta\} \times & \{-\operatorname{Re} b_\alpha\} & \times \mathbb{R}_Y \times \{0\} \times \mathbb{R}^2_{-\hbar}.
\end{aligned}
\tag{257}
$$

We claim that strings stretched between the D3-branes and the D5-branes give rise to the quantum mechanical system in question.

The $K$ D3-branes D3$_\alpha$, $\alpha \le K$, and the $m$ D5-branes D5$_i$, $i \le m$, share the three-dimensional spacetime $\mathbb{R} \times \mathbb{R}^2_{+\hbar}$. Strings stretched between them produce an $\mathcal{N} = 4$ hypermultiplet in the bifundamental representation of $\mathrm{U}(K) \times \mathrm{U}(m)$. Let

$$
\varphi^{00} \in \operatorname{Hom}(\mathbb{C}^K, \mathbb{C}^m), \tag{258}
$$

$$
\tilde{\varphi}^{00} \in \operatorname{Hom}(\mathbb{C}^m, \mathbb{C}^K) \tag{259}
$$

be the scalar fields of this multiplet. We are looking at the sector of this theory that is invariant under the supercharge $Q$ for the holomorphic–topological twist. There is an $\Omega$-deformation induced by the background RR two-form, and it has the effect of localizing the hypermultiplet to the quantum mechanical model with action [23]

$$
\frac{1}{\hbar} \int_{\mathbb{R}} \operatorname{tr}_{\mathbb{C}^m}(\varphi^{00} \, \mathrm{d}\tilde{\varphi}^{00}). \tag{260}
$$

Here we are using $\varphi^{00}$, $\tilde{\varphi}^{00}$ to denote the one-dimensional fields that descend from the three-dimensional scalar fields.

Similarly, from strings stretched between the $\overline{K}$ D3-branes D3$_\alpha$, $\alpha > K$, and the $n$ D5-branes D5$_i$, $i > m$, we get an $\mathcal{N} = 4$ hypermultiplet in the bifundamental representation of $\mathrm{U}(\overline{K}) \times \mathrm{U}(n)$ on the three-dimensional spacetime $\mathbb{R} \times \mathbb{R}^2_{-\hbar}$. By an $\Omega$-deformation, the theory localizes to a quantum mechanical system with action

$$
-\frac{1}{\hbar} \int_{\mathbb{R}} \operatorname{tr}_{\mathbb{C}^n}(\varphi^{11} \, \mathrm{d}\tilde{\varphi}^{11}), \tag{261}
$$

where

$$
\varphi^{11} \in \operatorname{Hom}(\mathbb{C}^{\overline{K}}, \mathbb{C}^n), \tag{262}
$$

$$
\tilde{\varphi}^{11} \in \operatorname{Hom}(\mathbb{C}^n, \mathbb{C}^{\overline{K}}). \tag{263}
$$

The branes D3$_\alpha$, $\alpha \le K$, and D5$_i$, $i > m$, intersect along the time axis $\mathbb{R}$, and from strings stretched between them we get fermions

$$
\varphi^{10} \in \operatorname{Hom}(\mathbb{C}^K, \mathbb{C}^n), \tag{264}
$$

$$
\tilde{\varphi}^{10} \in \operatorname{Hom}(\mathbb{C}^n, \mathbb{C}^K). \tag{265}
$$

They are described by the action

$$
\frac{1}{\hbar} \int_{\mathbb{R}} \operatorname{tr}_{\mathbb{C}^n}(\varphi^{10} \, \mathrm{d}\tilde{\varphi}^{10}). \tag{266}
$$

This is the dimensional reduction of the two-dimensional chiral fermions that arise from an intersection of D4-branes and D6-branes [65, 66].

In the same way, from strings stretched between $D3_\alpha$, $\alpha > K$, and $D5_i$, $i \leq m$, we get fermionic fields

$$\varphi^{01} \in \mathrm{Hom}(\mathbb{C}^{\overline{K}}, \mathbb{C}^m), \tag{267}$$

$$\tilde{\varphi}^{01} \in \mathrm{Hom}(\mathbb{C}^m, \mathbb{C}^{\overline{K}}), \tag{268}$$

described by the action

$$-\frac{1}{\hbar} \int_{\mathbb{R}} \mathrm{tr}_{\mathbb{C}^m}(\varphi^{01} \, \mathrm{d}\tilde{\varphi}^{01}). \tag{269}$$

These four quantum mechanical systems can be combined into the single quantum mechanical system described by the action (225), with the fields

$$\varphi = \begin{pmatrix} \varphi^{00} & \varphi^{01} \\ \varphi^{10} & \varphi^{11} \end{pmatrix}, \tag{270}$$

$$\tilde{\varphi} = \begin{pmatrix} \tilde{\varphi}^{00} & \tilde{\varphi}^{01} \\ \tilde{\varphi}^{10} & \tilde{\varphi}^{11} \end{pmatrix}. \tag{271}$$

The creation operator $\chi_i^\alpha$ adds a string stretched between $D3_\alpha$ and $D5_i$. The annihilation operator $\eta_\alpha^i$ removes a string between them.

This quantum mechanical system is coupled to the four-dimensional Chern–Simons theory that arises from the D5-branes and to the BF theory that arises on the D3-branes. As in the construction of line defects valued in Verma modules, the boundary conditions on the BF theory (at infinity, or at finite distance if we make the D3-branes end on NS5-branes) breaks the $GL(K|\overline{K})$ gauge symmetry to a Borel subgroup. Which Borel subgroup is selected is determined by the ordering of the D3-branes on $\mathbb{R}_X$. For the ordering (244) for $\{\mathrm{Re}\, b_\alpha\}$, it is the standard Borel subgroup.

The situation in which there are no strings stretched between the D3-branes and the D5-branes corresponds to the vacuum of the Fock space $\mathcal{F}$. Here is how the vacuum looks like for $(m|n) = (1|2)$ and $(K|\overline{K}) = (2|1)$:

$$|0\rangle = \qquad\qquad\qquad\qquad\qquad\qquad . \tag{272}$$

To project to a covariant representation $L(Y_{m|n}^\natural)$ of $\mathfrak{gl}(m|n)$, we fix the number of strings ending on each D3-brane. Let us illustrate how this works with an example in which $Y = (5,1,1)$. For this choice of $Y$, we have $Y' = (3,1,1,1,1)$, $Y_{m|n}^\natural = (5|2,0)$ and $\widetilde{Y}_{K|\overline{K}}^\natural = (0,-4|-3)$. The brane configuration for the highest-weight state of $L(Y_{m|n}^\natural) \otimes L(\widetilde{Y}_{K|\overline{K}}^\natural)$ is the following:

$$|\Omega_{Y_{m|n}^\natural}\rangle \otimes |\Omega_{\widetilde{Y}_{K|\overline{K}}^\natural}\rangle = \qquad\qquad\qquad\qquad\qquad . \tag{273}$$

A string ending on $D5_i$ from the left contributes $\varepsilon_i$ to the $\mathfrak{gl}(m|n)$ weight, and a string ending on $D3_\alpha$ from the right contributes $-\varepsilon_\alpha$ to the $\mathfrak{gl}(K|\overline{K})$ weight. This configuration is the tensor

product of two highest-weight vectors because we cannot shorten any of the strings and stretch it between another pair of D3-brane and D5-brane; by doing so we would get more than one strings between $D3_3$ and $D5_1$, but that is prohibited as such strings have necessarily coincident worldsheets and are fermionic. This brane diagram shows

$$|\Omega_{Y^\natural_{m|n}}\rangle \otimes |\Omega_{\widetilde{Y}^\natural_{K|\overline{K}}}\rangle = (\chi_2^3)^2 \chi_1^3 (\chi_1^2)^4 |0\rangle. \tag{274}$$

The other vectors in $L(Y^\natural_{m|n}) \otimes |\Omega_{\widetilde{Y}^\natural_{K|\overline{K}}}\rangle$ can also be represented by brane configurations. For example,

$$q_{21}|\Omega_{Y^\natural_{m|n}}\rangle \otimes |\Omega_{\widetilde{Y}^\natural_{K|\overline{K}}}\rangle = \sum_{\alpha=1}^3 \chi_2^\alpha \eta_\alpha^1 (\chi_2^3)^2 \chi_1^3 (\chi_1^2)^4 |0\rangle \tag{275}$$
$$= -4(\chi_2^3)^2 \chi_1^3 \chi_2^2 (\chi_1^2)^3 |0\rangle + (\chi_2^3)^3 (\chi_1^2)^4 |0\rangle$$

is a linear combination of two states, which one obtains from the highest-weight vector by extending one of the strings to the right:

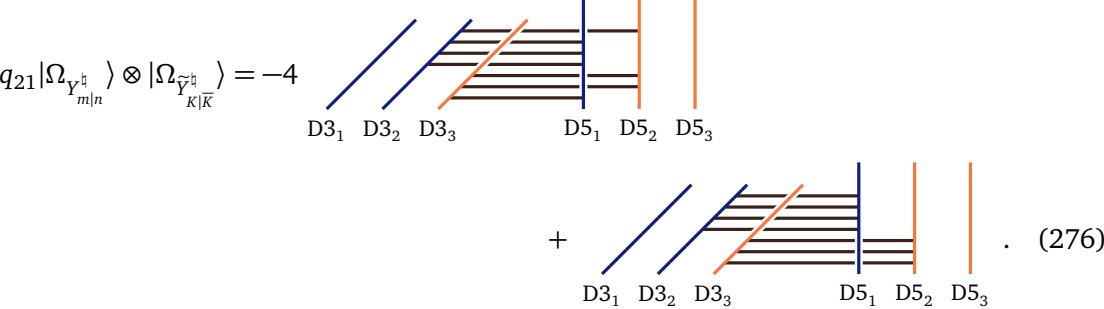

$$\tag{276}$$

The construction of a line defect in a contravariant representation of $\mathfrak{gl}(m|n)$ is analogous. For a contravariant representation, the D3-branes are placed to the right of the D5-branes. For example, for the same choice $(m|n) = (1|2)$, $(K|\overline{K}) = (2|1)$ and $Y = (5, 1, 1)$, we have $\widetilde{Y}^\natural_{m|n} = (-3| -1, -3)$ and $Y^\natural_{K|\overline{K}} = (5, 1|1)$, and the highest-weight vector is represented by the configuration

$$\tag{277}$$

## 4.4 Two-dimensional $\mathcal{N} = (2, 2)$ supersymmetric gauge theories

Applying S-duality and T-duality on $S^1$ to the brane configurations for a line defect, we obtain D2–D4–NS5 brane configurations which describe two-dimensional $\mathcal{N} = (2, 2)$ supersymmetric field theories. For a general choice of $(K|\overline{K})$ and $Y$, the resulting theory does not seem to admit a simple gauge theory description.

If we restrict to covariant representations with $K = 0$ and $Y_{m+1} = 0$ and contravariant representations with $\overline{K} = 0$ and $Y'_{n+1} = 0$, the two-dimensional theories are particularly nice. Let us consider these cases.

For a covariant representation with $K = 0$ and $Y_{m+1} = 0$, the relevant highest weights are $Y^\natural_{m|n} = (Y_1, \ldots, Y_m | 0, \ldots, 0)$ and $\widetilde{Y}^\natural_{K|\overline{K}} = (-Y'_{\overline{K}}, \ldots, -Y'_1)$. The brane configuration for the highest-weight vector of $L(Y^\natural_{m|n}) \otimes L(\widetilde{Y}^\natural_{K|\overline{K}})$ has one string stretched between $NS5_i$ and $D3_{\overline{K}-\alpha+1}$

for each $\alpha = 1, \ldots, Y_i$. The following diagram depicts the highest-weight vector for $(m|n) = (3|2)$, $(K|\overline{K}) = (0|4)$ and $Y = (4, 2, 1)$, for which $Y' = (3, 2, 1, 1)$, $Y^{\natural}_{m|n} = (4, 2, 1|0, 0)$ and $\widetilde{Y}^{\natural}_{K|\overline{K}} = (-1, -1, -2, -3)$:

$$\text{(D3}_1 \ \text{D3}_2 \ \text{D3}_3 \ \text{D3}_4 \qquad \text{D5}_1 \ \text{D5}_2 \ \text{D5}_3 \ \text{D5}_4 \ \text{D5}_5) \tag{278}$$

By moving the D3-branes past D5-branes, we can bring this configuration to another configuration in which the D3-branes are located between D5-branes and have no strings attached:[11]

$$\text{(D3}_1 \ \text{D3}_2 \quad \text{D3}_3 \quad \text{D3}_4 \qquad \quad \text{D5}_1 \qquad \text{D5}_2 \qquad \text{D5}_3 \qquad \text{D5}_4 \qquad \text{D5}_5) \tag{279}$$

The strings that were initially present get annihilated by the Hanany–Witten transition. The number of D3-branes between $\text{D5}_i$ and $\text{D5}_{i+1}$ is equal to $Y_i - Y_{i+1}$.

If we stretch strings between D5-branes in this configuration, then by the reverse Hanany–Witten moves we get a configuration for excited states in $L(Y^{\natural}_{m|n}) \otimes |\Omega_{\widetilde{Y}^{\natural}_{K|\overline{K}}}\rangle$. For example, the configuration

$$\tag{280}$$

represents an excited state with weight $(4, 2, 1|0, 0) - 3\alpha_1 - 2\alpha_2 - 2\alpha_3 - 2\alpha_4 = (1, 3, 1|0, 2)$. One such state is represented by the configuration

$$\tag{281}$$

Similarly, for a contravariant representation with $\overline{K} = 0$ and $Y'_{n+1} = 0$, we have $\widetilde{Y}^{\natural}_{m|n} = (0, \ldots, 0|-Y'_n, \ldots, -Y'_1)$ and $Y^{\natural}_{K|\overline{K}} = (Y_1, \ldots, Y_K)$, and the brane configuration for the highest-weight vector of $L(\widetilde{Y}^{\natural}_{m|n}) \otimes L(Y^{\natural}_{K|\overline{K}})$ can be brought to a configuration without any strings. Take an example with $(K|\overline{K}) = (3|0)$ and $Y = (2, 2, 1)$, for which $Y' = (3, 2)$, $\widetilde{Y}^{\natural}_{m|n} = (0, 0, 0|-2, -3)$ and $Y^{\natural}_{K|\overline{K}} = (2, 2, 1)$:

$$\tag{282}$$

This configuration represents an excited state with weight $(0, 0, 0|-2, -3) - \alpha_1 - \alpha_2 - \alpha_3 - \alpha_4 = (-1, 0, 0|-2, 0)$. The number of D3-branes between an adjacent pair of D5-branes can be read off from $Y'$.

---

[11]An obstruction to generalize the present argument to more general covariant and contravariant representations is that we do not understand what happens when a D3-brane passes through a D5-brane of the same color in the presence of the RR two-form for $\Omega$-deformation.

The dual D2–D4–NS5 brane configurations are of the type studied by Hanany and Hori [33] and realize quiver gauge theories. The quiver for the configuration (280) is

$$\tag{283}$$

and the quiver for the configuration (281) is

$$\tag{284}$$

The ranks of the gauge nodes are given by the numbers of F1-branes, and the ranks of the flavor nodes are given by the numbers of D3-branes. There are $\mathcal{N} = (4,4)$ cubic superpotential terms involving adjoint chiral multiplets.

The flavor symmetry for the chiral multiplets charged under the $m$th gauge node is doubled due to the lack of cubic superpotential term. From the brane point of view, this is because a D4-brane between $\mathrm{NS5}_m$ and $\mathrm{NS5}_{m+1}$ can be broken into half on one of the NS5-branes which has the same color as the D4-brane:

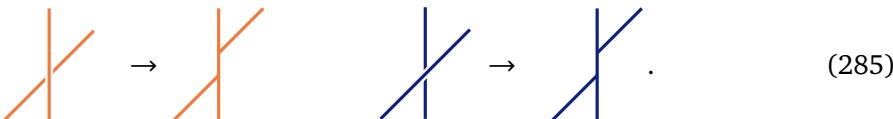

$$\tag{285}$$

Strings stretched between $\mathrm{D2}_m^{a_m}$, $a_m = 1, \ldots, M_m$, and one half of the D4-brane produce a fundamental chiral multiplet for $\mathrm{U}(M_m)$, while strings between those D2-branes and the other half of the D4-brane produce an antifundamental chiral multiplet.

## 4.5 Bethe/gauge correspondence for finite-dimensional representations

Generalizing the above brane construction, we can obtain the Bethe/gauge correspondence for spins valued in arbitrary finite-dimensional representations of $\mathfrak{gl}(m|n)$. This is essentially the correspondence proposed by Nekrasov [3].

Consider the rational $\mathfrak{gl}(m|n)$ spin chain of length $L$, with the $\ell$th spin takes values in finite-dimensional highest-weight representations $L(\lambda^\ell)$. For the moment, let us assume that the highest weights are all integral and satisfy

$$\lambda_1^\ell \geq \cdots \lambda_m^\ell \geq 0 \geq \lambda_{m+1}^\ell \geq \cdots \geq \lambda_{m+n}^\ell. \tag{286}$$

This is the case if all of the representations are of the type studied in section 4.4. We define nonnegative integers

$$K_r^\ell = \lambda_r^\ell - \lambda_{r+1}^\ell, \quad r = 1, \ldots, m-1, \tag{287}$$

$$K_m^\ell = \lambda_m^\ell, \tag{288}$$

$$\overline{K}_r^\ell = \lambda_r^\ell - \lambda_{r+1}^\ell, \quad r = m+1, \ldots, m+n-1, \tag{289}$$

$$\overline{K}_m^\ell = -\lambda_{m+1}^\ell. \tag{290}$$

We look at a sector of fix magnon numbers $(M_1, \ldots, M_{m+n-1})$. The Bethe equations depend only on the highest weights and the magnon numbers, so their form remain the same as in the case of Verma modules.

The gauge theory corresponding to this magnon sector is similar to the theory discussed in section 2.3 and has the same gauge symmetry. The difference is that the chiral multiplets $Q_i$, $\widetilde{Q}_i$, $i = 1, \ldots, m + n$, are replaced by chiral multiplets

$$R_r^\ell \in \mathrm{Hom}(\mathbb{C}^{K_r^\ell}, \mathbb{C}^{M_r}), \quad r = 1, \ldots, m, \tag{291}$$

$$\widetilde{R}_r^\ell \in \mathrm{Hom}(\mathbb{C}^{M_r}, \mathbb{C}^{K_r^\ell}), \quad r = 1, \ldots, m, \tag{292}$$

$$S_r^\ell \in \mathrm{Hom}(\mathbb{C}^{K_r^\ell}, \mathbb{C}^{M_r}), \quad r = m + 1, \ldots, m + n - 1, \tag{293}$$

$$\widetilde{S}_r^\ell \in \mathrm{Hom}(\mathbb{C}^{M_r}, \mathbb{C}^{K_r^\ell}), \quad r = m + 1, \ldots, m + n - 1. \tag{294}$$

Letting

$$K_r = \sum_{\ell=1}^{L} K_r^\ell, \tag{295}$$

$$\overline{K}_r = \sum_{\ell=1}^{L} \overline{K}_r^\ell, \tag{296}$$

we can combine them into chiral multiplets $R_r \in \mathrm{Hom}(\mathbb{C}^{K_r}, \mathbb{C}^{M_r})$, $\widetilde{R}_r \in \mathrm{Hom}(\mathbb{C}^{M_r}, \mathbb{C}^{K_r})$, $r = 1, \ldots, m - 1$, and $S_r \in \mathrm{Hom}(\mathbb{C}^{K_r}, \mathbb{C}^{M_r})$, $\widetilde{S}_r \in \mathrm{Hom}(\mathbb{C}^{M_r}, \mathbb{C}^{K_r})$, $r = m + 1, \ldots, m + n - 1$.

For $r < m$, the theory has the $\mathcal{N} = (4, 4)$ superpotential term $\mathrm{tr}_{\mathbb{C}^{K_r}}(\widetilde{R}_r \phi_r R_r)$, and a flavor symmetry $U(K_r)$ act on $R_r$ and $\widetilde{R}_r$. For $r = m$, the cubic superpotential is absent and two copies of $U(K_m)$ act separately on $R_m$ and $\widetilde{R}_m$. Similar statements hold for $S_r$ and $\widetilde{S}_r$. Under $U(1)_\hbar$, $R_r$ and $\widetilde{R}_r$ have charge $-1$ and $S_r$ and $\widetilde{S}_r$ have charge $+1$.

The gauge and matter contents of the theory can be encoded in a quiver. For $(m|n) = (3, 2)$, the quiver is

$$\begin{array}{c} \text{(quiver diagram)} \end{array} \tag{297}$$

Furthermore, the theory admits a brane construction. For example, for $(M_1, M_2, M_3, M_4) = (3, 2, 2, 2)$ and $(K_1, K_2, K_3 | \overline{K}_3, \overline{K}_4) = (2, 1, 1 | 1, 1)$, the brane configuration for the above quiver is

$$\begin{array}{c} \text{(brane diagram)} \end{array} \tag{298}$$

Note that in order to realize the flavor symmetry $U(K_m)^2 \times U(\overline{K}_m)^2$, each D4-brane between $\mathrm{NS5}_m$ and $\mathrm{NS5}_{m+1}$ needs to be brought to the NS5-brane of the same color and broken into half.

We turn on mass parameters for the global symmetries in such a way that higgsing give

the following masses:

$$(R_r^\ell)^{a_r}{}_l : \sigma_r^{a_r} - \mu_{r,l}^\ell - \frac{1}{2}\hbar, \tag{299}$$

$$(\widetilde{R}_r^\ell)^l{}_{a_r} : \tilde{\mu}_{r,l}^\ell - \sigma_r^{a_r} - \frac{1}{2}\hbar, \tag{300}$$

$$(S_r^\ell)^{a_r}{}_l : \sigma_r^{a_r} - \nu_{r,l}^\ell + \frac{1}{2}\hbar, \tag{301}$$

$$(\widetilde{S}_r^\ell)^l{}_{a_r} : \tilde{\nu}_{r,l}^\ell - \sigma_r^{a_r} + \frac{1}{2}\hbar. \tag{302}$$

We necessarily have $\mu_{r,l}^\ell = \tilde{\mu}_{r,l}^\ell$ and $\nu_{r,l}^\ell = \tilde{\nu}_{r,l}^\ell$ for $r \neq m$.

From these expressions for the masses, we see that for $r \leq m$, the pair $(R_r^\ell, \widetilde{R}_r^\ell)$ contributes to the vacuum equations the factor

$$\prod_{l=1}^{K_r^\ell} \frac{\sigma_r^{a_r} - \mu_{r,l}^\ell - \frac{1}{2}\hbar}{\sigma_r^{a_r} - \tilde{\mu}_{r,l}^\ell + \frac{1}{2}\hbar}, \tag{303}$$

and for $r \geq m$, the pair $(S_r^\ell, \widetilde{S}_r^\ell)$ contributes the factor

$$\prod_{l=1}^{\overline{K}_r^\ell} \frac{\sigma_r^{a_r} - \nu_{r,l}^\ell + \frac{1}{2}\hbar}{\sigma_r^{a_r} - \tilde{\nu}_{r,l}^\ell - \frac{1}{2}\hbar}. \tag{304}$$

For the Bethe/gauge correspondence to exist, the above factors should reproduce the factor

$$\frac{\sigma_r^{a_r} - \zeta^\ell + (-1)^{[r+1]}\lambda_{r+1}^\ell\hbar - \frac{1}{2}c_r\hbar}{\sigma_r^{a_r} - \zeta^\ell + (-1)^{[r]}\lambda_r^\ell\hbar - \frac{1}{2}c_r\hbar}, \tag{305}$$

in the Bethe equations. This is indeed possible if we identify the parameters as

$$\mu_{r,l}^\ell = \tilde{\mu}_{r,l}^\ell = \zeta^\ell - (\lambda_r^\ell + \frac{1}{2}c_r + \frac{1}{2})\hbar + l\hbar, \tag{306}$$

$$\nu_{r,l}^\ell = \tilde{\nu}_{r,l}^\ell = \zeta^\ell + (\lambda_{r+1}^\ell + \frac{1}{2}c_r - \frac{1}{2})\hbar + l\hbar. \tag{307}$$

We have

$$\prod_{l=1}^{K_r^\ell} \frac{\sigma_r^{a_r} - \mu_{r,l}^\ell - \frac{1}{2}\hbar}{\sigma_r^{a_r} - \tilde{\mu}_{r,l}^\ell + \frac{1}{2}\hbar} = \frac{\sigma_r^{a_r} - \mu_{r,K_r^\ell}^\ell - \frac{1}{2}\hbar}{\sigma_r^{a_r} - \mu_{r,1}^\ell + \frac{1}{2}\hbar} = \frac{\sigma_r^{a_r} - \zeta^\ell + (\lambda_r^\ell - K_r^\ell)\hbar - \frac{1}{2}c_r\hbar}{\sigma_r^{a_r} - \zeta^\ell + \lambda_r^\ell\hbar - \frac{1}{2}c_r\hbar}, \tag{308}$$

for $r \leq m$ and

$$\prod_{l=1}^{\overline{K}_r^\ell} \frac{\sigma_r^{a_r} - \nu_{r,l}^\ell + \frac{1}{2}\hbar}{\sigma_r^{a_r} - \tilde{\nu}_{r,l}^\ell - \frac{1}{2}\hbar} = \frac{\sigma_r^{a_r} - \nu_{r,1}^\ell + \frac{1}{2}\hbar}{\sigma_r^{a_r} - \nu_{r,\overline{K}_r^\ell}^\ell - \frac{1}{2}\hbar} = \frac{\sigma_r^{a_r} - \zeta^\ell - \lambda_{r+1}^\ell\hbar - \frac{1}{2}c_r\hbar}{\sigma_r^{a_r} - \zeta^\ell - (\lambda_{r+1}^\ell + \overline{K}_r^\ell)\hbar - \frac{1}{2}c_r\hbar}, \tag{309}$$

for $r \geq m$, so we obtain the factor (305) using the definitions of $K_r^\ell$ and $\overline{K}_r^\ell$.

Now, let us consider the case in which the representations of the spin variables are arbitrary finite-dimensional ones. Even in this general case, most of the above argument actually goes through, with the same definitions of $K_r^\ell$ and $\overline{K}_r^\ell$ for $r \neq m$ and the identifications (306)

and (307). The only place that fails is where we set $r = m$: if we choose nonnegative integers $K_m^\ell$, $\overline{K}_m^\ell$ and write down the product of the factors (303) and (304) for $r = m$, we get

$$\frac{\sigma_m^{a_m} - \zeta^\ell - \lambda_{m+1}^\ell \hbar - \frac{1}{2} c_m \hbar}{\sigma_m^{a_m} - \zeta^\ell + \lambda_m^\ell \hbar - \frac{1}{2} c_m \hbar} \left( \frac{\sigma_m^{a_m} - \zeta^\ell + (\lambda_m^\ell - K_m^\ell)\hbar - \frac{1}{2} c_m \hbar}{\sigma_m^{a_m} - \zeta^\ell - (\lambda_{m+1}^\ell + \overline{K}_m^\ell)\hbar - \frac{1}{2} c_m \hbar} \right), \tag{310}$$

whereas the Bethe equations do not contain the second fraction in the parenthesis.

We can cancel this unwanted factor if we introduce additional chiral multiplets

$$\widetilde{R}_m^\ell \in \mathrm{Hom}(\mathbb{C}^{M_m}, \mathbb{C}^{K_m^\ell}), \tag{311}$$

$$S_m^\ell \in \mathrm{Hom}(\mathbb{C}^{\overline{K}_m^\ell}, \mathbb{C}^{M_m}) \tag{312}$$

and give them appropriate masses. These chiral multiplets are produced by semi-infinite D4-branes ending on NS5$_m$ and NS5$_{m+1}$.

# Acknowledgments

We would like to thank Kevin Costello, Mykola Dedushenko and Philsang Yoo for helpful discussions. NI gratefully acknowledges support from the Institute for Advanced Study and the National Science Foundation under Grant No. PHY-1911298. The work of SFM is funded by the Natural Sciences and Engineering Research Council of Canada (NSERC). Research at Perimeter Institute is supported in part by the Government of Canada through the Department of Innovation, Science and Economic Development Canada and by the Province of Ontario through the Ministry of Colleges and Universities. Any opinions, findings, and conclusions or recommendations expressed in this material are those of the authors and do not necessarily reflect the views of the funding agencies.

# A Four-dimensional Chern–Simons theory with gauge supergroup from twisted string theory

In this appendix we present an alternative construction of four-dimensional Chern–Simons theory with gauge group GL($m|n$), using the framework of twists of superstring theory as developed in [67]. Twisted superstring theory refers to superstring theory in a particular RR background where the bosonic ghost for local supersymmetries may take a nonzero nilpotent vacuum expectation value $Q$. When one considers D-branes in such backgrounds, the coupling between D-branes and the bosonic ghost dictates that $Q$ is added to the BRST differential of the worldvolume theory [67]. Therefore, the field content of worldvolume theories of branes in twisted superstrings are naturally $Q$-cohomology of that of the supersymmetric gauge theories one would find in the absence of the additional RR background. As such, twisted superstrings affords a useful framework for studying protected sectors of supersymmetric gauge theories.

Costello and Li [67] give conjectural descriptions of such twists of superstrings in terms of topological strings. These conjectures have passed several consistency checks [67–69] and have been proven at the level of the free limit of the supergravity approximation [70]. Taking these conjectures as a starting point, one can derive simple descriptions of twists of worldvolume theories of D-branes using mathematical tools from the study of topological strings. Though such calculations require machinery from homological algebra, they have the benefit of calculational ease. Tractable models of twisted worldvolume theories can be determined from an Ext-algebra computation, and the action functional can be read off from an algebraic

structure and trace on the Ext-algebra; no term-matching arguments involving the Dirac–Born–Infeld action are required.

In this appendix, we work with field theory in the Batalin–Vilkovisky (BV) formalism as articulated by [71, 72]. In particular, we freely make use of the language of $L_\infty$-algebras. Much of the below is exposited elsewhere in the literature. The construction of twisted supergravity and the conjectural descriptions of twists of superstrings in terms of topological strings are given in [67]. Many of the examples below are worked out in [73] where more formal aspects of the framework are articulated and some mathematical applications are discussed. We hope the exposition of this appendix will have the simultaneous benefit of illustrating the calculational utility of twisted superstrings, and making our constructions parseable to more mathematically minded readers.

## A.1 Topological strings

We begin with some recollections on topological strings. The worldsheet theory of a topological string theory is a two-dimensional oriented topological quantum field theory. Treating such theories via the language of functorial field theory, the results of [74, 75] tell us that such theories are determined by the data of a Calabi–Yau category. Physically, we think of objects of this category as D-branes in our topological string theory, and the space of homomorphisms between two objects as the complex computing BRST cohomology of the states of open strings stretched between the branes. It is known that spaces of open string states have an algebra structure, with respect to which the action for open string field theory takes a simple form [76]. The data of a Calabi–Yau category is exactly what is needed to make precise this algebraic structure; this will be elaborated more on subsection A.2 below.

**Example A.1.** Let $M$ be a symplectic four-manifold and $X$ a Calabi–Yau three-fold. The SU(3)-invariant twist of type IIB string theory is given by the Calabi–Yau 5-category $\mathrm{Fuk}(M)\otimes\mathrm{Coh}(X)$. Here, $\mathrm{Fuk}(M)$ refers to the Fukaya category of $M$ and $\mathrm{Coh}(X)$ refers to the category of coherent sheaves on $X$. This describes a topological string theory that looks like a combination of the A-model into $M$ and the B-model into $X$.

Here the terminology is meant to indicate that the above mixed A-B model conjecturally arises from type IIB string theory in an RR background in which the bosonic ghost takes a vacuum expectation value given by an SU(3)-invariant nilpotent element of the ten-dimensional $\mathcal{N}=(2,0)$ supersymmetry algebra.

**Remark A.2.** Let us elaborate on our description of the A-model directions. For us the main relevant example will be when $M=\mathbb{R}^{2N}$. In this case, we will use a version of the Fukaya category that we will denote $\mathrm{Fuk}^0$ where we discard counts of pseudo-holomorphic discs with nonzero area. Explicitly, the objects in the category will consist of Lagrangians in $M$, and for two Lagrangians $L_1, L_2 \subset \mathbb{R}^{2N}$ with clean intersection, we have that $\mathrm{Hom}_{\mathrm{Fuk}^0}(L_1, L_2) = \Omega^\bullet(L_1 \cap L_2)$. This will suffice for our purposes as we will primarily care about perturbative phenomena on worldvolume theories of branes, so we may neglect worldsheet instantons.

In addition to this restriction on the space of homomorphisms, this category does not include as objects, coisotropic A-branes. To the authors' knowledge, it is an open mathematical problem to construct a version of the Fukaya category that includes as objects such branes. Fortunately, we will not need to consider such branes in our analysis.

To a topological string theory, we may associate two field theories which are versions of open string field theory and closed string field theory respectively. The former recovers twists of worldvolume theories of branes in the physical string while the latter contains twists of supergravity.

## A.2 Topological open string field theory

Let $\mathcal{C}$ be an $A_\infty$-category and let $\mathcal{F} \in \mathcal{C}$ be an object. Then $\mathrm{Hom}_\mathcal{C}(\mathcal{F}, \mathcal{F})$ is an $A_\infty$-algebra, and skew-symmetrizing the $A_\infty$-operations yields an $L_\infty$-algebra. Now suppose our $\mathcal{C}$ is in fact a Calabi–Yau $N$-category, and as such can be thought of as determining a topological string theory. Then for any object $\mathcal{F} \in \mathcal{C}$, we have an invariant pairing $\mathrm{tr}\colon \mathrm{Hom}_\mathcal{C}(\mathcal{F}, \mathcal{F}) \to \mathbb{C}[N]$.

In examples of interest, where $\mathcal{C}$ is attached to a $2N$-manifold thought of the target spacetime of our topological string, $\mathrm{Hom}_\mathcal{C}(\mathcal{F}, \mathcal{F})$ will arise as sections of a natural graded vector bundle over the support of $\mathcal{F}$, the $L_\infty$-structure maps will be given by polydifferential operators, and the trace map will factor through integration over the support of $\mathcal{F}$. In such instances, the data of this $L_\infty$-algebra and the trace pairing then determine the data of a perturbative $\mathbb{Z}_2$-graded BV theory – the space of fields of the theory is $\Pi \mathrm{Hom}_\mathcal{C}(\mathcal{F}, \mathcal{F})$ and the action is given by

$$S(\alpha) = \sum_{k \geq 1} \frac{1}{(k+1)!} \mathrm{tr}\big(\alpha \otimes \ell_k(\alpha^{\otimes k})\big), \tag{A.1}$$

where $\ell_k \colon \mathrm{Hom}_\mathcal{C}(\mathcal{F}, \mathcal{F})^{\otimes k} \to \mathrm{Hom}_\mathcal{C}(\mathcal{F}, \mathcal{F})$ are the $L_\infty$-structure maps. This theory is the worldvolume theory of the D-brane $\mathcal{F}$ in the topological string theory determined by $\mathcal{C}$. The conjectural descriptions of twists of superstrings in terms of topological strings imply that for $\mathcal{C}$ coming from a twist of a superstring theory, the worldvolume theory of $\mathcal{F}$ is a twist of the worldvolume theory of the corresponding brane in the physical string theory.

**Example A.3.** Consider the SU(3)-invariant twist of type IIB string theory on $\mathbb{R}^4 \times \mathbb{C}^3$ from example A.1, and consider a stack of $n$ D5-branes wrapping $\mathbb{R}^2 \times \mathbb{C}^2$. As explained in the above example, this twist of type IIB string theory is described by the Calabi–Yau category $\mathcal{C} = \mathrm{Fuk}^0(\mathbb{R}^4) \otimes \mathrm{Coh}(\mathbb{C}^3)$. The object describing our stack of branes is given by $(\mathbb{R}^2, \mathcal{O}^n_{\mathbb{C}^2})$. We have that

$$
\begin{aligned}
\mathrm{Ext}_\mathcal{C}\big((\mathbb{R}^2, \mathcal{O}^n_{\mathbb{C}^2}), (\mathbb{R}^2, \mathcal{O}^n_{\mathbb{C}^2})\big) &= \mathrm{Hom}_{\mathrm{Fuk}^0}(\mathbb{R}^2, \mathbb{R}^2) \otimes \mathrm{Ext}_{\mathrm{Coh}(\mathbb{C}^3)}(\mathcal{O}^n_{\mathbb{C}^2}, \mathcal{O}^n_{\mathbb{C}^2}) \\
&= \Omega^\bullet(\mathbb{R}^2) \otimes \mathrm{Ext}_{\mathrm{Coh}(\mathbb{C}^3)}(\mathcal{O}_{C^2}, \mathcal{O}_{\mathbb{C}^2}) \otimes \mathfrak{gl}(n) \\
&= \Omega^\bullet(\mathbb{R}^2) \otimes \Omega^{0,\bullet}(\mathbb{C}^2)[\varepsilon] \otimes \mathfrak{gl}(n).
\end{aligned}
\tag{A.2}
$$

In the last step, we have used the following general result:

**Lemma A.4.** Let $X$ be a Calabi–Yau manifold and let $Y \subset X$ be holomorphic. Then $\mathrm{Ext}_{\mathrm{Coh}(X)}(\mathcal{O}_Y, \mathcal{O}_Y) \cong \Omega^{0,\bullet}(Y, \wedge^\bullet N_{X/Y})$.

We can describe the $L_\infty$-structure as follows. There is an $L_\infty$-structure on $\Omega^\bullet(\mathbb{R}^2) \otimes \Omega^{0,\bullet}(\mathbb{C}^2) \otimes \mathfrak{gl}(n)$ given by

$$\ell_1 = \mathrm{d} \otimes 1_{\Omega^{0,\bullet}(\mathbb{C}^2)} \otimes 1_{\mathfrak{gl}(n)} + 1_{\Omega^\bullet(\mathbb{R}^2)} \otimes \bar{\partial} \otimes 1_{\mathfrak{gl}(n)}, \tag{A.3}$$

$$\ell_2 = \wedge \otimes \wedge \otimes [-,-]_{\mathfrak{gl}(n)}, \tag{A.4}$$

$$\ell_k = 0, \quad k \geq 3. \tag{A.5}$$

The $L_\infty$-structure on $\Omega^\bullet(\mathbb{R}^4) \otimes \Omega^{0,\bullet}(\mathbb{C}^2)[\varepsilon] \otimes \mathfrak{gl}(n)$ is given by the semidirect product

$$\big(\Omega^\bullet(\mathbb{R}^4) \otimes \Omega^{0,\bullet}(\mathbb{C}^2) \otimes \mathfrak{gl}(n)\big) \ltimes \varepsilon\big(\Omega^\bullet(\mathbb{R}^4) \otimes \Omega^{0,\bullet}(\mathbb{C}^2) \otimes \mathfrak{gl}(n)\big). \tag{A.6}$$

The trace pairing induced from the Calabi–Yau structure on $\mathcal{C}$ is given

$$\mathrm{tr}\colon \alpha \mapsto \int_{\mathbb{R}^2 \times \mathbb{C}^{2|1}} \mathrm{Tr}(\alpha) \wedge \Omega, \tag{A.7}$$

where $\Omega$ denotes the holomorphic volume form on $\mathbb{C}^2$ and Tr is the Killing form on $\mathfrak{gl}(n)$. Thus, we find that the action of the theory is exactly

$$S(\alpha, \beta) = \int_{\mathbb{R}^2 \times \mathbb{C}^{2|1}} \mathrm{Tr}\left(\frac{1}{2}\beta(d + \bar{\partial})\alpha + \frac{1}{6}\beta \wedge [\alpha, \alpha]\right) \wedge \Omega, \qquad (A.8)$$

for

$$\alpha \in \Omega^{\bullet}(\mathbb{R}^2) \otimes \Omega^{0,\bullet}(\mathbb{C}^2) \otimes \mathfrak{gl}(n), \qquad (A.9)$$

$$\beta \in \Omega^{\bullet}(\mathbb{R}^2) \otimes \Omega^{0,\bullet}(\mathbb{C}^2)\varepsilon \otimes \mathfrak{gl}(n). \qquad (A.10)$$

This is exactly the holomorphic–topological twist of six-dimensional $\mathcal{N} = (1,1)$ super Yang–Mills theory, dubbed the rank $(1,1)$ partially holomorphic topological twist in [77].

### A.3  Topological closed string field theory

Let $Z$ be the worldsheet theory determined by the Calabi–Yau category $\mathcal{C}$. Naively, the closed string states of the theory should be given by the local operators of the worldsheet theory, $Z(S^1)$. However, the worldsheet theory in the physical string is coupled to two-dimensional gravity – closed string states should be those local operators invariant under reparametrizations of the worldsheet. Since the worldsheet theory is topological, Cartan's magic formula tells us that small reparametrizations will act homotopically trivially on the space of local operators. In the setting of topological strings, there is a natural homotopy action of $S^1$ on $Z(S^1)$ – the closed string states will be the invariants $Z(S^1)^{S^1}$. In terms of categorical data, this is computed by the cyclic cochains of the category $\mathcal{C}$, $\mathrm{HC}^{\bullet}(\mathcal{C})$. There is a natural way to equip a shift of $\mathrm{HC}^{\bullet}(\mathcal{C})$ with an odd Poisson tensor and an $L_{\infty}$-structure. In examples in which the graded vector space underlying $\mathrm{HC}^{\bullet}(\mathcal{C})$ arises as the space of sections of some graded vector bundle, this gives $\mathrm{HC}^{\bullet}(\mathcal{C})$ the structure of a $\mathbb{Z}_2$-graded Poisson BV theory. The constructions of the $L_{\infty}$- and shifted Poisson structures in this generality are extraneous for our purposes – we will be focused on the following examples.

**Example A.5.** Suppose $\mathcal{C} = \mathrm{Coh}(X)$ with $X$ Calabi–Yau. Then the cyclic formality theorem [78] tells us that there is an equivalence of $L_{\infty}$-algebras

$$\mathrm{HC}^{\bullet}(\mathcal{C}) \cong \left(\mathrm{PV}^{\bullet,\bullet}(X)[\![t]\!], \ell_1 = \bar{\partial} + t\partial, \ell_2 = \{-, -\}\right), \qquad (A.11)$$

where $t$ is a parameter of degree 2, $\partial$ denotes the divergence operator, and $\{-, -\}$ denotes the Schouten bracket of polyvector fields. The Poisson tensor has Poisson kernel $(\partial \otimes 1)\delta_{\Delta(X)}$, where $\Delta(X) \subset X \times X$ denotes the diagonal. This theory is Kodaira–Spencer gravity articulated as Bershadsky–Cecotti–Ooguri–Vafa theory studied by [79–81].

**Example A.6.** Suppose $\mathcal{C} = \mathrm{Fuk}(M)$ with $M$ being a symplectic manifold. The Hochschild (co)chains admit a description in terms of the quantum cohomology of the target. Together with the abstract $L_{\infty}$-structure and the $\mathbb{Z}_2$-graded Poisson structure, we expect that the result will be a version of the Kähler gravity [82]. We will discard worldsheet instantons coming from the A-model directions of the twists of string theory we consider. Therefore, our ansatz will be that the closed string field theory for the A-model directions is described by the $L_{\infty}$-algebra $\Omega^{\bullet}(M)$ with $L_{\infty}$-structure given by $\ell_1 = d$, $\ell_2 = \wedge$ and Poisson structure given by the wedge and integrate pairing. We will abusively continue to denote the closed string field theory in the A-model sans worldsheet instantons by $\mathrm{HC}^{\bullet}(\mathrm{Fuk}(M))$.

**Example A.7.** Putting the above two examples together, we can describe the closed string field theories for the twists of type IIB string theory we are interested in. The closed string

field theory for the SU(3)-invariant twist of type IIB string theory on $\mathbb{R}^4 \times \mathbb{C}^3$ is given by the $L_\infty$-algebra $\Omega^\bullet(\mathbb{R}^4) \otimes \mathrm{PV}^{\bullet,\bullet}(\mathbb{C}^3)[[t]]$ with

$$\ell_1 = \mathrm{d} \otimes 1_{\mathrm{PV}^{\bullet,\bullet}(\mathbb{C}^3)[[t]]} + 1_{\Omega^\bullet(\mathbb{R}^4)} \otimes (\bar{\partial} + t\partial), \tag{A.12}$$

$$\ell_2 = \wedge \otimes \{-,-\}, \tag{A.13}$$

$$\ell_k = 0, \quad k \geq 3. \tag{A.14}$$

The Poisson tensor is given by the Poisson kernel $(\partial \otimes 1)\delta_{\Delta(\mathbb{C}^3)}\delta_{\Delta(\mathbb{R}^4)}$.

### A.4 Closed–open map

Given a Calabi–Yau category $\mathcal{C}$ and an object $\mathcal{F} \in \mathcal{C}$, there is always an $L_\infty$-map

$$\mathrm{HC}^\bullet(\mathcal{C}) \to \mathrm{CE}^\bullet\big(\mathrm{Hom}_\mathcal{C}(\mathcal{F}, \mathcal{F})\big). \tag{A.15}$$

Here, the target denotes Chevalley–Eilenberg cochains on the $L_\infty$-algebra $\mathrm{Hom}_\mathcal{C}(\mathcal{F}, \mathcal{F})$; this is a model for Hamiltonian vector fields on the formal moduli space describing fluctuations of the brane $\mathcal{F}$. This map takes a closed string field and produces a single trace-operator on the worldvolume theory of $\mathcal{F}$, which describes how the closed string field couples to the worldvolume theory of $\mathcal{F}$. We will wish to apply this to examples where $\mathcal{C} = \mathrm{Fuk}^0(\mathbb{R}^{10-2N}) \otimes \mathrm{Coh}(\mathbb{C}^N)$, and $\mathcal{F} = (\mathbb{R}^{5-N}, \mathcal{O}^n_{\mathbb{C}^k})$ for $k \leq N$. Then we have that

$$\mathrm{HC}^\bullet(\mathcal{C}) = \Omega^\bullet(\mathbb{R}^{10-2N}) \otimes \mathrm{PV}^{0,\bullet}(\mathbb{C}^N)[[t]], \tag{A.16}$$

$$\mathrm{Hom}_\mathcal{C}(\mathcal{F}, \mathcal{F}) = \Omega^\bullet(\mathbb{R}^{5-N}) \otimes \Omega^{0,\bullet}(\mathbb{C}^k)[\varepsilon_{k+1}, \ldots, \varepsilon_N] \otimes \mathfrak{gl}(n). \tag{A.17}$$

This map should be thought of as given by a sum of disk amplitudes with boundary on the brane $\mathcal{F}$ and with an arbitrary number of marked points on the interior labeling closed string insertions. We will only consider single closed string insertions of the form $1 \otimes \mu \in \Omega^\bullet(\mathbb{R}^{10-2n}) \otimes \mathrm{PV}^{0,\bullet}(\mathbb{C}^N)$. In particular the field does not depend on the A-twisted directions of spacetime, or the parameter $t$. For such fields we have the following explicit formula for the linear component of the closed–open map

$$1 \otimes w_1^{a_1} \cdots w_N^{a_N} \partial_{w_1}^{b_1} \cdots \partial_{w_N}^{b_N} \mapsto I(\alpha), \tag{A.18}$$

where

$$I(\alpha) = \frac{1}{(n+1)!} \int_{\mathbb{R}^{5-N} \times \mathbb{C}^{k|N-k}} \mathrm{Tr}(w_1^{a_1} \cdots w_N^{a_N} \varepsilon_{k+1}^{b_{k+1}} \cdots \varepsilon_N^{b_N}$$
$$\times \partial_{\varepsilon_{k+1}}^{b_{k+1}} \alpha \wedge \cdots \wedge \partial_{\varepsilon_N}^{b_N} \alpha \wedge \partial_{w_1}^{b_1} \alpha \wedge \cdots \wedge \partial_{w_k}^{b_k} \alpha) \wedge \Omega. \tag{A.19}$$

A version of this result including formulas for the deformation to all orders in open string insertions is proved in [81]. It is worth emphasizing that deriving formulas for this map at all orders is an extremely nontrivial problem – for $\mu \in \mathrm{PV}^{2,0}$ this is the content of the holomorphic analogue of Kontsevich's theorem on deformation quantization.

**Example A.8.** Consider the SU(3)-invariant twist of type IIB string theory on $\mathbb{R}^4 \times \mathbb{C}^3$. We fix once and for all coordinates $z$, $w_1$, $w_2$ on $\mathbb{C}^3$. We saw in the example above that a stack of $n$ D5-branes wrapping $\mathbb{R}^2 \times \mathbb{C}^2_{z,w_1}$ gives rise to a holomorphic–topological twist of six-dimensional $\mathcal{N} = (1,1)$ super Yang–Mills theory with gauge group U($n$). Let us now consider what happens when we turn on a field $1 \otimes w_1 w_2 \in \Omega^\bullet(\mathbb{R}^4) \otimes \mathrm{PV}^{0,\bullet}(\mathbb{C}^3)$. Recall that the fields of the relevant twist of six-dimensional $\mathcal{N} = (1,1)$ super Yang–Mills theory were given by $\Omega^\bullet(\mathbb{R}^2) \otimes \Omega^{0,\bullet}(\mathbb{C}^2_{z,w_1})[\varepsilon] \otimes \mathfrak{gl}(n)$.

The image of the closed string field $w_1 w_2$ under the closed open map becomes the functional

$$I(\alpha) = \int_{\mathbb{R}^2 \times \mathbb{C}^{2|1}} \mathrm{Tr}(\alpha w_1 \partial_\varepsilon \alpha \wedge \Omega). \tag{A.20}$$

Equivalently, this deforms the $L_\infty$-structure on $\Omega^\bullet(\mathbb{R}^2) \otimes \Omega^{0,\bullet}(\mathbb{C}^2)[\varepsilon] \otimes \mathfrak{gl}(n)$ so that $\ell_1 = \mathrm{d} \otimes 1_{\Omega^{0,\bullet}(\mathbb{C}^2[\varepsilon])} \otimes 1_{\mathfrak{gl}(n)} + 1_{\Omega^\bullet(\mathbb{R}^2)} \otimes (\bar\partial + w_1 \partial_\varepsilon) \otimes 1_{\mathfrak{gl}(n)}$. The differential $w_1 \partial_\varepsilon$ has the effect of deforming the complex of fields of the theory into

$$\Omega^\bullet(\mathbb{R}^2) \otimes \left( \Omega^{0,\bullet}(\mathbb{C}^2)\varepsilon \xrightarrow{w_1 \partial_\varepsilon} \Omega^{0,\bullet}(\mathbb{C}^2) \right) \otimes \mathfrak{gl}(n). \tag{A.21}$$

This is the Koszul resolution of the locus $w_1 = 0$, so is quasi-isomorphic to $\Omega^\bullet(\mathbb{R}^2) \otimes \Omega^{0,\bullet}(\mathbb{C}) \otimes \mathfrak{gl}(n)$. This is exactly four-dimensional Chern–Simons theory as a $\mathbb{Z}_2$-graded BV theory.

**Remark A.9.** Note that the above construction differs slightly from the construction of four-dimensional Chern–Simons theory via $\Omega$-deformation in [16]. Conjecturally, the quadratic superpotential we have introduced should describe those components of the RR two-form used in [16] that are not exact for the twist we are performing. However, checking this explicitly is a difficult task. Moreover, at the level of field theory, the construction in [16] came from subjecting the holomorphic–topological twist of six-dimensional $\mathcal{N} = (1, 1)$ super Yang–Mills theory with BV fields given by $\Omega^\bullet(\mathbb{R}^4) \otimes \Omega^{0,\bullet}(\mathbb{C}) \otimes \mathfrak{gl}(n)$ to a B-type $\Omega$-background along $\mathbb{R}^2 \subset \mathbb{R}^4$. Such a construction involves replacing a factor of $\Omega^\bullet(\mathbb{R}^2)$ with the Cartan model for $S^1$-equivariant cohomology of $\mathbb{R}^2$, which is given by the abelian dg Lie algebra $(\Omega^\bullet(\mathbb{R}^2)[u]^{S^1}, \mathrm{d} + u \iota_{\partial_\theta})$, where $u$ is an equivariant parameter, and $\partial_\theta$ is the infinitesimal action of rotations. The localization theorem for equivariant cohomology tells us that for generic values of the equivariant parameter, the complex of fields of our theory is quasi-isomorphic to $\Omega^\bullet(\mathbb{R}^2) \otimes \Omega^{0,\bullet}(\mathbb{C}) \otimes \mathfrak{gl}(n)$. Note that the relation $u\mathcal{L}_{\partial_\theta} = [\mathrm{d}, u \iota_{\partial_\theta}]$ coming from Cartan's magic formula tells us that the infinitesimal action of rotations on the fields of our theory is homotopically trivial.

However, in the above construction we instead work with a more minimal twist of six-dimensional $\mathcal{N} = (1, 1)$ super Yang–Mills theory; the twist of the previous paragraph is gotten from deforming the differential on $\Omega^\bullet(\mathbb{R}^2) \otimes \Omega^{0,\bullet}(\mathbb{C}^2)[\varepsilon] \otimes \mathfrak{gl}(n)$ so that $\ell_1 = \mathrm{d} \otimes 1_{\Omega^{0,\bullet}(\mathbb{C})[\varepsilon]} \otimes 1_{\mathfrak{gl}(n)} + 1_{\Omega^\bullet(\mathbb{R}^2)} \otimes (\bar\partial + \varepsilon \partial_{w_1}) \otimes 1_{\mathfrak{gl}(n)}$. Instead of taking this further twist and working equivariantly along the topological plane that the $w_1$-plane becomes, we turned on a deformation coming from a quadratic superpotential. It is worth noting that there is a map from a twist of the four-dimensional $\mathcal{N} = 4$ superconformal algebra to the closed string sector of the SU(3)-invariant twist of type IIB string theory – the quadratic superpotential deformation considered above lies in the image of this map. Moreover, note that we have that $\mathcal{L}_{w_1 \partial_{w_1}} = [\varepsilon \partial_{w_1}, w_1 \partial_\varepsilon]$; we see that the superconformal deformation also makes the complexified action of rotations exact for the B-twist supercharge. This appears to be part of a general pattern where a superconformal deformation of a holomorphic theory is equivalent to an $\Omega$-deformation of a further topological twist [73, 83, 84].

**Remark A.10.** It is also interesting to consider the superpotential $w_1 w_2$ as a deformation of the entire topological string theory, that is, as a deformation of the category of branes. Morally, it should deform the category of coherent sheaves on $\mathbb{C}^3$ to the category of matrix factorizations for the superpotential $w_1 w_2$; the B-model directions of the topological string are turned into a Landau–Ginzburg B-model. The category of matrix factorizations in this case can be described as the category of modules for the Jacobi algebra of the superpotential $w_1 w_2$, which in this case is just the algebra $\mathbb{C}[z]$. Thus, we see that the SU(3)-invariant twist of type IIB string theory localizes to a six-dimensional topological string theory on $\mathbb{R}^4 \times \mathbb{C}$; this makes contact with the work of [85].

## A.5 Four-dimensional Chern–Simons theory with gauge supergroup from the SU(3)-invariant twist of type IIB string theory

In this section we will arrive at four-dimensional Chern–Simons theory with gauge supergroup using the formalism developed in the previous subsections. The calculation is essentially an easy corollary of the examples therein.

We consider the SU(3)-invariant twist of type IIB string theory on $\mathbb{R}^4 \times \mathbb{C}^3$ with a configuration of D-branes as in the following table:

| | $\mathbb{R}^2$ | $\mathbb{R}^2$ | $\mathbb{C}_z$ | $\mathbb{C}_{w_1}$ | $\mathbb{C}_{w_2}$ | |
|---|---|---|---|---|---|---|
| $n$ D5 | $\circ$ | $\times$ | $\times$ | $\times$ | $\circ$ | (A.22) |
| $m$ D5 | $\circ$ | $\times$ | $\times$ | $\circ$ | $\times$ | |

A cross mark means that the D5-brane extends in that direction. We also turn on a closed string field given by the quadratic superpotential $w_1 w_2$. We arrive at a field theory description for this system by first computing the open string field theory using the techniques above, and then applying the closed–open map.

Let $\mathcal{F}_1 = (\mathbb{R}^2, \mathcal{O}^m_{\mathbb{C}^2_{z,w_1}})$, $\mathcal{F}_2 = (\mathbb{R}^2, \mathcal{O}^n_{\mathbb{C}^2_{z,w_2}})$ denote the objects in the categories of D-branes corresponding to the stacks of $n$ and $m$ D5-branes, respectively. We first wish to compute $\mathrm{Hom}_{\mathcal{C}}(\mathcal{F}_1 \oplus \mathcal{F}_2, \mathcal{F}_1 \oplus \mathcal{F}_2)$. Since Hom commutes with direct sums, we have four summands:

- From example A.8, we have that

$$\mathrm{Hom}_{\mathcal{C}}(\mathcal{F}_1, \mathcal{F}_1) = \Omega^\bullet(\mathbb{R}^2) \otimes \Omega^{0,\bullet}(\mathbb{C}^2_{z,w_1})[\varepsilon_2] \otimes \mathfrak{gl}(m), \quad (A.23)$$

$$\mathrm{Hom}_{\mathcal{C}}(\mathcal{F}_2, \mathcal{F}_2) = \Omega^\bullet(\mathbb{R}^2) \otimes \Omega^{0,\bullet}(\mathbb{C}^2_{z,w_2})[\varepsilon_1] \otimes \mathfrak{gl}(n). \quad (A.24)$$

Our convention above is that $\varepsilon_i$ denotes a section of the normal bundle of $\mathbb{C}^2_{z,w_j} \subset \mathbb{C}^3$, where $i \neq j$.

The trace pairing is given by

$$\mathrm{tr}(\alpha) = \int_{\mathbb{R}^2 \times \mathbb{C}^{2|1}} \mathrm{d}z \, \mathrm{d}w_j \, \mathrm{tr}(\alpha). \quad (A.25)$$

- We can compute, using free resolutions of the structure sheaves of the $w_i$-planes, that

$$\mathrm{Hom}_{\mathcal{C}}(\mathcal{F}_1, \mathcal{F}_2) = \Omega^\bullet(\mathbb{R}^2) \otimes \Omega^{0,\bullet}(\mathbb{C}_z) \otimes \mathrm{Hom}(\mathbb{C}^m, \mathbb{C}^n)[-1], \quad (A.26)$$

$$\mathrm{Hom}_{\mathcal{C}}(\mathcal{F}_2, \mathcal{F}_1) = \Omega^\bullet(\mathbb{R}^2) \otimes \Omega^{0,\bullet}(\mathbb{C}_z) \otimes \mathrm{Hom}(\mathbb{C}^n, \mathbb{C}^m)[-1]. \quad (A.27)$$

Each of these are abelian $L_\infty$-algebras, with $\ell_1 = d \otimes 1_{\Omega^{0,\bullet}(\mathbb{C}_z)} \otimes 1_{\mathrm{Hom}} + 1_{\Omega^\bullet(\mathbb{R}^2)} \otimes \bar\partial \otimes 1_{\mathrm{Hom}}$. There is a natural trace pairing on the direct sum

$$\Omega^\bullet(\mathbb{R}^2) \otimes \Omega^{0,\bullet}(\mathbb{C}_z) \otimes T^* \mathrm{Hom}(\mathbb{C}^m, \mathbb{C}^n)[-1]. \quad (A.28)$$

Letting $X$s denote fields valued in $\mathrm{Hom}(\mathbb{C}^m, \mathbb{C}^n)$ and $Y$s fields valued in the cotangent direction, the pairing is given by

$$\mathrm{tr}(X_1 + Y_1, X_2 + Y_2) = \int \mathrm{d}w \big( \mathrm{Tr}_{\mathbb{C}^n}(X_1 Y_2) - \mathrm{Tr}_{\mathbb{C}^m}(Y_1 X_2) \big). \quad (A.29)$$

The action functional induced by this pairing and abelian $L_\infty$-structure is exactly the BV action for a free hypermultiplet in the Kapustin twist. Restricted to fields of ghost number 1, this recovers exactly the action (145).

Thus we see that the entire space of open string states is given by

$$\mathcal{E} = \Omega^\bullet(\mathbb{R}^2) \otimes \Omega^{0,\bullet}(\mathbb{C}_z) \otimes \begin{pmatrix} \Omega^{0,\bullet}(\mathbb{C}_{w_1})[\varepsilon_2] \otimes \mathfrak{gl}(n) \\ \oplus \\ \Omega^{0,\bullet}(\mathbb{C}_{w_2})[\varepsilon_1] \otimes \mathfrak{gl}(m) \\ \oplus \\ T^*\mathrm{Hom}(\mathbb{C}^m, \mathbb{C}^n)[-1] \end{pmatrix}. \qquad (A.30)$$

We now determine the $L_\infty$-structure. This is as usual gotten by skew-symmetrizing the natural $A_\infty$-structure on $\mathrm{Hom}_\mathcal{C}(\mathcal{F}_1 \oplus \mathcal{F}_2, \mathcal{F}_1 \oplus \mathcal{F}_2)$. In terms of the above direct summands, the $A_\infty$-structure is given in terms of the following operations:

- $\mathrm{Hom}_\mathcal{C}(\mathcal{F}_i, \mathcal{F}_i) \otimes \mathrm{Hom}_\mathcal{C}(\mathcal{F}_i, \mathcal{F}_i) \to \mathrm{Hom}_\mathcal{C}(\mathcal{F}_i, \mathcal{F}_i)$ , $\quad A \otimes B \mapsto AB$ ;

- $\mathrm{Hom}_\mathcal{C}(\mathcal{F}_i, \mathcal{F}_j) \otimes \mathrm{Hom}_\mathcal{C}(\mathcal{F}_j, \mathcal{F}_i) \to \mathrm{Hom}_\mathcal{C}(\mathcal{F}_j, \mathcal{F}_i)$ , $\quad Y \otimes X \mapsto YX$ ;

- $\mathrm{Hom}_\mathcal{C}(\mathcal{F}_i, \mathcal{F}_i) \otimes \mathrm{Hom}_\mathcal{C}(\mathcal{F}_j, \mathcal{F}_i) \to \mathrm{Hom}_\mathcal{C}(\mathcal{F}_j, \mathcal{F}_i)$ , $\quad A \otimes X \mapsto AX$ ,

where $i, j = 1, 2, i \neq j$.

This induces the following $L_\infty$-structure:

- The first kind of $A_\infty$-operation above gives $L_\infty$-structures on $\mathrm{Hom}_\mathcal{C}(\mathcal{F}_i, \mathcal{F}_i)$ with

$$\ell_1 = \mathrm{d} \otimes 1_{\Omega^{0,\bullet}(\mathbb{C}^2)[\varepsilon_j]} \otimes 1_\mathfrak{g} + 1_{\Omega^\bullet(\mathbb{R}^2)} \otimes \bar{\partial} \otimes 1_\mathfrak{g}, \qquad (A.31)$$

$$\ell_2 = \wedge \otimes [-,-]_\mathfrak{g}, \qquad (A.32)$$

where $\mathfrak{g} = \mathfrak{gl}(m)$ for $i = 1$ and $\mathfrak{g} = \mathfrak{gl}(n)$ for $i = 2$. As explained in example A.3 these are precisely holomorphic–topological twists of the six-dimensional $\mathcal{N} = (1,1)$ vector multiplets for $\mathfrak{gl}(m)$ and $\mathfrak{gl}(n)$.

- There is a bracket

$$\left(\Omega^\bullet(\mathbb{R}^2) \otimes \Omega^{0,\bullet}(\mathbb{C}_z) \otimes T^*\mathrm{Hom}(\mathbb{C}^m, \mathbb{C}^n)[-1]\right)^{\otimes 2}$$

$$\to \Omega^\bullet(\mathbb{R}^2) \otimes \Omega^{0,\bullet}(\mathbb{C}_z) \otimes \begin{pmatrix} \Omega^{0,\bullet}(\mathbb{C}_{w_1})[\varepsilon_2] \otimes \mathfrak{gl}(m) \\ \oplus \\ \Omega^{0,\bullet}(\mathbb{C}_{w_2})[\varepsilon_1] \otimes \mathfrak{gl}(n) \end{pmatrix}, \quad (A.33)$$

explicitly given by wedging the form factors and taking the commutator of the matrices in $T^*\mathrm{Hom}(\mathbb{C}^m, \mathbb{C}^n)$.

- Finally, there is a bracket

$$\Omega^\bullet(\mathbb{R}^2) \otimes \Omega^{0,\bullet}(\mathbb{C}_z) \otimes \begin{pmatrix} \Omega^{0,\bullet}(\mathbb{C}_{w_1})[\varepsilon_2] \otimes \mathfrak{gl}(m) \\ \oplus \\ \Omega^{0,\bullet}(\mathbb{C}_{w_2})[\varepsilon_1] \otimes \mathfrak{gl}(n) \end{pmatrix}$$

$$\otimes$$

$$\Omega^\bullet(\mathbb{R}^2) \otimes \Omega^{0,\bullet}(\mathbb{C}_z) \otimes T^*\mathrm{Hom}(\mathbb{C}^m, \mathbb{C}^n)$$

$$\to \Omega^\bullet(\mathbb{R}^2) \otimes \Omega^{0,\bullet}(\mathbb{C}_z) \otimes T^*\mathrm{Hom}(\mathbb{C}^m, \mathbb{C}^n), \quad (A.34)$$

explicitly given by wedge product of forms and the natural action of $\mathfrak{gl}(m) \oplus \mathfrak{gl}(n)$ on $T^*\mathrm{Hom}(\mathbb{C}^m, \mathbb{C}^n)$.

The last of these brackets encodes the coupling between the hypermultiplets in the Kapustin twist and the twist of the six-dimensional $\mathcal{N} = (1, 1)$ vector multiplet. The second bracket, which breaks the $\mathbb{Z}$-grading down to a $\mathbb{Z}_2$-grading, encodes an extra gauge symmetry.

The open string field theory we have found can formally be regarded as four-dimensional Chern–Simons theory on $\mathbb{R}^2 \times \mathbb{C}_z$ for a dg Lie superalgebra. We may schematically encode the above brackets by writing the above dg Lie algebra as

$$
\left(
\begin{array}{c|c}
\Omega^{0,\bullet}(\mathbb{C}_{w_1})[\varepsilon_2] \otimes \mathfrak{gl}(m) & \mathrm{Hom}(\mathbb{C}^m, \mathbb{C}^n)[-1] \\
\hline
\mathrm{Hom}(\mathbb{C}^n, \mathbb{C}^m)[-1] & \Omega^{0,\bullet}(\mathbb{C}_{w_2})[\varepsilon_1] \otimes \mathfrak{gl}(n)
\end{array}
\right).
\tag{A.35}
$$

Let us now analyze the effect of the closed string field $w_1 w_2$. As we saw before, the image of this closed string field under the closed open map only affects the differential on the above dg Lie superalgebra. Explicitly, the deformation looks like

$$
\left(
\begin{array}{c|c}
\left( \Omega^{0,\bullet}(\mathbb{C}_{w_1})\varepsilon_2 \xrightarrow{w_1 \partial_{\varepsilon_2}} \Omega^{0,\bullet}(\mathbb{C}_{w_1}) \right) \otimes \mathfrak{gl}(m) & \mathrm{Hom}(\mathbb{C}^m, \mathbb{C}^n)[-1] \\
\hline
\mathrm{Hom}(\mathbb{C}^n, \mathbb{C}^m)[-1] & \left( \Omega^{0,\bullet}(\mathbb{C}_{w_2})\varepsilon_1 \xrightarrow{w_2 \partial_{\varepsilon_1}} \Omega^{0,\bullet}(\mathbb{C}_{w_2}) \right) \otimes \mathfrak{gl}(n)
\end{array}
\right)
$$
$$
\cong \left(
\begin{array}{c|c}
\mathfrak{gl}(m) & \mathrm{Hom}(\mathbb{C}^m, \mathbb{C}^n)[-1] \\
\hline
\mathrm{Hom}(\mathbb{C}^n, \mathbb{C}^m)[-1] & \mathfrak{gl}(n)
\end{array}
\right).
\tag{A.36}
$$

The remaining Lie brackets equip the above with with the structure of the Lie superalgebra $\mathfrak{gl}(m|n)$. Thus, we have found exactly four-dimensional Chern–Simons theory for the Lie algebra $\mathfrak{gl}(m|n)$ as claimed.

We note that the BRST transformations induced by the above Lie brackets are slightly different from those identified in the main body of the paper. This is an artifact of working with a particular model for the underlying $L_\infty$ algebra. For comparison, we explicate the BRST transformations below.

Note that the cochain complex underlying our $L_\infty$ algebra arises naturally as the totalization of a $\mathbb{Z} \times \mathbb{Z}/2$-graded cochain complex, where the fields valued in $\mathrm{Hom}(\mathbb{C}^m, \mathbb{C}^n) \oplus \mathrm{Hom}(\mathbb{C}^n \oplus \mathbb{C}^m)$ are placed in bidegree $(\bullet, 1)$. Though the lie brackets arising from the coupling of the hypermultiplets to the vector multiplets broke the $\mathbb{Z}$-grading down to a $\mathbb{Z}/2$-grading, these brackets are easily seen to preserve the above grading $\mathbb{Z} \times \mathbb{Z}/2$ grading.

We fix the following notation for components of our fields

$$
\alpha_{ij} \in \Omega^i(\mathbb{R}) \otimes \Omega^{0,j} \otimes (\mathfrak{gl}(m) \oplus \mathfrak{gl}(n)), \qquad \beta_{ij} \in \Omega^i(\mathbb{R}) \otimes \Omega^{0,j} \otimes (\mathrm{Hom}(\mathbb{C}^m, \mathbb{C}^n) \oplus \mathrm{Hom}(\mathbb{C}^n, \mathbb{C}^m))
\tag{A.37}
$$

and denote the corresponding linear operators the same way. The BRST variations determined by the then take the form

$$
Q\alpha_{ij} = d\alpha_{(i-1)j} + \bar{\partial}\alpha_{i(j-1)} + \sum_{a+c=i, b+d=j} [\alpha_{ab}, \alpha_{cd}] + \sum_{a+c=i, b+d=j} [\beta_{ab}, \beta_{cd}],
\tag{A.38}
$$

$$
Q\beta_{ij} = d\beta_{(i-1)j} + \bar{\partial}\beta_{i(j-1)} + \sum_{a+c=i, b+d=j} [\alpha_{ab}, \beta_{cd}].
\tag{A.39}
$$

The brackets in these equations are the relevant brackets on the $L_\infty$ algebra we've identified. It would be interesting to construct an explicit $L_\infty$ equivalence between the BV complex we have identified and the $L_\infty$ algebra consisting of the fields $\mathcal{A}, c, b, B$ and BRST transformations from section 169.

**Remark A.11.** As in remark A.10, we can consider the effect of the quadratic superpotential as a deformation of the entire category. The result is a six-dimensional topological string theory on $\mathbb{R}^4 \times \mathbb{C}$. The two stacks of D5-branes we have considered will localize to a stack of D4- and anti-D4-branes wrapping $\mathbb{R}^2 \times \mathbb{C}$. This set up is very reminiscent of the topological strings construction of three-dimensional Chern–Simons theory with gauge supergroup of [86] and should lend itself to a holographic realization of the Yangian of $\mathfrak{gl}(m|n)$ generalizing the analysis of [85]

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
