# Peer review of "Superspin chains from superstring theory"

_SciPost Physics, doi:SciPost Phys. 13, 083 (2022)_

## Round 1 · Referee Report · Anonymous (Referee 1) · 2022-4-6

Strengths

  1. The paper is extremely well-organized and well-written. Whenever needed, concise and pedagogical reviews are given both on integrability and the brane construction of supersymmetric gauge theories, most of which I found more accessible than the ones that can be found in the literature.
  2. Careful, detailed, and thorough analysis: detailed exposition is given on how the spin-chain description arises from the brane construction and the string duality. The authors are also careful on every detail and various subtle points (e.g. the difference between compact and non-compact representations.)

Weaknesses

  1. It certainly provides new perspectives on the Bethe/gauge correspondence and its relation to the 4d holomorphic Chern-Simons theory. However the analysis in this paper does not seem to lead to new results either in spin chain or supersymmetric gauge theories, at least for the moment. Of course, having a complete understanding on the subject is a stepping stone for the future discovery (and I personally feel that this will be the case), but it could be seen as a weak point of this paper.
  2. The main achievement of the paper is extending the previous results on the relation between the Bethe/gauge correspondence and the 4d holomorphic Chern-Simons theory to spin chains with super-group symmetries. It is certainly of interest from a mathematical point of view, but the super-spin chain is a rather exotic object, not commonly found in the condensed matter system. So the relevance and the importance of the paper to a wide physics community could be debated.

Report

This paper explains how the Bethe/gauge correspondence for the super spin chain proposed by Nekrasov can be understood from the 4d holomorphic Chern-Simons theory constructed by Costello. The key idea is to realize both in terms of intersecting branes and use the duality of string theory.

The results are new and important and the paper is well-written. So I think it is clearly above the scientific threshold for the publication in SciPost (and other journals). The only thing I would like to ask the authors is to clarify the point below.

Requested changes

  1. In subsection 3.4.2, the authors explain the so-called fermionic duality of spin chains using the Hanany-Witten effects of branes. However, in the duality frame they use (which is given in (3.13)), there seem to be no NS5 branes although they use the NS5 branes in the discussion. I would like to ask the authors to clarify this point.

  • validity: top
  • significance: high
  • originality: good
  • clarity: top
  • formatting: perfect
  • grammar: perfect

Author:  Nafiz Ishtiaque  on 2022-04-09  [id 2367]

(in reply to Report 1 on 2022-04-06)
Category:
remark
correction

We thank the referee for the well-considered report and for pointing out important typos in section 3.4.2 discussing fermionic dualities.

Indeed, in section 3.4.2 the duality frame in which fermionic duality corresponds to Hanany-Witten transitions is not exactly the one in eq. 114 (this equation no. refers to the scipost submission, in the current arxiv version it is the eq. 3.13 as the referee pointed out), rather it is the S-dual of it. In the S-dual frame, we have D3 branes ending on NS5s. Fermionic duality corresponds to swapping the positions of two adjacent NS5 branes with opposite gradings (i.e. wrapping different omega deformation planes), along with the D3 branes attached to them. This causes D3 branes and NS5 branes to intersect creating D1 branes according to the Hanany-Witten mechanism. We note importantly, that in section 3.4.2 when we say F1 branes are being created/destroyed in the transition we should also clarify that these F1s are the S-duals of the aforementioned D1s. These changes will be made in the final version of the paper.

Finally, we would like to address the first weakness of the paper, as mentioned by the referee. While we agree that we have not presented any new results in the language of superspin chains in this paper, we believe that on the supersymmetric gauge theory side our proposal for the quiver gauge theories that are dual to the noncompact superspin chains is new (eq. 73 in the scipost submission, or eq. 2.71 in the arxiv version). This, in particular, suggests new quiver varieties (Higgs branches of these quiver gauge theories) whose equivariant cohomology carries representations of super Yangians (conjecture 1 and 2 in section 1) -- this is a concrete new proposal that can be subjected to further mathematical investigations.

---

## Round 1 · Referee Report · Anonymous (Referee 2) · 2022-8-16

Strengths

1- very clear 2- very systematic

Weaknesses

1- collection of new and known results, not always clear which is which

Report

The paper certainly meets the requirements for publishing.
The authors are very clear and present precise statements establishing as facts statements that were more or less already present in the literature at the level of vague conjectures.
To my knowledge this is the most complete dictionary of the gauge-Bethe correspondence. It is to be hoped (expected?) that it will lead to progress either in the study of supersymmetric gauge theories, or in integrable models, or both.

---

## Editorial Decision

published